# Evaluation and intercomparison of wildfire smoke forecasts from multiple modeling systems for the 2019 Williams Flats fire

Xinxin Ye[1], Pargoal Arab[1], Ravan Ahmadov[2,3], Eric James[2,3], Georg A. Grell[3], Bradley Pierce[4], Aditya Kumar[4], Paul Makar[5], Jack Chen[5], Didier Davignon[6], Gregory R. Carmichael[7], Gonzalo Ferrada[7], Jeff McQueen[8], Jianping Huang[8], Rajesh Kumar[9], Louisa Emmons[10], Farren L. Herron-Thorpe[11], Mark Parrington[12], Richard Engelen[12], Vincent-Henri Peuch[12], Arlindo da Silva[13], Amber Soja[14,15], Emily Gargulinski[15], Elizabeth Wiggins[15], Johnathan W. Hair[15], Marta Fenn[15], Taylor Shingler[15], Shobha Kondragunta[16], Alexei Lyapustin[13], Yujie Wang[13], Brent Holben[13], David M. Giles[13], Pablo E. Saide[1,17]

1 Department of Atmospheric and Oceanic Sciences, University of California, Los Angeles, Los Angeles, CA, USA

2 Cooperative Institute for Research in Environmental Sciences, CU Boulder, Boulder, CO, USA

3 NOAA Global Systems Laboratory, Boulder, CO, USA

4 University of Wisconsin – Madison Space Science and Engineering Center, Madison, WI, USA

5 Air Quality Research Division, Environment and Climate Change Canada, Ontario, Canada

6 Canadian Meteorological Centre Operations, Environment and Climate Change Canada, Quebec, Canada

7 College of Engineering, University of Iowa, Iowa City, IA, USA

8 NOAA/NWS National Centers for Environment Prediction, Boulder, CO, USA

9 Research Application Laboratory (RAL), National Center for Atmospheric Research (NCAR), Boulder, CO, USA

10 Atmospheric Chemistry Observations and Modeling (ACOM) Laboratory, NCAR, Boulder, CO, USA

11 Washington State Department of Ecology, Lacey, Washington, USA

12 European Centre for Medium-Range Weather Forecasts (ECMWF), Reading, UK

13 NASA Goddard Space Flight Center, Greenbelt, MD, USA

14 National Institute of Aerospace, Hampton, VA, USA

15 NASA Langley Research Center, Hampton, VA, USA

16 Center for Satellite Applications and Research, NOAA, Boulder, CO, USA

17 Institute of the Environment and Sustainability, University of California, Los Angeles, Los Angeles, CA, USA

*Correspondence to*: Xinxin Ye (xinxinye@ucla.edu); Pablo E. Saide (saide@atmos.ucla.edu)

**Abstract.** Wildfire smoke is one of the most significant concerns of human and environmental health, associated with its substantial impacts on air quality, weather, and climate. However, biomass burning emissions and smoke remain among the largest sources of uncertainties in air quality forecasts. In this study, we evaluate the smoke emissions and plume forecasts from twelve state-of-the-art air quality forecasting systems during the Williams Flats fire in Washington State, U.S., August 2019, which was intensively observed during the Fire Influence on Regional to Global Environments and Air Quality (FIREX-AQ) field campaign. Model forecasts with lead times within one day are intercompared under the same framework based on observations from multiple platforms to reveal their performance regarding fire emissions, aerosol optical depth (AOD), surface $PM_{2.5}$, plume injection, and surface $PM_{2.5}$ to AOD ratio. The comparison of smoke organic carbon (OC) emissions suggests a large range of daily totals among the models with a factor of 20 to 50. Limited representations of the diurnal patterns and day-to-day variations of emissions highlight the need to incorporate new methodologies to predict the temporal evolution and reduce uncertainty of smoke emission estimates. The evaluation of smoke AOD (sAOD) forecasts suggests overall underpredictions in both the magnitude and smoke plume area for nearly all models, although the high-resolution models have a better representation of the fine-scale structures of smoke plumes. The models driven by FRP-based fire emissions or assimilating satellite AOD data generally outperform the others. Additionally, limitations of the persistence assumption used when predicting smoke emissions are revealed by substantial underpredictions of sAOD on 8 August 2019 mainly over the transported smoke plumes, owing to the underestimated emissions on the 7th. In contrast, the surface smoke $PM_{2.5}$ ($sPM_{2.5}$) forecasts show both positive and negative overall biases for these models, with most members presenting more considerable diurnal variations of $sPM_{2.5}$. Overpredictions of $sPM_{2.5}$ are found for the models driven by FRP-based emissions during nighttime, suggesting the necessity to improve vertical emission allocation within and above the planetary boundary layer (PBL). Smoke injection heights are further evaluated using the NASA Langley Research Center's Differential Absorption High Spectral Resolution Lidar (DIAL-HSRL) data collected during the flight observations. As the fire became stronger over 3-8 August, the plume height became deeper with the day-to-day range of about $2 - 9$ km a.g.l. However, narrower ranges are found for all models with a tendency of overpredicting the plume heights for the shallower injection transects and underpredicting for the days showing deeper injections. The misrepresented plume injection heights lead to inaccurate vertical plume allocations along the transects corresponding to transported one-day-old smoke. Discrepancies in model performance for surface $PM_{2.5}$ and AOD are further suggested by the evaluation of their ratio, which cannot be compensated by solely adjusting the smoke emissions but are more attributable to model representations of plume injections, besides other possible factors including the evolution of PBL depths and aerosol optical property assumptions. By consolidating multiple forecast systems, these results provide strategic insight on pathways to improve smoke forecasts.

**1 Introduction**

Wildfire is a natural ecological process that is necessary to maintain ecosystem structure and function (He et al., 2019; Pausas and Keeley, 2019), but also a crucial concern of public health, environment, and climate (Field et al., 2009; Jacobson, 2014; Kanda et al., 2001; Page et al., 2002; Reid et al., 2016). Smoke produced by fires is composed of considerable quantities of aerosols and trace gases originating from emissions of biomass combustion, including primary air pollutants such as particulate matter (PM), nitrogen oxides ($NO_x$), carbon monoxide (CO), and ammonia ($NH_3$), as well as trace metals and volatile organic compounds (VOCs). Composition of fire smoke evolves over time and space through complex chemical transformations, aerosol processes, and interactions with other atmospheric components (Sokolik et al., 2019), which lead to formation of secondary air pollutants such as $O_3$ and secondary aerosols (Baker et al., 2016). Air pollutants associated with smoke plumes, especially fine aerosol particles, can be transported over long distances and lead to degradation of local to regional air quality and harmful exposures over large areas (Colarco et al., 2004; Larsen et al., 2018). In recent decades, the risks posed by wildfires on human health and property have been costly and increasing in North America (McClure and Jaffe, 2018; Rappold et al., 2017), associated especially with the increases in severity and frequency of large wildfires and development of the urban-wildland interface over the western U.S. (Westerling, 2006; Williams et al., 2019) and owing mostly to anthropogenic climate change (Abatzoglou and Williams, 2016).

Numerical models of atmospheric chemistry and transport play an important role in advancing our understanding of the diverse impacts of wildfire smoke on air quality and the climate system, interpreting observed smoke plume characteristics, as well as providing valuable information on regulatory and health advisory purposes for decision-making during smoke events. Biomass burning emissions have been incorporated into many modeling systems to account for wildfire impacts in global operational or near-real-time (NRT) air quality forecasts (e.g. Inness et al., 2019; Pierce et al., 2007; Randles et al., 2017). Additionally, multiple regional air quality prediction systems across different regions in North America have also included smoke predictions (Ahmadov et al., 2017; Chen et al., 2019; Herron-Thorpe et al., 2014; Lee et al., 2017; Pavlovic et al., 2016). Effective performance of these air quality forecasting systems has been reported during fire seasons (e.g. Chen et al., 2019; Pan et al., 2020; Yuchi et al., 2016). However, many of the investigations that evaluate simulations of wildfire smoke are implemented in a retrospective way with the proxies for fire emissions already known, and only a few of them were aimed at demonstrating predictive skill and using different evaluation metrics. Also, most evaluations were performed for a single model or different versions of a similar system. Therefore, a multi-model intercomparison of fire smoke predictions by current modeling systems under a common framework is lacking.

Although advances have been made in a number of forecasting systems, biomass burning emissions and smoke are still within the greatest sources of uncertainties in air quality predictions (Carter et al., 2020; Kaiser et al., 2012; Pan et al., 2020b), relating to many challenges that remain unresolved. Firstly, wildfires occur sporadically in many places, thus the emissions are inherently unpredictable. Biomass burning emissions within a forecasting window are usually estimated by persistence, which means that the emissions estimated based on the latest satellite observations are assumed to persist in the forecasting window. However, emissions from wildfires depend on many factors, including meteorology, fuel conditions, combustion

stage, fire containment activities, etc., and thus can undergo substantial daily and diurnal variability (Saide et al., 2015). This limits the potential of smoke forecasts especially for large wildfires with drastic day-to-day changes in their behavior.

In addition to the assumption of persistence, the detection of fire activity and quantification of emissions are also challenging and highly uncertain (Darmenov and da Silva, 2013; Kaiser et al., 2012). This is associated with limitations of the spatiotemporal coverage and resolution of satellite measurements, as well as the complexity of fuels and combustion processes. Satellite-based fire emission estimates are primarily derived from satellite observations of burned area, active fire counts, and/or fire radiative power, with constraints from satellite retrievals of aerosol optical depth (AOD), CO, or $CO_2$ (Wiedinmyer
et al., 2011; Kaiser et al., 2012; Darmenov and da Silva, 2013; Li et al., 2019). While advances have been made in fire emission estimation with both top-down and bottom-up approaches, considerable uncertainties in fire emission inventories are reported in the literature. For instance, emissions of biomass burning aerosols differ by a factor of 4 to 7 over North America across inventories, driven mostly by dry matter differences (Carter et al., 2020). Sensitivity studies have shown substantial emission-related uncertainty in smoke forecasts and the radiative effect of carbonaceous aerosols, contributed by different spatiotemporal
distributions and magnitudes of fire emissions (Carter et al., 2020; Garcia-Menendez et al., 2014). This can substantially limit the accuracy of fire smoke forecasts and the potential of air quality models.

      Parameterization of plume injection height is another essential factor in the simulation and forecast of smoke transport, lifetime, and chemistry (Paugam et al., 2016). Plume injection heights are defined as the altitudes at which fire emissions are entrained into the boundary layer, the free troposphere, and even the lower stratosphere, resulting from the updrafts generated
by heat and buoyancy above fires (Freitas et al., 2007). Profound impact of plume injection height on transport of smoke constituents has been proven, as emissions injected into the free stable troposphere can be transported over long distances owing to stronger winds and fewer scavenging processes (Ansmann et al., 2018; Dirksen et al., 2009; Val Martín et al., 2006); on the other hand, plumes injected in the PBL are expected to have a much stronger effect on local air quality. The smoke plume injection processes are dependent on meteorological conditions and fire characteristics, which are both highly dynamic
and make the representation of injection heights challenging. A variety of plume rise models have been developed and implemented in CTMs to parameterize the vertical distribution of fire emissions by taking fire buoyancy and atmospheric conditions into account, including empirical-statistical approaches such as adapted formulations of stacks injections (Briggs, 1975, 1965; Pavlovic et al., 2016; Raffuse et al., 2012), methods considering microphysics and entrainment (Freitas et al., 2007) and fire-energy thermodynamics (Anderson et al., 2011; Chen et al., 2019; Sofiev et al., 2012), and integrated systems
that fully resolve plume dynamics (Mandel et al., 2014). Other approaches have simply considered arbitrary vertical distributions of fire emissions, such as a uniform vertical distribution below the modeled mixed layer height, or specified altitudes determined empirically (Val Martin et al., 2012). Previous studies have evaluated performance of these approaches. For example, the Briggs approach gives mostly injection heights below the PBL height (Mallia et al., 2018) and compares lower than the Multi-angle Imaging Spectro Radiometer (MISR) plume heights, because of the different processes controlling
the uplift of wildfire plumes compared to the plume rise of stacks (Raffuse et al., 2012; Sofiev et al., 2012). Former studies are

generally performed for retrospective cases or offline comparisons against satellite measurement of plume heights, and the performance of diverse plume injection parameterizations deployed in smoke forecasts is yet to be intercompared.

Air quality and smoke forecasting data from multiple modeling systems were collected during the NOAA/NASA Fire Influence on Regional to Global Environments and Air Quality (FIREX-AQ) field campaign, which provides a unique opportunity to extensively understand the status and prospects on wildfire smoke forecasting. The FIREX-AQ field campaign (https://www.esrl.noaa.gov/csd/projects/firex-aq/) took place in the late summer of 2019 from 21 July to 5 September. This comprehensive field investigation provided detailed airborne and ground-based observations of the chemistry, composition, fuel, and meteorology for smoke from wildfires and agricultural fires across the continental U.S., which provides an opportunity to validate and improve real-time smoke forecasts. The air quality forecasting data from multiple systems were used to support the flight planning, including centralized collection and archival of information from major groups providing smoke forecasts covering the continental U.S. More importantly, a variety of observation platforms were involved, including four instrumented research aircrafts, satellites, and ground-based stationary and mobile laboratories. Particularly, the NASA DC-8 aircraft flying laboratory deployed to Boise, ID and Salina, KS from 17 July to 5 September 2019 and collected in-situ and remote sensing measurements from multiple fires. The Differential Absorption Lidar-High Spectral Resolution Lidar (DIAL-HSRL) (Hair et al., 2018) on board the DC-8 collected observations of aerosol optical properties profiles, which is a valuable dataset to validate model predictions of plume structure and injection.

In this paper, we present the evaluation and intercomparison of the forecasts of fire emissions and smoke plume from twelve global and regional air quality forecasting systems that represent the state of the art, with the aim of enhancing our knowledge about the main factors controlling their performance. The evaluation is carried out focusing on the Williams Flats fire that occurred in August 2019, Washington State, the U.S. to demonstrate the forecasting performance in multiple dimensions, including fire emissions, total column and surface aerosol loading, and plume injections. In section 2, we describe the modeling systems with a brief overview of their primary differences. Section 3 provides the analysis of how the model forecasts perform by comparisons to satellite-derived AOD, surface observations of $PM_{2.5}$ concentrations, and airborne observations of vertical plume structures and plume heights. We also investigate the joint performance for surface $PM_{2.5}$ and total column smoke aerosols, as indicated by AOD. The summary and conclusions are presented in section 4, along with discussions on pathways to improve the accuracy and tackle the challenges in smoke forecasting.

## 2 Descriptions of forecast models

Twelve forecast systems that provide fire smoke predictions are incorporated within the following intercomparison, including three global and nine regional systems. A summary of the model descriptions can be found in Table 1, and the domains of the models are shown in Fig. 1. The evaluation here is restricted to an area between 44.0° N - 50.0° N and 110.0° W

- 122.0° W, focusing on the smoke plumes from the Williams Flats fire. Description of the fire event is given in section 3. Although some of these forecast systems produce multiple forecasting cycles a day, only one cycle a day was used in this study, as denoted by the initial time in Table 1. In the following, the forecast systems are described separately in brief (section 2.1), along with a summary of main differences in their numerical methodology, especially regarding biomass burning emissions and plume rise parameterizations (section 2.2).

## 2.1 Forecast models

### 2.1.1 CAMS

The Copernicus Atmosphere Monitoring Service (CAMS) is operated by the European Centre for Medium-Range Weather Forecasts (ECMWF) on behalf of the European Commission and provides global atmospheric composition forecasts using the ECMWF Integrated Forecast System (IFS) (Benedetti et al., 2009; Flemming et al., 2015; Inness et al., 2019), which provides 5-day forecasts of atmospheric composition including reactive gases, aerosols, and greenhouse gases (http://atmosphere.copernicus.eu/) at an effective horizontal resolution of 0.4 degrees. Emissions from global anthropogenic activities are provided by the CAMS_GLOB_ANT v2.1 inventory (Granier et al., 2019). Emissions of organic matter (OM), black carbon (BC), and $SO_2$ from fires are obtained using the Global Fire Assimilation System (GFAS) v1.2 based on the Moderate Resolution Imaging Spectroradiometer (MODIS) observations of fire radiative power (FRP) (Kaiser et al., 2012). Plume injection height is parameterized with a method derived from MISR fire plume observations (Sofiev et al., 2012). The diagnostics of total and fine-mode AOD use bulk optical properties for each aerosol species that have been pre-computed with a standard code for Mie scattering (Bozz et al., 2020; Wiscombe, 1980). Specifically, the smoke OM aerosol properties are based on the "continental" mixtures and BC properties are based on soot model described in the Optical Properties of Aerosols and Clouds (OPAC) database (Bozzo et al., 2020; Hess et al., 1998). For the hydrophilic types, the optical properties changes with the relative humidity (RH) for the refractive index and density based on Koepke et al. (1997), and the size distribution is modified by applying growth factors (Table A2 in Bozzo et al., 2020) to the mode radius. MODIS AOD data (550 nm) with a variational bias correction applied (Dee and Uppala, 2009) are routinely assimilated in a 4D-Var framework using aerosol total mixing ratio as the control variable (Benedetti et al., 2009). Retrievals of reactive gases are also assimilated, including ozone, carbon monoxide, formaldehyde, and nitrogen dioxide (Inness et al., 2015, 2019).

### 2.1.2 GEOS-FP

GEOS-FP (GEOS "Forward Processing") is a near-real time (NRT) forecast system led by the NASA Global Modeling and Assimilation Office (GMAO). It provides NRT forecasts of meteorological fields, aerosols, and tracers globally and twice a day for a period of 120 hours with the grid resolution of about 25 km (0.25° in latitude and 0.3125° in longitude). GEOS-FP uses the same modeling configuration as the Modern-Era Retrospective Analysis for Research and Applications, version 2 (MERRA-2) (Randles et al., 2017). GEOS-FP uses the Goddard Chemistry Aerosol Radiation and Transport (GOCART)

aerosol model (Chin et al., 2002; Colarco et al., 2010) which includes simplified sulfur chemistry and tracks aerosol mass mixing ratio of dust, sea salt, hydrophobic and hydrophilic types of BC and organic carbon (OC), and sulfate. Biomass burning emissions are from the Quick Fire Emission Dataset (QFED) v2.4 (Koster et al., 2015). By employing the FRP observations from MODIS, QFED uses FRP-to-emission coefficients adjusted to improve model agreement with AOD estimates. Smoke emissions are distributed within the PBL. While dust and sea-salt emissions are wind-driven (Randles et al., 2017), anthropogenic aerosol emissions are obtained from annually varying global datasets (Diehl et al., 2012). Aerosols are assumed to be externally mixed in modes of fixed mean diameter and standard deviation (Colarco et al., 2014; Randles et al., 2017). The AOD is diagnosed by integrating the extinction over the column and aerosol species. The species-specific mass extinction coefficients are derived from Mie theory for spherical particles (Wiscombe 1980) or the T-matrix approach using the updated optics for nonspherical dust as described in Meng et al. (2010). The extinction coefficients for sulfate and hydrophilic carbonaceous aerosol are a function of RH following Chin et al. (2002). RH affects both the index of refraction and the size distribution. Assumed optical properties are primarily from the Optical Properties of Aerosols and Clouds (OPAC) dataset (Hess et al. 1998) with updated dust optical properties that incorporate nonsphericity (Meng et al. 2010; Colarco et al. 2014). GEOS-FP also includes data assimilation of satellite AOD retrievals corresponding to bias-corrected AOD estimates using MODIS radiances and a Neural-network framework (Albayrak et al., 2013).

### 2.1.3 RAQMS

The Realtime Air Quality Modeling System (RAQMS) is a forecast system that provides global prediction of aerosol and reactive gases at 1-degree resolution (Pierce et al., 2007, 2009). It uses EDGAR anthropogenic emissions, and biomass burning emissions are estimated using gridded carbon fuel consumption databases, MODIS fire detections, fire weather severity index, and published emission ratios (Petrenko et al., 2012; Soja et al., 2004). RAQMS forecasts are initialized with assimilation of OMI and MLS ozone and MODIS AOD retrievals using a statistical digital filter (Pierce et al., 2007). The same settings for aerosol optical properties and AOD calculation as used in WISC WRF-Chem for is employed in this model (see section 2.1.5).

### 2.1.4 HRRR-Smoke

The High-Resolution Rapid Refresh coupled with smoke (HRRR-Smoke) is an operational smoke forecasting system run by NOAA/NWS. It is based on NOAA's weather forecasting model HRRR (https://rapidrefresh.noaa.gov/hrrr) and a coupled meteorology-chemistry model WRF-Chem (Grell et al., 2005). HRRR-Smoke simulates primary aerosols from wildland fires in real time on a 3-km resolution grid over the entire continental U.S. (Ahmadov et al., 2017). It ingests FRP data from the MODIS sensor on Terra and Aqua satellites and the Visible Infrared Imaging Radiometer Suite (VIIRS) sensor on the Suomi National Polar-orbiting Partnership (S-NPP) and NOAA-20 to calculate fire size, heat flux, and fire emissions. Fire smoke is treated as a chemically inert tracer and no other aerosol sources are included in this model. Both dry and wet deposition processes of the smoke tracer are represented. The model also includes direct feedback of the smoke aerosol on radiation. The AOD at 550 nm is diagnosed using the mass extinction coefficient of 4.5 $m^2g^{-1}$. In addition, fire size and heat flux determined

by FRP data is used to calculate plume injection heights for the flaming emissions using concurrently simulated meteorological fields by the model (Freitas et al., 2007; Grell and Baklanov, 2011; Paugam et al., 2015). The RAP-Smoke model (13.5-km resolution), which covers the entire North America, provides lateral boundary conditions (LBCs) of smoke concentrations to HRRR-Smoke (Ahmadov et al., 2019).

### 2.1.5 WISC WRF-Chem

WISC WRF-Chem is an experimental regional forecast system led by University of Wisconsin, which runs in near-real time with its 8-km grid-resolution domain nested into RAQMS. The Goddard Chemistry GOCART is used as the aerosol scheme (Chin et al., 2002). The AOD at 550 nm is computed by vertical integration of the aerosol extinction. Changes in the optical properties of OC associated with hygroscopic growth is accounted for, with the mole fraction and density values being used from look up tables and are a function of RH (OC mole fraction values range from 1 to 0.01 for RH values of 1% to 99% respectively). Extinction efficiencies are used as a function of mole fraction with values ranging from 1.37 to 2.18 for mole fraction values of 0.01 to 1. The initial and boundary conditions for aerosols are provided by the RAQMS (Pierce et al., 2007). Meteorological initial and boundary conditions are provided by the NOAA Global Forecast System (GFS) V15 release, which uses the Finite-volume Cubed-Sphere (FV3) dynamic core. The AOD assimilation is not performed within the model but is included through the RAQMS initial conditions. Smoke emissions were estimated using the geostationary fire detections from the GEOS-15 and the Brazilian Biomass Burning Model (3BEM), which is a bottom-up biomass burning emission estimation approach included in the PREP_CHEM_SRC emissions preprocessor (Freitas et al., 2011; Pierce et al., 2009).

### 2.1.6 UCLA WRF-Chem

UCLA WRF-Chem provided experimental forecasts during the FIREX-AQ field campaign at a spatial resolution of 4 km over the western U.S. The model is based on the WRF-Chem v3.6.1 and configured with a simplified aerosol-aware microphysics scheme (Saide et al., 2016; Thompson and Eidhammer, 2014) to reduce computational costs compared to full-chemistry runs. The system was deployed successfully in a previous NASA field campaign targeting smoke from fires (Redemann et al., 2021). The meteorological initial and boundary conditions are acquired from the 12-km North American Model (NAM) Non-hydrostatic Multiscale Model (Janjic and Gall, 2012). Two categories of aerosols are considered, i.e. water and ice friendly aerosols, which are non-reactive and only undergo wet deposition. The smoke aerosols are considered to be fully contained in the water friendly aerosols. Ambient aerosol extinction and AOD at 550 nm are computed based on the two aerosol tracers (water and ice friendly aerosols) using fixed RH-dependent mass extinction efficiencies to consider aerosol hygroscopic growth. Dry extinction is computed using RH of 20% and it is used to estimate $PM_{2.5}$ concentrations (using the mass extinction efficiency of 3.5 $m^2 g^{-1}$).

Smoke emissions at 0.1° resolution are obtained from QFED v2.4 and processed using the fire_emiss preprocessor (https://www2.acom.ucar.edu/wrf-chem/wrf-chem-tools-community). Each grid cell from which the fire smoke is released is

assigned a burned area of 0.25 km$^2$ in the plume rise parameterization scheme (Freitas et al., 2007, 2010). The same scheme was used in NCAR WRF-Chem, UIOWA WRF-Chem, and WISC WRF-Chem. Besides, this model ingests MODIS AOD observations on Terra and Aqua satellites to provide constraints on fire emissions in near-real time using a inversion modeling framework (Saide et al., 2015, 2016). The inversion scheme optimizes emissions for six fire complexes with the largest average emissions over the last four days.

### 2.1.7 UIOWA WRF-Chem

The University of Iowa provided air-quality forecasts based on WRF-Chem v3.9.1 (UIOWA WRF-Chem) (http://bio.cgrer.uiowa.edu/FIREX-AQ/model_info.html) using the Model for Ozone and Related Tracers-4 (MOZART-4) (Emmons et al., 2010) with aqueous chemistry as the chemistry scheme and the Model for Simulating Aerosol Interactions and Chemistry (MOSAIC) (Fast et al., 2006; Gao et al., 2016; Zaveri et al., 2008) for aerosols. MOSAIC uses a sectional representation of aerosol size distribution, with detailed aerosol interactions with radiation and clouds described by Chapman et al. (2009). The aerosol size distributions are described in eight size bins, and the chemical constituents of the aerosol are assumed to be internally mixed within each bin, and externally mixed between bins. The optical properties of each bin are calculated using the Mie code (Toon and Ackerman, 1981), then their summations over all bins give the bulk optical properties of the aerosol population (Fast et al., 2006; Barnard et al., 2010). Aerosol hygroscopicity is considered by calculating aerosol water content using the Zdanovskii-Stokes-Robinson (ZSR) method (Zdanovskii, 1948; Stokes and Robinson, 1966). The model included tracers with no lifetime for understanding the impact of processes such as smoke plume rise. The GFS data was used for the meteorological initial and boundary conditions, and the Whole Atmosphere Community Climate Model (WACCM) (Marsh et al., 2013) forecasts for chemical species and aerosols. Biomass burning emissions are also obtained from the QFED v2.4 and the plume rise was also parameterized using the method of Freitas et al. (2007, 2010).

### 2.1.8 NCAR WRF-Chem

The NCAR WRF-Chem is a regional forecast system (https://www.acom.ucar.edu/firex-aq/) led by NCAR Atmospheric Chemistry Observations and Modeling (ACOM) using WRF-Chem v3.9.1 (Kumar et al., 2021). The Meteorological IC and BC are provided by the GFS model. Biomass burning emissions are produced each day using the near-real time Fire Inventory from NCAR (FINN) based on MODIS Rapid Response fire counts (Wiedinmyer et al., 2011). Plume rise by Freitas et al. (2007, 2010) is used to distribute the fire emissions vertically. The model uses MOZART for gas-phase chemistry and GOCART for aerosol processes, with the chemical IC and BC coming from the WACCM forecasts. The aerosol optical properties are calculated using the parameterized Mie theory (Ghan and Zaveri, 2007) with the hygroscopicity parameter of OC of 0.14.

### 2.1.9 NAQFC

The National Air Quality Forecasting Capability (NAQFC) model is developed by NOAA/Air Resources Laboratory (ARL) and National Centers for Environmental Prediction (NCEP) to provide operational 48-hour air quality prediction over the U.S. (Lee et al., 2017; Pan et al., 2020a) at 12-km resolution. It is based on the U.S. Environmental Protection Agency (EPA) Community Multi-scale Air Quality (CMAQ) v5.02 (Byun and Schere, 2006) modeling system, and is off-line driven by the 12-km North American Model (NAM) Non-hydrostatic Multiscale Model (Janjic and Gall, 2012). It uses the U.S. EPA National Emission Inventory (NEI) 2014v2 and correction factors based on satellites to account for trends in $NO_x$ emissions (Tong et al., 2015). Wildfire emissions, thermal, and speciation characteristics are determined in near-real time by the U.S. Forest Service BlueSky smoke emission package (Larkin et al., 2009) and the NOAA/NESDIS Hazard Mapping System (HMS) for fire location and strength (Ruminski and Kondragunta, 2006). The BlueSky wildfire heat flux was used in the Briggs equation (Briggs, 1975) to determine smoke plume injection heights. Monthly averaged concentrations for 36 gaseous and aerosol species obtained from GEOS-Chem were used as lateral boundary conditions below 7-km altitude, and a clean-air scenario static condition was used between 7 km and model top. NAQFC used the Carbon-Bond 2005 (CB05) (Yarwood et al., 2005) for gas-phase mechanism and followed largely the EPA's AERO4 module for aerosol processes (Binkowski and Roselle, 2003) with the related emission and removal processes in CMAQ v4.6. The AERO4 represents particle size as three modes. The processes of coagulation, particle growth and new particle formation are included. Extinction of aerosols is represented using a parametric approximation to Mie extinction (Binkowski and Roselle, 2003; Byun and Ching, 1999; Evans and Fournier, 1990). Then the AOD is derived by integrating the Mie extinction coefficients over the column.

### 2.1.10 AIRPACT

The Air Information Report for Public Awareness and Community Tracking (AIRPACT) v5 is an air quality prediction system primarily for Idaho, Oregon, and Washington (Herron-Thorpe et al., 2014). Meteorology fields predicted by the University of Washington WRF (Skamarock et al., 2008) model at 4-km resolution are used to drive CMAQ v5.02, which accounts for the chemical and physical processes of air components, including emissions, transport, vertical mixing, dilution, rain-out and deposition. The CMAQ model includes the CB05 and AERO6 chemical and aerosol mechanisms. Aerosol extinction at 550 nm is estimated using the approximation of the Mie extinction efficiency (Binkowski and Roselle, 2003; Byun and Ching, 1999), with the reference refractive indices of aerosols based on the OPAC database (Hess et al., 1998). The model assumes that organics influence neither the water content nor the ionic strength of the aerosol particles; only the equilibrium of the sulfate, nitrate, ammonium and water system is considered (Binkowski and Roselle, 2003). The chemical boundary conditions are derived from the WACCM to account for long-range transport of pollutants from outside the domain. The WACCM simulations incorporates fire emissions from the FINN data (Wiedinmyer et al., 2011). Fire emissions in AIRPACT are derived from the BlueSky framework with fire locations determined by the Satellite Mapping Automated Reanalysis Tool for Fire Incident Reconciliation (SMARTFIRE) v2. Plume rise is represented using a modified WRAP DEASCO3 method (Mavko and Morris, 2013).

### 2.1.11 FireWork

Environment and Climate Change Canada (ECCC) operates the Regional Air Quality Deterministic Prediction System (RAQDPS), which uses the Global Environmental Multi-scale - Modelling Air quality and CHemistry (GEM-MACH) model (Moran et al., 2010) v5.0 with full description of atmospheric chemistry and meteorological processes to provide 48-hours forecasts at 10-km resolution. The Forest Fire Smoke Model (FireWork) operational system is run as an additional version of the RAQDPS by including smoke emissions (Chen et al., 2019; Pavlovic et al., 2016). Gas-phase chemistry is accounted for by using the Young and Boris scheme, and aerosols processes are represented by the Canadian Aerosol Module (CAM) (Gong et al., 2003) with two bins. Aerosol optical properties within the model are decoupled from the meteorological model components – the model's radiative transfer routines make use of climatological aerosol radiative properties in its radiative transfer processes. AODs from the FireWork forecast were generated using post-processing of the model outputs. The operational 2-bin model output was first mapped to a 12-bin representation based on an assumed fixed size distribution, then optical properties were calculated based on speciated $PM_{2.5}$ (for nitrate, sulfate, OC, BC, dust and seasalt) using formula from IMPROVE. Biomass burning emissions are computed by the Canadian Forest Fire Emission Prediction System (CFFEPS) v2.03 (Chen et al., 2019), which is a bottom-up system linked to the Canadian Wildland Fire Information System (CWFIS) (Lee et al., 2002) with the hourly changes in biomass fuel consumption parameterized considering forecasted meteorology at fire locations. The fire-activity information is based on initial NRT fire hotspot data from three satellite sensors, including the Advanced Very High Resolution Radiometer (AVHRR), MODIS, and the Visible Infrared Imaging Radiometer Suite (VIIRS). Smoke emissions and the energy generated from wildfires is a product of burned area with diurnally adjusted fire growth rates (Lawson et al., 1996), fuel consumed per unit area calculated considering meteorology, combustion stages and burn durations, and emission factors following the literature (Urbanski, 2014). The forecasts initialized at 12:00 UTC are used in the evaluation, which ingested the most recent data of current day's hotspot by the initialization time. A plume rise parameterization based on fire-energy thermodynamics is used to define the smoke injection height and the vertical distribution of emissions (Anderson et al., 2011; Chen et al., 2019).

### 2.1.12 ARQI

An experimental version of GEM-MACH at 2.5 km resolution is maintained by the Air Quality Modelling and Integration (ARQI) group within ECCC (Makar et al., 2021). This domain is nested within the 10-km operational FireWork domain to generate forecasts for Alberta and Saskatchewan, Canada continuously since 2012, and has been used during several field campaigns. In support of the FIREX-AQ, the experimental forecast system was set up by ECCC with the domain covering the northwest U.S. and southwest Canada to track smoke flows from and towards Canada. In order to make these forecasts available during the forecasting window used in FIREX-AQ, ARQI was initialized at 12:00 UTC on the previous day of the forecast day, using the previous day 03:00 UTC update of CFFEPS emissions. Thus, these forecasts had smoke emissions that were nearly behind by one day compared to the rest of the forecasting systems, and this may account for some of the

discrepancies between the ARQI and FireWork forecasts, both of which used the same forest fire emissions processing framework. The KPP model with Rodas3 solver is used for gas-phase chemistry. Aerosol processes are also depicted by the CAM module with 12 bins. AOD is generated using an on-line Mie scattering approach assuming homogeneous spherical aerosols (Bohren and Huffman, 1983). Similarly as to FireWork, the biomass burning emissions are estimated by the CFFEPS v4.0 following (Chen et al., 2019) but with some modifications: (1) Plume rise is calculated using the full GEM-MACH vertical resolution of temperature for radiative balance; (2) A 24-hour off-line simulation using an a priori GEM forecast is used to create spin-up conditions for the GEM-MACH 2.5-km simulation, but the forest fire emissions used in the model were calculated by CFFEPSv4.0 on-line within GEM-MACH. This allowed the forest fire emissions of particulate matter to modify the weather within the simulation, in turn modifying the forest fire plume rise behavior on subsequent time steps. A more detailed description of the ARQI model and these feedback effects may be found in Makar et al (2021).**2.2 Main differences of the forecast systems**

The modeling systems included in this work represent a considerable diversity in forecasts accounting for fire smoke over the western U.S. Besides the spatial coverages, resolutions, and the driving meteorology, they differ in several major aspects:

1) Biomass burning emission input. Diverse approaches of quantifying emissions are involved, which can be broadly categorized into top-down estimates based on satellite FRP and FRP-to-emission coefficients, and bottom-up estimates based on fire detections (hotspots reflecting burned area), fuel categories, and emission factors.

2) Plume injection. Smoke emissions are distributed within the PBL for GEOS-FP and RAQMS (severity dependent), while plume rise parameterizations are employed in the other models. All the WRF-Chem simulations used the default version of the plume rise parameterization within WRF-Chem (Freitas et al., 2007, 2010), HRRR-Smoke and CAMS used satellite FRP observations in their parameterizations , while ARQI (Makar et al., 2021) and FireWork (Chen et al., 2019) used hourly calculated fire energy and thermodynamic (temperature) profile at hotspot locations.

3) Diurnal cycle of smoke emissions. Diverse patterns are adopted in the systems from nearly flat profiles to strong diurnal variations, which will be shown in the comparison. Most models use fixed diurnal profiles, except for UCLA WRF-Chem for which the pattern can be modified by the inverse modeling of fire emissions.

4) Initialization time. The forecasts considered here were mostly initialized at 00:00 UTC or 12:00 UTC, which leads to different time of validity for satellite observations and thus fire emissions in the models, depending on the latency of data by initialization times.

5) Complexity of chemical mechanisms. Smoke chemistry is treated in different ways, ranging from tracers in HRRR-Smoke and UCLA WRF-Chem, simplified GOCART (Chin et al., 2002) chemistry in GEOS-FP, NCAR WRF-Chem, and WISC WRF-Chem, and full chemistry by other models that may have differences in their treatment of organic aerosol (e.g. ARQI and FireWork).

6) Assimilation of satellite AOD data. All the three global operational forecasting systems incorporated assimilation of satellite AOD data. For the regional models, WISC WRF-Chem was initialized with assimilated aerosol fields from RAMQS, and UCLA WRF-Chem used AOD retrievals to constrain fire emissions. Other models did not make use of chemical data assimilation in their systems.

7) Treatment of aerosol processes and assumptions of aerosol physical and chemical properties, which are relevant to aerosol optical properties calculation. The direct aerosol feedbacks linked to radiative forcing are considered in many models (CAMS, GEOS-FP, ARQI, HRRR-Smoke, UCLA WRF-Chem, UIOWA WRF-Chem, WISC WRF-Chem, ARQI, and NCAR WRF-Chem), while the indirect feedbacks are enabled in three models (UIOWA WRF-Chem, UCLA WRF-Chem, and ARQI). Regarding the diagnosis of AOD, UCLA WRF-Chem, HRRR-Smoke, WISC WRF-Chem, RAQMS, and GEOS-FP used mass-concentration-based aerosol extinction, while CAMS, UIOWA WRF-Chem, NCAR WRF-Chem, NAQFC, AIRPACT, and ARQI used the Mie theory, and FireWork used a diagnostic post-processing calculation to estimate AOD from model-generated aerosols. The uncertainty of AOD calculations owing to the different methods and assumptions about the mixing state, density, refractive index, and hygroscopic growth of aerosols has been estimated as 30 – 35 % (Curci et al., 2015), with the general tendency for the different AOD methods to result in negative biases relative to satellite observations. This uncertainty introduced by these factors are not treated explicitly in this work.

8) Boundary conditions of chemical compositions. The chemical LBCs are critical to the representation of components transported from outside the forecasting domain. UIOWA WRF-Chem, WISC WRF-Chem, NCAR WRF-Chem and AIRPACT use LBCs from global forecasts, while monthly climatological fields are used for FireWork, ARQI, NAQFC, and UCLA WRF-Chem. HRRR-Smoke takes chemical LBCs from another regional model, i.e., RAP-Smoke.

## 3 Evaluation of smoke forecasts for the Williams Flats Fire

The Williams Flats Fire was the largest wildfire event sampled during the FIREX-AQ field campaign. In this paper, we focus the model evaluations on this event. First reported at about 03:23 PDT (Pacific Daylight Time, or UTC-7) 2 August 2019, the fire was ignited by various lightning strikes related to an early morning thunderstorm in the Colville Reservation, Washington State, located about 8 km to the southeast of Keller, WA, and 80 km to the northwest of Spokane, WA. The high-pressure system over the fire area produced above normal temperature and low humidity, which in combination with the wind pattern resulted in spreading of the blaze to the north bank of the Columbia River. The fire event led to numerous evacuation orders in the area. Fuel burned in the fire event includes short grass, timber, and tall brush. Containment efforts were somewhat hampered by steep terrain, limited access, and primitive road conditions near the fire. The storm that occurred on 9-10 August brought large amounts of rain over the fire, which caused localized flash flooding washing out several roads. Although the storms caused challenges for firefighting efforts, the cooler temperature and rain dramatically moderated the fire behavior, and

the percentage of containment reached 65% on 13 August (https://inciweb.nwcg.gov/incident/6493/). According to the final Incident Status Reports (ICS-209) report, prepared by the Bureau of Indian Affairs for the Williams Flats fire, the unit burned a total of 44,446 acres.

The burned area measurements from the USDA Forest Service National Infrared Operations (NIROPS) Unit with airborne thermal infrared (IR) imaging are used to present the evolution of this event. As shown in Fig. 2, a slightly increasing trend in the daily burned area growth can be seen on 4-6 August, and on 7 August the fire expanded abruptly with the daily increment of burned area being almost triple of that on 6 August. Therefore, the predicted burned area growth by assuming persistence, namely maintaining the same burned area growth on the previous day, may lead to significant underprediction of the fire

expansion on the 7[th] and overestimation on the 8[th]. Considering the temporal evolution of this event, the 7-day period of 3-9 August 2019 is of interest in this evaluation, which corresponds to the actively expanding stage of the fire with intense emissions and abundant observations. Evaluation of the models' performance is carried out from multiple perspectives, including (1) fire emissions and their diurnal variation pattern, (2) total column aerosol loading via comparisons against satellite AOD retrievals, (3) surface air quality impact via comparisons against in-situ $PM_{2.5}$ measurements, and (4) vertical

plume structure and fire plume injection compared against airborne lidar observations. Additionally, further discussion is presented on the surface $PM_{2.5}$ to AOD ratio and possible ways to reduce the discrepancies in performance between the two terms.

**3.1 Comparison of fire emissions**

In this section, the biomass burning (BB) emissions from the Williams Flats fire that are used in models are intercompared

in terms of the evolution of daily emissions and diurnal variation patterns. Figure 3 shows the comparison of daily total BB organic carbon (OC) emissions for eight models. Due to data availability, not all models could be included. Emissions from each forecasting system are derived by aggregating emissions at grid pixels representing the Williams Flats fire from 00:00 PDT to 23:00 PDT per day, with the value for each hour derived from the latest forecast cycle. An illustration of the grid pixels on the emission distribution maps are included in Fig. S1. Moreover, the emission estimates derived by a detailed analysis

developed by the FIREX-AQ Fuel2Fire group (https://www-air.larc.nasa.gov/missions/firex-aq/index.html) is also shown in Fig. 3. As a bottom-up emissions dataset, the BB emissions are derived using Fuel Characteristic Classification System (FCCS) fuelbeds, VIIRS, Geostationary Operational Environmental Satellite (GOES), MODIS, and ground-based intelligence (Soja et al., 2004), and provided at a temporal resolution of 1 s for both flaming and smoldering conditions. We converted the total carbon emissions into carbonaceous aerosol emissions (OC and BC) using a fixed percentage of 2% (Soja et al., 2004), and

then extracted the partition of OC emissions following an up-to-date relationship between the ratio of OC and BC emission factors ($EF_{OC}/EF_{BC}$) and Modified Combustion Efficiency (MCE) for western U.S. wildland fuels (Jen et al., 2019), with the MCE assumed to be 0.84 and 0.95 for smoldering and flaming emissions, respectively.

The result indicates a large spread of the daily total BB OC emissions within forecasting systems, with the differences in these estimates ranging from a factor of about 20 to 50 on 5-9 August (Fig. 3). The factors on the days before and after are even higher, owing to the different pace of the models ingesting satellite fire detections. Meanwhile, the FRP-based emissions that were used by UCLA WRF-Chem, UIOWA WRF-Chem, GEOS-FP, HRRR-Smoke, and CAMS tend to be overall higher than the hotspot-based emissions by a mean factor of 5.6. This is especially the case for the QFED emissions used by UCLA & UIOWA WRF-Chem and GEOS-FP, which are 6.4 times higher than the hot-spot based emissions on average. The Fuel2Fire emissions are within these two categories. The emission estimates tend to show an increasing trend throughout the days, which is consistent with the typical fire behavior and Fuel2Fire emissions. The large spread in the emission estimates could be due to multiple factors, including differences in the identification of ecosystems or fuel load, land cover classification, type of satellite fire detections (e.g., different sensors and pixel sizes), and the method and timing of ingesting satellite observations. Future work needs to be performed to understand the large spread between the smoke emissions and to reduce their uncertainty.

Diurnal factors of the smoke emissions are evaluated against the observed patterns derived by the GOES-17 Wildfire Automated Biomass Burning Algorithm (WFABBA) FRP product generated by the Cooperative Institute for Meteorological Satellite Studies (CIMSS) at the University of Wisconsin, Madison. As shown in Fig. 4, the FRP data exhibits discernable diurnal fire activities, which peaked towards late afternoon (14:00-19:00 PDT) with a substantial day-to-day variability. By contrast, most models assumed fixed diurnal profiles. A variety of patterns are found for the models, which show relatively smaller variations, e.g., GEOS-FP, RQAMS, and AIRPACT, and more pronounced peaks for the other models. Overall, NCAR WRF-Chem peaked the earliest (14:00 PDT) and ARQI peaked the latest (17:00 PDT). However, most model patterns deviate from the FRP observations. The day-to-day variation and bimodal patterns on some of the days were not captured by any of the models. UCLA WRF-Chem incorporated an inversion technique to constrain fire emissions, which allowed the emissions to be pushed later resulting in a better agreement with the FRP data. But the coarse time resolution of the scaling factors (8-hours) greatly limited how much the diurnal profile could be modified. Additionally, the nighttime fire activity was not well described on the nights of 2-3 and 7-8 August by most models, except for ARQI due to its later peaks. Note that FireWork used similar diurnal factors to ARQI, but it is not shown in Fig. 4.

Multiple ways to improve the representation of diurnal emission variations can be drawn from these results. First, forecasts would likely benefit from including diurnal cycles based on geostationary FRP, coinciding with recent literature (Wiggins et al., 2020). For doing this, at least one day of "spin-up" would have to be performed or using near-real-time incorporation of data. A modeling system for this goal has been reported, which adopts a strategy of hourly-sequential warm-start runs with FLEXPART-WRF (Solomos et al., 2015, 2019), which allows the emissions to be updated every hour using METEOSAT geostationary observations. However, due to large day-to-day variability in diurnal cycles, it still does not guarantee that persisting the latest diurnal pattern into the forecasting window will provide better results. One possibility is to utilize fire weather forecast, which are currently used to predict fire danger. Thus, future work is needed to investigate methods for forecasting the diurnal behavior of fires.

**3.2 Evaluation of smoke AOD forecasts against satellite data**

**3.2.1 Data and statistical metrics**

The AOD at 550 nm from the MCD19A2 Version 6 product (Lyapustin and Wang, 2018) is used for the evaluation. It is a MODIS Terra and Aqua satellites combined Level 2 product based on the Multi-Angle Implementation of Atmospheric Correction (MAIAC) algorithm producing AOD data at 1-km pixel resolution (https://lpdaac.usgs.gov/products/mcd19a2v006/). Compared to other algorithms, the MAIAC algorithm provides more available AOD data over smoke plumes with its capability to accurately classify thick smoke, which is frequently identified as clouds by other methods (Lyapustin et al., 2018). With Terra and Aqua's sun-synchronous low earth orbit, the MAIAC data has a higher nominal resolution than geostationary data but at lower temporal refresh rates. The equatorial crossing time for the MODIS Terra is 10:30 and 22:30 LST, and 01:30 and 13:30 LST for the MODIS Aqua. Locations in low- and mid-latitudes are scanned twice per day by each of the satellites. Higher latitudes can receive more frequent data coverage with up to six orbits per day in Alaska and northern Canada. As the MAIAC AOD is retrieved from visible band (470 nm) measurements, only daytime data are available. The AOD accuracy is evaluated as ± (0.05+15 %) or even better ± (0.05+10 %) in a global validation (Lyapustin et al., 2018). A recent assessment over North America against ground-based observations at the AErosol RObotic NETwork (AERONET) sites indicates that MAIAC performs well for this region over a wide range of surface conditions, with a bias of 0.015 and an RMSE of 0.062 over western North America (Jethva et al., 2019). MAIAC also shows extended coverage over the continent U.S. compared to VIIRS product or other MODIS algorithms, owing to its pixel selection process and ability to retrieve aerosol information over brighter surfaces (Jethva et al., 2019; Superczynski et al., 2017). For post-processing, the data were filtered according to the quality assessment flags, keeping the retrievals with cloud masks indicating "clear" or "possibly cloudy" and adjacency flags of "clear" or "adjacent to a single cloudy pixel". The tiles of retrievals were concatenated to produce hourly snapshots with the overpassing time rounded to full hours. The filtered AOD data were spatially mapped onto the grid corresponding to each model's resolution for consistency. There were 14 hourly scenes in total on 4-8 August 2019 (2-3 snapshots per day). The evaluation was also performed at four specific grid resolutions (0.1˚, 0.2˚, 0.5˚, and 1.0˚) to examine the model performance at different spatial re-gridding resolutions.

To evaluate the AOD retrievals during the Williams Flats fire over our region of analysis, we compared the MAIAC data against the AERONET sun photometers data (Version 3, Level 2.0, cloud-cleared and quality-assured) (Giles et al., 2019). During FIREX-AQ, multiple temporary NASA AERONET platforms - Distributed Regional Aerosol Gridded Observation Networks (DRAGON) - were deployed to collect sun photometer measurements (Holben et al., 1998, 2018). Fixed DRAGON sites operated in Missoula, Taylor Ranch, and McCall (https://aeronet.gsfc.nasa.gov/new_web/DRAGON-FIREX-AQ_2019.html). Along with the permanent AERONET sites, the AOD retrievals are available at 27 ground sites (Fig. S2). As MODIS MAIAC AOD data at 550 nm is used in model evaluation, for consistency, the AERONET AOD at 500 nm is

converted to 550 nm using the Ångström exponent retrieved for 440 - 675 nm. Following the collocation strategy reported in Jethva et al. (2019), we used two sets of spatiotemporal averaging windows to get the AOD matchups. The MAIAC data is re-gridded onto a 0.1°- (0.4°-) resolution grid, and then bilinearly interpolated onto the locations of the sites; the AERONET data is averaged within 0.5-hour (1-hour) time windows centered at the overpass time of MAIAC. To avoid values after re-gridding driven by very sparse MAIAC pixels contained in the respective grid boxes, the minimum number of 1-km satellite observations contained in each grid cell is required to be larger than 20 % of the maximum possible 1-km pixels contained in a grid box. Figures 5a and 5c shows the scatterplots constructed by using the matchups between AERONET and MAIAC. The MAIAC AOD is highly correlated ($r \sim 0.84$ and $0.89$) with AERONET and shows small positive biases. As expected, the larger spatial and temporal averaging intervals yield a larger number of data pairs. The dependence of MAIAC bias on the magnitude of AOD is examined, and the result for the bins with the number of matchups larger than five is shown (Figs. 5b, 5d). The median error is less than 0.015, and an increasing trend towards higher aerosol loading is notable for AOD bins with their center values larger than 0.1. The spread of the errors becomes greater as AOD increases. This result demonstrates acceptable accuracy of the MAIAC AOD during this wildfire event. It also suggests a tendency of slightly larger positive bias and increased variability in the retrieval errors over the areas with significant smoke impacts.

Regarding the model forecasts, coincident predictions at the closest hour relative to observations were derived from the most recent forecast cycle (initialized within 24 hours) excluding the spin-up period. For consistency with AOD measurements, the forecasts are also filtered to exclude cloudy conditions based on the cloud water mixing ratio or cloud fraction, depending on data availability. Specifically, for HRRR-Smoke, UCLA WRF-Chem, AIRPACT, ARQI, and NCAR WRF-Chem, the grid cells with total column cloud water $>10^{-6}$ kg m$^{-2}$ were filtered out. For GEOS-FP, the grid cells with low cloud fraction or middle cloud fraction $>10\%$ were masked. For UIOWA WRF-Chem, grid columns with more than five vertical layers with clouds were excluded. Although no cloud filter was implemented for the other models as cloud variables were not archived, the grid cells with clouds can be mostly masked when filtering together with the observations. After the cloud screening, the temporally and spatially collocated prediction and observation data were kept for the comparisons.

Three sets of evaluations are presented here, focusing on standard statistical measures, the magnitude of smoke AOD enhancements (sAOD), and spatial coverage of smoke plumes. The sAOD was derived by subtracting a background AOD, represented by the average of the lowest 20% values within the entire region of comparison, for each modeled and observed distribution map. The statistical metrics used in the evaluation include correlation coefficient (r), ratio, mean bias (MB), normalized mean bias (NMB), root-mean-square error (RMSE), and normalized mean error (NME), which are calculated as follows:

$$MB = \frac{1}{n}\sum_i(m_i - o_i)$$
$$(1)$$

$$RMSE = \sqrt{\frac{1}{n}\sum_i (m_i - o_i)^2}$$

(2)

$$NMB = \frac{\sum_i (m_i - o_i)}{\sum_i o_i} \times 100\%$$

(3)

$$NME = \frac{\sum_i |m_i - o_i|}{\sum_i o_i} \times 100\%$$

(4)

$$r = \frac{\sum_i [(m_i - \bar{m}) \times (o_i - \bar{o})]}{\sqrt{\sum_i (m_i - \bar{m})^2 \times \sum_i (o_i - \bar{o})^2}}$$

(5)

$$ratio = \frac{1}{n}\sum_i \frac{m_i}{o_i}$$

(6)

Here the subscript $i$ represents the pairing of $N$ observations ($o$) and model predictions ($m$) by spatial location and time.
Overbars indicate averages over location and/or time. These metrics were also used for comparison of surface PM$_{2.5}$ forecasts
against ground-based observations in section 3.5.

In addition to standard statistical measures, the spatial extent of the smoke plume is evaluated based on how well the
predicted location of the plume compares with the actual smoke plume detected by MODIS AOD. For each observation scene,
observed and modeled plume coverages are compared by producing the Figure of Merit in Space (FMS) and False Alarm Rate
(FAR) (Boybeyi et al., 2001; Mosca et al., 1998; Rolph et al., 2009) defined as:

$$FMS = \frac{a}{a+b+c} \times 100\%,$$

(7)

$$FAR = \frac{b}{a+b} \times 100\%.$$

(8)

These two categorical scores are calculated by counting the grid cells for model predictions and observations with sAOD >
0.05 that fall into the four categories listed in Table 2. FMS/100 is equivalent to the threat score or critical success index (CSI)
in verifying meteorological forecasts; it is defined as the ratio of the intersection to the union of the plume areas. FMS ranges
from 0% to 100%, with a high value indicating a good model performance. It should be noted that, since missing AOD
retrievals exist due to cloud contaminations, the filtered data do not always indicate the exact coverage and outline of smoke

plumes. Therefore, a small value of FMS does not necessarily suggest poor model performance. Although an imperfect metric, the FMS is useful for revealing model performance on a per-snapshot basis (Rolph et al., 2009).

**3.2.2 Model performance statistics of smoke AOD (sAOD)**

In this section, the evaluation results of sAOD forecasts are presented. The time period for evaluation was 4–8 August 2019, since there were multiple models that had not included emissions from the Williams Flats fire on 3 August, and on 9

August showers and cloudy weather resulted in very few AOD observations. It should be noted that because of the different setups for the chemical LBCs as summarized in section 2.2, there may be systematic discrepancies in their AOD predictions. Additionally, HRRR-Smoke does not consider non-smoke sources; the models using simplified chemistry can struggle to represent background aerosols arising from secondary formation that is not resolved within the mechanism. Figure 6 shows the comparison of the background AOD estimated from MAIAC data and model forecasts per hourly scene. While the observed

background ranges between 0.06–0.14, the modeled counterparts show less variability except for RAQMS. Most models have smaller background AOD than the observations, and systematic discrepancies can be seen among the models. Thus, these discrepancies are excluded in the following comparison by subtracting the background values from the total AOD.

Figure 7 shows the map of MAIAC sAOD from Terra MODIS at about 20:00 UTC 5 August 2019, along with the forecasts by the twelve models. Comparison of sAOD distributions for the other times is provided in the supplement (Fig. S3-

S15). As seen in Fig. 7, the observed areas with sAOD>0.05 can largely represent the smoke plume, and this threshold is used to evaluate the spatial extents of smoke plumes in the following section (3.2.3). The smoke plume for this day can be separated into three categories: 1) the fresh intense plume nearby the fire blowing east with the peak sAOD reaching above 1.0; 2) an older plume in the vicinity of the fire (over Washington State, Oregon and Idaho) from emissions earlier in the day or the previous day that's likely within the boundary layer; and 3) smoke transported further away from the fire (i.e. the band of high

sAOD extending over northern Montana and southern Canada) associated with emissions from the previous days that was injected into the free-troposphere. Note that the scattered enhancements over the southeast of the region in Fig. 7 are due to a small fire located in Idaho and also the scattered low clouds, as elevated satellite AOD retrievals have been seen around the rim of clouds (Ignatov et al., 2004; Kondragunta et al., 2008) owing to a high relative humidity environment near clouds and thus the hygroscopic growth of some particles. Overall, the high-resolution regional models tend to be more effective in

depicting the fine structure of the plume transport, but also show a higher risk of displacing the narrow plumes. All models represented the fresh plume but with a significant variability in the spatial coverage and magnitude, with the FRP-driven emissions resulting in higher sAOD than hotspots-driven emissions, in consistency with their relative emissions magnitudes shown in section 3.1. Most models also show a representation of the nearby aged plume, but again the magnitude is highly variable, and the locations of the smoke differ substantially, likely related to the diurnal emission profiles, model resolution,

as well as the driving meteorology. The misrepresented spatial pattern could also be due to the observed diurnal pattern on 4 August having a double-peak structure, differing from any of the diurnal patterns assumed by the forecasts (Fig. 4). Conversely,

the band of enhanced sAOD related to plume injection on the previous day seems to be only shown by a few models (HRRR-Smoke, UCLA WRF-Chem, and FireWork, Fig. 7), but there is still a large variability in the magnitudes. The representation of plume injection is further evaluated in section 3.4. While this analysis is for a single overpass, the overall model performance follows similar pattern and the result can be generalized for most days (Figs. S3-S15).

The quantitative evaluation of modeled sAOD during 4–8 August are summarized in Table 3, with the corresponding scatter plots shown in Fig. S16. In order to examine model performance in the vicinity of the fire and over transported smoke, the same statistical metrics are calculated over the fresh-plume impacted (Fp) areas and other (Ot) areas separately, and the fresh-plume area boundaries are defined by examining satellite visible images (see Fig. 11). The total number of points included within the comparison was from 610 to 622623 for the entire analysis region, depending on the grid resolutions. Overall, although some of the models show nearly unbiased predictions (UIOWA WRF-Chem and CAMS), negative biases in sAOD are seen for all models, with the MB ranging between -0.070 and -0.004, and NMB between -4.3% and -87.4%. The underpredictions are also seen over the Ot and Fp areas, except for CAMS and UIOWA WRF-Chem over the Fp areas, owing to likely the overpredicted emissions prior to times of the satellite overpasses. The absolute deviations of modeled sAOD against observations are large, with the NME of 61.4% to 90.2%, the RMSE of 0.11 and 0.17, and the correlation coefficients ($r$) <= 0.50. These results suggest that the spatial distribution patterns of sAOD were not well represented, which has been indicated by the discrepancies in the plume locations and spatial patterns shown in the map comparisons.

Although the characteristics of the models differ in a variety of dimensions, it is noteworthy that the models incorporating assimilation of satellite observations, including GEOS-FP, CAMS, RAQMS, UCLA WRF-Chem, and WISC WRF-Chem, are within the six models showing less underpredictions of sAOD (NMB ≥ -51%). Meanwhile, the five models ingesting FRP-based fire emissions, which are UIOWA WRF-Chem, GEOS-FP, CAMS, UCLA WRF-Chem, and HRRR-Smoke, rank within the seven models with comparably less bias (NMB ≥ -52.9%). While improvements in 1-day AOD forecasts when assimilating AOD are expected (e.g. Kumar et al., 2019; Saide et al., 2013), FRP-based emission inventories generally use AOD to tune the conversion of FRP to emissions (Ichoku and Ellison, 2014; Kaiser et al., 2012; Koster et al., 2015), which can explain these results. Future work could explore this topic by performing sensitivity simulations to determine the major factors of the AOD forecast errors.

Besides the characteristics of model settings, the horizontal resolution used for re-gridding of model and observation data may also influence the performance statistics. In order to isolate the impact of grid resolution, the data was mapped onto four grid resolutions (0.1˚, 0.2˚, 0.5˚, and 1.0˚) and examined the models' performance accordingly. As shown in Fig. 8, although the spatial resolution changes, the sAOD statistics for the twelve models remain within ranges of about 0.05 – 0.55 for correlation coefficient ($r$) (with $r^2<0.3$), -90% – 20% for NMB, and 60% – 95% for NME. The ranges are slightly larger compared to the statistics shown for the original horizontal resolutions. However, the forecast performance is still poor in terms of $r^2$ (<0.3) and NME (>60%), even when comparing at a resolution of 1.0˚. Regarding variations in the statistics against

the re-gridding resolution, the NMB does not have distinctive changes, which is expected because the spatial smoothing could not yield much improvement in the mean bias against observations. For the NME, most models present decreasing trends when the re-gridding resolution gets coarser, except for NCAR WRF-Chem and UIOWA WRF-Chem which show slight increases or mixed trends. In contrast to this, the variations of $r$ values are more complex. With the re-gridding resolution getting coarser, we may expect an increased $r$ due to that some extreme outliers in sAOD distributions may get smoothed out, but only four among the twelve models, namely ARQI, AIRPACT, UCLA WRF-Chem, and NAQFC, show the increasing trends. For the other models, mixed trends in $r$ are shown when the re-gridding resolution becomes coarser. Overall, the relative ranking of the models' statistical performance does not vary significantly, and the horizontal grid resolution does not seem to be a decisive factor for models' performance. Thus, in the following section, the evaluations were performed based on model data at their original horizontal resolutions (i.e., without horizontal re-gridding).

### 3.2.3 Smoke AOD magnitude, temporal evolution, and spatial matching of plumes

In addition to the point-to-point comparisons, in this section the predictions are evaluated in perspectives of the temporal evolution of sAOD magnitude and spatial extent of the smoke plumes. An sAOD threshold has (sAOD> 0.05) been applied to filter sAOD representing the areas with pronounced smoke impact. This threshold is qualitative and chosen by visually examining the observation maps of sAOD and satellite visible images. Meanwhile, to exclude the grid cells significantly impacted by other smaller fires, only the data within a smaller area indicated by the red dashed box in Fig. 7 (see the MODIS sAOD map) were filtered for the analysis in this section.

The temporal evolution of the sAOD magnitude is shown in Fig. 9. In consistency with the statistical results, the overestimations of sAOD magnitudes occurred for some overpasses for the fresh-plume areas, particularly for the models driven by FRP-based emissions. Temporal variability in the model performance is noticeable, which is closely associated with the limitation of forecasted fire emissions based on the assumption of persistence. For example, on 4 August, nearly all the models show underestimations in sAOD over the fresh-plume area mostly resulting from the underpredicted emissions, due to delayed ingestion of emission information from satellite observations. An additional reason is that 4 August was the first day when the Williams Flats fire became active in most models, except for HRRR-Smoke and FireWork that already included it on 3 August. In comparison, on 5 and 6 August the burned area increased steadily without dramatical elevation (see the day-to-day increment of burned area in Fig. 2). Accordingly, the models show some skills, since the assumption of persistence managed to produce comparable emissions against the actual fire activity. However, the stronger burning activity was observed on the 7th (Fig. 2), leading to underestimations of the emissions. As the last overpass time of the Aqua-MODIS was about 14:00 PST, well before the peaking hour of FRP at about 17:00 PST on 7 August (Fig. 4), the impact of underestimated fire emissions was not shown by the modeled sAOD over the fresh-plume areas. However, this change in fire behavior generated a large underprediction on 8 August over other areas (Fig. 9). As indicated by the observed sAOD distribution at 19:00 UTC on 8

August (Fig. S14), the elevated sAOD was mostly contributed by the transported smoke aerosols resulting from the enhanced fire emissions on late afternoon 7 August. These results show that the assumption of persistence of smoke emissions degraded the forecasts. Future work needs to be performed to find strategies to predict changes in the smoke emissions over the forecasting window. Additionally, the representation of plume injection plays a critical role in the forecasted sAOD for the transported smoke plumes on 7 August, which is discussed further in section 3.4.

Consistency of the modeled and forecasted spatial coverage of smoke plume is also examined. As shown in Fig. 9, the models underpredicted the total number of grid cells with sAOD > 0.05 for most of the snapshots, suggesting underestimation in the area of the smoke plumes. The accuracy of the predicted smoke areas is evaluated by the metrics of FAR and FMS for each MODIS snapshot during 4-8 August. These two metrics are derived at the original grid resolutions of different models (Fig. 10) and at fixed grid resolutions (0.1˚, 0.2˚, 0.5˚, and 1.0˚) as well (Fig. S17), and the results show similar features. The maximum FMS can reach as high as 80% with the medians for the models ranging from 10% to 70%. The FAR scores are generally low, and the median values are below 45%. There is a noticeable group of models showing relatively better performance with lower FAR and higher FMS score, which include CAMS, RAQMS, GEOS-FP, UCLA WRF-Chem, and UIOWA WRF-Chem. As analyzed previously, these models also show better performance for the statistics of sAOD and the total number of grid cells with sAOD>0.05. These models used FRP-based emissions (except for RAQMS) and incorporated assimilations of satellite AOD observations (except for UIOWA WRF-Chem). Other factors such as complexity of chemical mechanisms, chemical LBCs, horizontal resolution, initial time of forecast, and dynamic core used to drive the meteorological dispersion and transport, do not seem to be determining for these metrics.

### 3.3 Evaluation of surface PM$_{2.5}$ forecasts

### 3.3.1 Data and statistical metrics

The model forecasts of surface PM$_{2.5}$ mass concentrations during 4-9 August 2019 are evaluated against the hourly measurements collected from the AirNow (https://www.airnow.gov/) network. The observations were accessed from the OpenAQ Platform (https://openaq.org) and were originally collected by state, local or tribal monitoring agencies using federal reference or equivalent monitoring methods approved by the U.S. Environmental Protection Agency (EPA). As noted by AirNow, although the preliminary data quality assessments are performed, the data were not fully verified and validated through the quality assurance procedures that the monitoring organizations used to officially submit and certify data on the U.S. EPA Air Quality System (AQS). Compared to the AQS data that are used for regulatory purposes, such as determining attainment of the National Ambient Air Quality Standards (NAAQS), the observations from AirNow are used to report the Air Quality Index (AQI) to the public, and they have a better completeness during extraordinary air pollution events such as wildfires. Additionally, the AirNow data has also been compared with the U.S. EPA's Air Data (https://www.epa.gov/outdoor-

air-quality-data) to check the consistency. The missing data in AirNow were filled in by combining these two datasets. The locations of the 86 monitoring stations within the domain of analysis are shown in Fig. 11. By examining the visible images based on the GOES-17 data and MODIS MAIAC AOD maps, 14 sites are selected as "fresh-plume stations" (Fp) that show immediate impact by the fresh smoke from the Williams Flats Fire on 4-7 August, i.e. the stations located within the fresh-plume borders on any of the days, as denoted by the red dots in Fig. 11.

The $PM_{2.5}$ observations are hourly averages reported at the end of each hour, namely centered on the half hour. For the modeled counterparts, surface $PM_{2.5}$ values are derived by bilinearly interpolating the modeled 2-D forecasts at the lowest model level onto the latitude and longitude coordinates of each monitoring station. The inconsistency in spatiotemporal representations of the model forecasts and observations could be a source of model-observation differences. It should also be noted that most model data are provided as hourly files, while some of them come as three-hourly snapshots (see Table 1). Thus, the limitation in output frequency could contribute to errors compared to $PM_{2.5}$ observations for some models. .

The comparison is restricted to the 83 sites that have forecasts from all the twelve models, thus three stations (No.1, 18, and 29 in Fig. 11) are omitted since they are not covered by all models. The common statistical measures as used for sAOD (described in section 3.2.1) are used for the hourly $PM_{2.5}$ forecasts at all the 83 stations, as well as separately over the stations categorized as "fresh-plume sites" (Fp) and all the other sites (Ot) (Fig. 11). Similar to the comparisons for sAOD, the surface $PM_{2.5}$ mass concentration enhancements ($sPM_{2.5}$) are derived, with the background represented by the average of the lowest 20% values for the model forecasts over the entire region of analysis, and 10% over the observation stations. Note that a lower percentage is used for the observations due to the sparseness of stations. Three sets of comparisons are presented, which focus on (1) the overall statistical measures and their spatial patterns, (2) diurnal forecast performance, and (3) day-to-day performance.

### 3.3.2 Results

Table 4 gives the statistical comparison results of the hourly surface $sPM_{2.5}$ from the twelve models and measurements over all 83 monitors. Also included are the statistical measures for the 14 stations in the category of "fresh-plume" (referred to as Fp) and the other 69 stations (referred to as Ot), respectively. The scatter plots of all pairs of prediction and observation data and for the Fp and Ot stations separately are shown in Figs. S18-S20. Maps in Figs. S21-S23 show statistical results by station. Overall, FireWork and NAQFC generally rank within the best four for all the statistical metrics used here. It is interesting to note that these models are the two operational forecasts for predicting air quality in Canada and the U.S., where performance against surface monitoring stations is generally the primary metric for model evaluation. As revealed by the results for all stations, half of the models give positively biased $sPM_{2.5}$ predictions. Specifically, there are three models, i.e. FireWork, NAQFC, and RAQMS, showing nearly unbiased predictions with NMB of -4.00%, 0.30%, and -5.50%, and the ratios of predictions versus observations are between 0.95 and 1.00. However, in terms of the correlation coefficients and absolute errors, all the models show low performance with $r < 0.35$ ($r^2 < 0.13$), RMSE > 9.8 µg m$^{-3}$, and NME > 70%, which

is similar to the results for sAOD. It is worth mentioning that there is a discrepancy when comparing between the ranking of the best models for $sPM_{2.5}$ and sAOD. However, as the spatial and temporal representations of the observed surface $PM_{2.5}$ and AOD incorporated into the above statistics are very different, the statistics are not exactly comparable. Thus, the discrepancies in statistics for AOD and surface $PM_{2.5}$ are further discussed in section 3.5 by using coincident data. The statistical measures indicate that the $sPM_{2.5}$ errors can barely be recovered through simple bias corrections, suggesting the mismatch in spatial distributions of surface smoke aerosols.

In terms of the comparison over different groups of stations, the correlation coefficients for the Fp stations are larger compared to the Ot stations for most models, except for UCLA WRF-Chem and UIOWA WRF-Chem, although the values are all less than 0.3 ($r^2 < 0.09$, Table 4). By contrast, for the other five statistical measures, most models tend to show reduced performance for the Fp stations compared to the Ot stations, especially for RMSE which can also be seen in the RMSE distributions (Fig. S22). It is likely because the model performance over the Fp stations is more closely impacted by errors in fire emissions, and RMSE is more driven by large absolute values of the model-observation differences compared to the other statistical metrics. These results suggest that the spatial structures of $sPM_{2.5}$ from the fresh smoke plumes appears to be slightly better captured by forecasts compared to the outer areas; however, considerable biases still contribute to the larger RMSE in fresh smoke than the outer areas. It is also noteworthy that, there are five forecasts (HRRR-Smoke, UCLA WRF-Chem, UIOWA WRF-Chem, GEOS-FP, and CAMS) showing large overpredictions for the Fp stations with NME > 40% and MB > 6 ug m$^{-3}$, which are all driven by FRP-based fire emissions and have shown relatively fewer negative biases of sAOD. As discussed later in this section, the overpredictions in $sPM_{2.5}$ by these models mostly happened in the evening through early morning. Among these five models, CAMS shows even larger NMB for the Ot sites compared to the Fp sites, likely due to model processes besides fire emissions. Another notable feature is that some models give different signs of biases over the Fp and Ot sites. For example, HRRR-Smoke and FireWork show positive biases for Fp sites and negative biases for Ot sites, and the opposite is seen for WISC WRF-Chem, NAQFC, and RAQMS. For models showing the same sign, the magnitudes of NMBs between Fp and Ot sites can be quite different. This points to discrepancies in the model performance for the fresh versus aged smoke, which could be generated due to multiple reasons including issues in plume injection, downwind chemical evolution, and transport of fire smoke.

As the fire activity, plume injection and vertical mixing changes significantly from daytime to nighttime, it is useful to compare the diurnal feature of forecast performance. The predicted and observed $sPM_{2.5}$ are compared diurnally (from 00:00 to 23:00 PDT) for the two categories of stations (Fp and Ot) respectively. Figures 12a and 12b show observed $sPM_{2.5}$ statistics. The mean $sPM_{2.5}$ over the Fp sites is about three times larger than the values for the Ot sites, as they are impacted by smoke more directly. Also, there is a more prominent diurnal variability in $sPM_{2.5}$ for the Fp sites with overall higher values in nighttime than in daytime. The peak to valley difference of the means is about 8.5 μg m$^{-3}$ (76% of the minimum) for the Fp sites and about 2.0 μg m$^{-3}$ (35% of the minimum) for the Ot sites. For the Fp sites, the means have an early afternoon peak (13:00 PDT), which seems to be observed at a small portion of the sites depending on the episodical fire emissions and wind

directions. The increase that occurred around 18:00 to 20:00 PDT can be attributed to the fresh smoke emitted during the peak hours of the fire activity reaching the sites. Also, the boundary layer collapse along with transition of convective boundary layer into stable boundary layer during the early evening and the continued burning on some of the days during these hours can also play a role. By contrast, a decreasing trend is seen from 00:00 to 10:00 PDT, which is mostly related to the reduced emissions later in the night, the spatial dispersion of the smoke, and the PBL growth that leads to vertical dilution in the morning. For the Ot sites, there is a peak in the early morning (08:00 PDT) possibly due to anthropogenic activity. While the upper decile and quartile follow a similar trend than the mean for the Fp sites, the lower decile, lower quartile and median show relatively flat diurnal profiles, likely due the fresh plume not impacting all sites simultaneously. For Ot sites the behavior of all statistics is similar. We thus focus on comparing the mean of the models.

Comparisons of the modeled diurnal means of sPM$_{2.5}$ against the observations are given in Figs. 12c and 12d. The corresponding diurnal variations of the model statistical measures are included in Figs. S25 and S26. For models, the forecasted diurnal variability of the means is mostly stronger than the observations, and the peak to valley differences range from 4.6 - 45.3 µg m$^{-3}$ (36% - 474% of the minimum) over the Fp sites and 1.0 - 6.4 µg m$^{-3}$ (37% - 113%) over the Ot sites. For the Fp sites, the early afternoon peak is captured by FireWork and HRRR-Smoke, although there are differences in timing, width, and magnitude. Also, most models capture the early morning decrease related to development of daytime PBL and dilution, and RAQMS, UIOWA WRF-Chem, and CAMS show the decrease 3-5 hours later than observed. Another important feature is the timing and magnitude of evening buildup that are quite different among the models. UCLA WRF-Chem, UIOWA WRF-Chem, and GEOS-FP show the buildup 1-2 hours earlier than observed with much higher concentrations; while FireWork, WISC WRF-Chem, NAQFC, and RAQMS show it 2-3 hours later, and the nighttime concentrations are overall lower than the observations. CAMS captured the daytime trends and late-afternoon buildup well; however, it shows overpredictions during nighttime to early morning. The overpredictions during the late afternoon to early morning by UCLA WRF-Chem, UIOWA WRF-Chem, GEOS-FP, HRRR-Smoke, and CAMS seem to dominate the large positive biases and NMB for the Fp sites. This helps to explain the discrepancy between the performance for sAOD and sPM$_{2.5}$. Overall, these differences in the evening and nighttime evolutions can be attributed to the model uncertainties in diurnal pattern of emissions, the larger proportion of emissions allocated within the PBL, and underestimated nighttime PBL heights as well. The timing of collapse of PBL may also play a significant role, as the models produce emissions at surface rather than injected above after that, which could subsequently lead to more enhanced surface PM$_{2.5}$ when the fire emissions continue.

The diurnal variation of the average sPM$_{2.5}$ for the Ot sites also show patterns deviating from the observations. Most models (except for UCLA WRF-Chem, NCAR WRF-Chem, and HRRR-Smoke) show a common feature of a valley in the afternoon, however it is not seen for the observations. It could be a joint consequence of the larger dispersion volume owing to higher PBL depth than reality, and/or issues with other sources, e.g. anthropogenic activities. The early-morning peak is captured by NAQFC, AIRPACT, and ARQI, earlier by about one hour than the observations (except for ARQI). Similar to the

result over the Fp sites, stronger overpredictions are seen during the evening to early morning for some models (CAMS, GEOS-FP, WISC WRF-Chem, UIOWA WRF-Chem, UCLA WRF-Chem), , which is likely due to similar reasons.

Considering the coincidence of sAOD and sPM$_{2.5}$, the Terra- and Aqua-MODIS overpassing times are between 11:00 to 15:00 PDT, when the models with large nighttime sPM$_{2.5}$ overpredictions (except for HRRR-Smoke) tend to agree the best with the observed diurnal patterns, but still overpredicting (Figs. 12c and 12d). As mentioned previously, these models use FRP-based fire emissions and show less underpredictions for sAOD compared to the other models. This suggests discrepancies in sAOD and sPM$_{2.5}$ prediction performance, which will be further discussed in the following sections.

The model representation of the day-to-day evolution of sPM$_{2.5}$ magnitudes is also evaluated. Figure 13 shows the comparison of the spread of modeled daily average sPM$_{2.5}$ for the two categories of stations. The NMB and NME results generally reflect similar information and are presented in Figs. S27 and S28. For the Fp sites, the observed means show an overall increasing trend with a peak on the 7$^{th}$ and a slight decline afterwards. Most models generally captured the day-to-day variations of the sPM$_{2.5}$, with several models showing underpredictions on the 4$^{th}$ or 5$^{th}$ and occasionally overpredicting on the following days (6-9 August), including ARQI, HRRR-Smoke, AIRPACT, UCLA WRF-Chem, UIOWA WRF-Chem, FireWork, and NAQFC. This could be attributed to delayed ingestion of fire emission information based on satellite observations collected prior to a forecast cycle, which has also been shown with the variations of the spread of magnitudes for the modeled sAOD. By contrast, the overestimations on 4 and 5 August are seen for WISC WRF-Chem, GEOS-FP, and CAMS, which could be linked to the larger emissions injected at levels close to the land surface and within the PBL than reality. In addition, significant overestimations of the 75$^{th}$ and 90$^{th}$ percentiles are shown by the models driven by FRP-based fire emissions, especially on 9 August for the three models using QFED data (i.e., UCLA WRF-Chem, UIOWA WRF-Chem, and GEOS-FP). These overpredictions could be related to overpredicted total emissions and/or issues in the vertical allocation of emissions (i.e., putting too much amount of smoke within PBL) as will be discussed in section 3.5.

For the Ot sites, the observed daily averages of sPM$_{2.5}$ show smaller day-to-day variations compared to the Fp sites (Fig. 13). In contrast to the variations of the observed means over the Fp sites that peaks on 7 August, the result over the Ot sites peaks on 5 August and reduces slightly afterwards. Besides meteorology, this decreasing trend after the 5$^{th}$ is likely attributed to the evolution of fire activity, which showed higher plume injection heights on 7 and 8 August, leading to lofted smoke above the PBL and generating less impact near the land surface, as will be discussed in sections 3.4 and 3.5. The observed decreasing trend is overall depicted by some of the models, such as HRRR-Smoke, AIRPACT, NAQFC, GEOS-FP, CAMS, and RAQMS, although with temporal shifts and biases in their magnitudes. In comparison, the nearly flat or opposite trends of the daily means is seen for UCLA WRF-Chem, UIOWA WRF-Chem, ARQI, and FireWork, which is likely related to allocating too much emissions close to the land surface compared to observations, which will be shown by the comparison against lidar data for the aged plumes in the section 3.4.3.

### 3.4 Evaluation of vertical structure of smoke plumes using NASA DC-8 aircraft data

### 3.4.1 Observations and derivation of smoke plume heights and PBL heights

The vertical allocation of fire emissions can significantly affect both the total column smoke aerosol loading and surface aerosol concentrations in adjacent areas of the fire and remotely downwind. In this section, we evaluate the vertical plume structures and fire plume injections predicted by the models, based on the observations acquired by the DIAL-HSRL (Hair et al., 2018) from the DC-8 aircraft that sampled the Williams Flats fire plume on four days, namely 3, 6, 7, and 8 August 2019, during the FIREX-AQ field campaign. The DIAL-HSRL system is capable of providing measurements of aerosol depolarization (355 nm, 532 nm, 1064 nm), aerosol/cloud extinction (532 nm), and backscatter coefficient (355 nm, 532 nm, 1064 nm) above and below the flight height at a temporal resolution of 10 s. In addition, the partial-column AOD were derived by integrating the aerosol extinction profile at 532 nm in nadir, when the flight height was above 5.15 km. In this section, the plume structures are compared using the DIAL-HSRL backscatter coefficient (at 532 nm) profiles, as they provided more detailed structures and less missing data than extinction, and forecasts of $PM_{2.5}$ concentration profiles (as most models did not provide extinction or backscattering profiles). Besides, the AOD derived from the lidar data is also used to analyze the relation between vertical aerosol distribution and the ratio of surface $PM_{2.5}$ concentration versus total column aerosol loading in section 3.5.

All the models evaluated in the preceding sections, except for NAQFC, are examined in this section. The maps of flight tracks are shown in Fig. 14. As summarized in Table 5, the observations of 15 flight transects that sampled straight along the smoke plume are selected, among which 11 transects represent fresh plumes close to the fire on those four days, and 4 transects represent the aged plumes sampled on 7 and 8 August. The modeled counterparts of smoke plume structures along these transects are numerically derived using the 3-D forecasts of $PM_{2.5}$ concentration fields valid at the nearest hour of model output relative to the sampling time of a profile. Then the model data are bilinearly interpolated onto the latitude and longitude coordinates of the sample profile at each of the model levels.

Smoke plume heights and PBL heights are determined from the DIAL-HSRL data and model forecasts to evaluate models' performance for plume injection. The method used to determine plume and PBL heights is based on the vertical gradients of aerosol backscatter or $PM_{2.5}$ concentrations. Aerosol profile measurements made by lidar have long been used to determine PBL heights, or more appropriately the mixed layer (ML) heights (Hayden et al., 1997; Scarino et al., 2014; Tucker et al., 2009) in the daytime since aerosol gradients can indicate the level below which the aerosol species emitted within PBL tend to be well mixed and dispersed. For aerosol profiles through a smoke plume, strong aerosol gradients can indicate the plume top, which can be higher than the PBL heights when emissions are injected above the PBL. Following this heritage, for each of the 5-point moving average backscatter profiles below 10.5 km, the highest level where the local minimum vertical gradient is less than a threshold $k$ is derived; if this criterion is not satisfied at any level, the level of the global minimum vertical gradient over the entire profile is used. Then this level is referred to as plume top-height ($h_{plume}$) if the profile is in-plume, i.e.,

when the maximum backscatter below 8 km is larger than a threshold $b$, and otherwise referred to as PBL height ($h_{PBL}$) (when the profile is out-of-plume). The value of $b$ is chosen as $2.1\times10^{-3}$ km$^{-1}$sr$^{-1}$ for 6 August, as the backgrounds increased significantly due to dispersed smoke from fire emissions on the previous days, and $b = 1.2\times10^{-3}$ km$^{-1}$sr$^{-1}$ for the other days. For out-of-plume profiles, $k = -1.2\times10^{-6}$ km$^{-2}$sr$^{-1}$; for in-plume profiles $k = -4.0\times10^{-6}$ km$^{-2}$sr$^{-1}$, which is smaller than out-of-plume condition, in order to avoid picking up heights affected by in-plume variations due to vertical mixing and plume rise. The results are also visually inspected and filtered to exclude the impact from the incoming smoke from the fires in Siberia, which can be seen on 3 August.

As for the model data, $h_{plume}$ is derived from the vertical profiles of PM$_{2.5}$ mass concentrations using a similar method. For each of the PM$_{2.5}$ profiles, $h_{plume}$ is obtained only if the profile is in-plume, i.e., when the difference between the maximum and minimum PM$_{2.5}$ below 8 km is larger than 5.0 µg m$^{-3}$. The data impacted by the Siberian fires on 3 August were also excluded, with the masked levels manually tuned for each model. Also, the data below 100 m above ground level are excluded to avoid selecting the level impacted by strong emissions near the surface. The threshold of vertical gradient $k$ is modified to $-2.5\times10^{-3}$ µg m$^{-4}$ for both in-plume and out-of-plume conditions. An additional modification is applied when the local minimum (or the global minimum) is smaller than $-3.5\times10^{-3}$ µg m$^{-4}$ that can occur at a certain level adjacent to the very intense injected fire emissions, and the vertical gradient of PM$_{2.5}$ is much stronger than near the plume top. Thus, the $h_{plume}$ is tuned upward by using the highest level at which the gradient is less than $3.0\times10^{-3}$ µg m$^{-4}$.

Depending on data availability and the agreement of the modelled PBL heights compared to the results estimated by forecasted virtual potential temperature and PM$_{2.5}$ profiles, the modeled $h_{PBL}$ is derived in different ways for three classes of models:

(1) for CAMS and RAQMS, PBL heights are not available in their outputs, so the $h_{PBL}$ is determined as the ML heights derived from the out-of-plume PM$_{2.5}$ profiles.

(2) for UCLA WRF-Chem, UIOWA WRF-Chem, and ARQI, the $h_{PBL}$ results diagnosed by PBL parameterization in model are usually lower than the ML heights estimated by the vertical PM$_{2.5}$ profiles (as will be shown in section 3.4.2). By examining the potential temperature profile and land use, it is found that the underestimation mostly happened over the area downwind to the Columbia River, where the model-diagnosed PBL heights tend to represent the top of thermal internal boundary layer relating to the underlying water body. Therefore, $h_{PBL}$ is determined from virtual potential temperature profiles ($\theta_v$) for these three models as the lowest level at which $\partial\theta_v/\partial z = 1.3$ K km$^{-1}$. The method, using a threshold of vertical $\theta_v$ gradient, is found to outperform other methods based on turbulence kinetic energy (TKE), $\theta_v$, or Richardson number for estimating convective boundary layer depth (LeMone et al., 2013).

(3) for the other models not mentioned in (1) or (2), the model-diagnosed $h_{PBL}$ that came with the forecasts are used.

It should be noted that the $h_{PBL}$ derived by the above methods are compatible with the lidar results only during daytime when the term of ML heights is applicable. In this work, lidar measurements for the selected transects were mostly collected during the daytime; however, for the data collected in the evening, e.g., as late as 18:49 LST (Local Standard Time, or UTC-8) for D3T3 (see Table 5), the lidar $h_{PBL}$ tends to represent the residual layer height, since the aerosol layer remains after the collapse of daytime boundary layer and transition into nocturnal boundary layer due to radiative cooling. Therefore, to ensure the compatibility of model and lidar data, the model $h_{PBL}$ after 16:00 PDT (15:00 LST) is derived as the higher one between the $h_{PBL}$ values at the current hour and 16:00 PDT for the same location, which allows to capture the top of the residual layer. The modeled $h_{PBL}$ and $h_{plume}$ for each of the selected transects are compared against the lidar results. Two sets of evaluations are shown, focusing on the fresh plumes and aged plumes respectively.

### 3.4.2 Evaluation for smoke plumes close to the fire

The evolution of vertical smoke plume structure and plume rise are demonstrated for the fresh plumes close to the fire, using the DIAL-HSRL observations that were collected on 3, 6, 7, and 8 August. As shown with the observed backscatter coefficients (532 nm) in Fig. 15, obvious day-to-day variability of the plume rise behavior and fire activity occurred on these days. Slight plume injection is shown on the 3rd, with a layer of enhanced backscatter located right above the PBL top at about 3 km above sea level (a.s.l. hereafter). The lofted layer of aerosols in between 4 and 7 km a.s.l. is associated with incoming smoke plume from the Siberian fires. On the 6th, the flight sampled the plume from approximately 12:00 to 15:00 PDT, which is earlier than for the other days, and no injection was observed. The fire just started to become active at the time of D6T1 with $h_{plume}$ being lower than the $h_{PBL}$. Elevated emission heights are seen afterwards for D6T2 and D6T3, while no injection above the PBL was observed and the plume tended to be well mixed within the PBL. This behavior can be partly explained that the fire emissions peaked late on that day (19:00 PDT, Fig. 4) and the measurements took place earlier. In contrast, strong plume injections above the PBL were seen on the 7th and 8th. The $h_{plume}$ reached 7 km a.s.l. on the 7th, and even stronger injections were triggered on the 8th by the thermodynamic convection related to the active burning. The flight track sampled through multiple pyrocumulonimbus (pyroCb) pulses on that day, which was generated by the convection along with the abundant heat and moisture released during the burning. Consequently, the emissions were significantly elevated with the $h_{plume}$ getting as high as 10 km a.s.l.

Based on the DIAL-HSRL data, the predicted vertical plume structures are evaluated. Figure 16 shows the comparison for the transect D7T3 sampled on the 7th, and the similar results along the other transects are provided in the supplement (Fig. S30 to S39). Overall, there is a large spread of the modeled plume heights. ARQI, HRRR-Smoke, UCLA WRF-Chem, WISC WRF-Chem, FireWork, and NCAR WRF-Chem tend to show plume injections above the PBL, while AIRPACT, CAMS, GEOS-FP, and RAQMS show smoke generally well mixed within the PBL. For UIOWA WRF-Chem, although the fire emissions seem to be allocated mostly close to the land surface, lofted plume exists over the downwind area above the PBL, which corresponds to injected smoke that occurred earlier (see Fig. S40).

The median $h_{\text{plmue}}$ for each of the eleven fresh plume transects are evaluated by statistical metrics that have already been used in previous sections for sAOD and sPM$_{2.5}$. Note that in the following evaluation all the heights are converted to above ground level (a.g.l.). The statistics are presented in Table 6, and the observed median $h_{\text{plume}}$ and model-observation difference are shown in Fig. 17a. The total number of points incorporated varies between 8 and 11, since for some models the fire had not been active yet in the forecasts on 3 August. Overall, multiple models have high linear correlation (eight models with $r$ over 0.7), indicating models following the observed trend of injections getting deeper as the days went by. However, all three global models which tend to inject their emission in the mixed layer are within this group, and thus the correlation might reflect the concurrent increasing trend in the daytime PBL heights. This means that the high correlation coefficients might not be a good indicator of smoke injection performance. Meanwhile, only four models (ARQI, HRRR-Smoke, UCLA WRF-Chem, and FireWork) show biases < 1 km with NMB < 20%, and all models have biases < 2 km with NMB < 40% and NME of 30-50%.

For the day-to-day variations, the observed $h_{\text{plume}}$ presents considerable variability associated with plume injection behavior, with the medians along each transect ranging from about 2 to 9 km a.g.l (Fig. 17a). While, most models show overpredicted $h_{\text{plume}}$ on the 3$^{\text{rd}}$ and 6$^{\text{th}}$ and underpredictions on the 7$^{\text{th}}$ and 8$^{\text{th}}$; the range of predicted $h_{\text{plume}}$ is smaller than observed for all the models (Fig. 17c), which means that the day-to-day variability in plume injection behaviors on these days were not captured by any of the models. The underestimation of temporal variation in plume injection heights is consistent with a previous study (Val Martin et al., 2012).

Overall, there is not a single model that performs the best all the time. For instance, the models that tend put emissions within the PBL (e.g. GEOS-FP, CAMS and RAQMS) performed better on the 3$^{\text{rd}}$ and 6$^{\text{th}}$, while the models with more intermediate injections performed better on the 7$^{\text{th}}$ (HRRR-Smoke and UCLA WRF-Chem). The performance of the global models may also probably be limited by the coarser vertical resolutions compared to high-resolution models, as they have limited representation of the fine-scale vertical smoke structures. While, the models with deeper injections (e.g. NCAR WRF-Chem) performed the best on 8 August for the pyroCb (Fig. 17a). This is also confirmed by comparing plume injection magnitude represented by the difference between the medians of plume heights and PBL heights (median $h_{\text{plume}}$ – median $h_{\text{PBL}}$) for each transect. As shown in Fig. 17c, for ARQI, HRRR-Smoke, UCLA WRF-Chem, FireWork, and NCAR WRF-Chem, the cases with stronger injections than observations, i.e. the data points above the 1:1 line, are mostly associated with the overpredictions of plume heights and vice versa. Some exceptions exist when the models give a higher injection magnitude but still underpredicted the plume height, which can be attributed to the underprediction of PBL height. By contrast, for GEOS-FP, CAMS, and RAQMS, the differences (median $h_{\text{plume}}$ – median $h_{\text{PBL}}$) are around zero for most cases. Thus, obvious underestimations in $h_{\text{plume}}$ are seen when strong injections are present. Another interesting result is that, while there are multiple models using the same plume rise parameterization scheme (all WRF-Chem-based configurations except HRRR-Smoke), they present large variations in their performance (e.g. D7T3, Fig. 16). Although these models used the same injection parameterization, the burned area is specified differently, which together with differences in meteorological fields and grid

resolutions can account for the large spread of the predicted plume heights. Additionally, the two systems using satellite FRP in their plume injection estimation (HRRR-Smoke and CAMS) also show very different behavior, which can be explained by the different plume rise parameterizations used in these models. Future sensitivity analysis is needed to determine the key factors contributing to their performance.

### 3.4.3 Evaluation for aged smoke plumes

Comparison of plume heights along the transects that sampled through aged smoke plumes on 7 and 8 August shows consistent features as the transects through the fresh plumes the day before. Figure 18 presents comparisons of the aged plume over northwest Montana on 7 August (D7T1). The observed smoke plume is mostly well mixed within the PBL, corresponding to the emissions injected within the PBL as observed in the fresh plume on the 6[th] (see Figs. S33-S35). The model forecasts have captured the plume heights as shown by the observations, with the smoke being within the PBL and reaching the ground surface. Similar features can be found for D7T2 (Fig. S41). While multiple models predicted injections into the free-troposphere on the 6[th] (ARQI, HRRR-Smoke, UCLA WRF-Chem, FireWork, and NCAR WRF-Chem), these lofted plumes are not represented along this flight transect, likely because they were advected faster and in a different direction than the plume in the PBL. Moreover, although the vertical location of the aged smoke is captured by the models, the spatial variability is not well represented, possibly owing to errors in the temporal profile of emissions (Fig. 4).

In contrast, significant injection into the free troposphere that happened on 7 August resulted in a large portion of the aged smoke not mixing down to the surface on the next day, as suggested by the lidar data (Fig. 19, D8T2). Although the lofted smoke is partially represented by some models showing stronger injections on the 7[th] (Fig. 16), the observed lofted smoke covered a much larger area, with the core of the observed plume (~00:25 UTC) being not captured by any model. This is likely due to the earlier diminishing injections and moderate burning activity in the late afternoon by the models, as can be confirmed by the comparisons of vertical plume structures (Figs. S36 and S37 for D7T4 and D7T5), as well as the diurnal emission evolution profiles (Fig. 4). Additionally, there is a tendency of the forecasts showing a larger proportion of smoke mixed within the PBL than indicated by the observations. This can be responsible for the discrepancies between $sPM_{2.5}$ and sAOD performance (i.e., although there is an evident underestimation of sAOD for the transported smoke plume on the 8[th], the predicted $sPM_{2.5}$ shows overestimations for some models). These results highlight again the significance of resolving both temporal and vertical representations of fire emissions in models to improve forecasts of transported smoke plumes. An important parameter of plume injection parametrizations is the percentage of emissions that are injected into the free troposphere. This parameter is generally assumed as a constant depending on the fuel category (Freitas et al., 2007), which can have a large impact on the surface smoke aerosol concentrations. Thus, more detailed evaluations of vertical partition of emissions are needed. Another possible reason for the discrepancy is the potentially enhanced evaporation of organic aerosol near the surface compared to lofted plumes (Selimovic et al., 2019), which is a process not included by any of the forecasts evaluated here.

**3.5 Synergetic evaluation of surface PM$_{2.5}$, AOD and their ratio**

The ratio between surface PM$_{2.5}$ and AOD has been widely considered in evaluations of model performance (e.g. Lennartson et al., 2018), and is also critical for studies on estimating surface PM$_{2.5}$ based on satellite AOD retrievals. As an intensive performance metric, this ratio is less dependent on mass concentrations and emission than PM$_{2.5}$ or AOD, often referred to as extensive parameters and dependent on mass concentrations. This ratio is dependent on the vertical allocation of smoke aerosols and aerosol optical properties, and thus can be used to evaluate models in terms of these aspects. This is especially important for models performing assimilations of AOD data, as misrepresentations of these ratios can lead to erroneous PM$_{2.5}$ concentrations (Saide et al., 2020).

In this section, the forecasts of surface PM$_{2.5}$/AOD ratio are evaluated for the twelve models. General examples of the ratios under different typical mixing and layering situations of smoke are demonstrated using the DIAL-HSRL data and surface PM$_{2.5}$ measurements. However, considering the sparse coincidence of DC-8 flight measurements and surface PM$_{2.5}$ data, statistical evaluation of the ratio relied on MAIAC AOD retrievals. Two sets of evaluations are presented for the model representation of the ratio regarding probability distribution and day-to-day evolution.

**3.5.1 Observations of surface PM$_{2.5}$ to AOD ratio**

Examples of the ratios observed under typical conditions of smoke aerosol profiles are shown in Fig. 20. The ratios are derived using surface PM$_{2.5}$ and AOD calculated using aerosol extinction from the DIAL-HSRL collocated within 5 km of distance. A condition commonly occurring is that of smoke aerosols well mixed within the PBL under a clean free troposphere (Fig. 20a), yielding ratios in the 70-90 range. In contrast, Figs. 20b and 20c provide typical results for near-surface smoke and lofted smoke above the PBL, with the ratios becoming much higher (402.2) or lower (38.9) compared to the general situation. A mix of these two conditions can also occur yielding ratios similar to the well mixed ones (e.g., Fig 20d). Therefore, the surface PM$_{2.5}$ to AOD ratio is applicable to suggest vertical placement of smoke aerosols. One situation that could obscure the relationship of ratio and vertical smoke layering is the clean or non-smoke cases, for which an example is presented in Fig. 20e. In this case, the enhanced backscatter above the PBL is attributable to scattered clouds, and the filtered extinction profile shows well mixed PBL aerosols with the AOD of 0.11 and the ratio of 126.9, which also tends to be larger than the general value for smoke mixed within PBL. Thus, for a clear indication, an AOD filter is applied in the following analysis to extract data pairs representing columns more likely impacted by smoke aerosols. Meanwhile, cloudy conditions where aerosol hygroscopic growth could complicate the analysis are excluded by focusing on the period of 4 to 8 August when clear-sky conditions prevailed.

**3.5.2 Evaluation of forecasted ratio**

Before the evaluation was performed, tests and visual examining were conducted to find the best criteria to filter observations and forecasts over smoke plume-affected areas, and the filters based on sAOD yielded more appropriate results than using AOD. A filter of sAOD>0.05 is employed for observations. While for model forecasts, due to the considerable range of magnitude of the forecasted fire emissions among the models, there is not a single sAOD threshold that can be appropriate for all the forecasts. Consequently, the threshold is chosen as 0.05 for most models, and reduced values of 0.02

(for HRRR-Smoke, WISC WRF-Chem, and FireWork) and 0.01 (for NCAR WRF-Chem, AIRPACT, ARQI, and NAQFC) are chosen to account for the lower sAOD magnitudes in these forecasts. Figure 21 shows an example of filtered observations and forecasts. Similar to the analysis in the previous subsection, for most models the high ratios correspond to the areas adjacent to the fire emission hotspot and transported smoke mixed down to the surface. It should be noted that for models with relatively low emissions, the areas impacted by smoke plumes cannot be well distinguished from background and other sources (e.g. Fig.

7), thus the distribution of the ratio is high biased due to inclusion of low background AOD values. Reduced ratios are shown in Fig. 21 over northwest Montana for some models (e.g., HRRR-Smoke, UCLA WRF-Chem, UIOWA WRF-Chem, WISC WRF-Chem, FireWork, NAQFC, NCAR WRF-Chem), indicating an aged smoke plume located above the PBL.

    Figure 22 shows the comparison of the probability distributions of the ratios for observations and model forecasts on 4-8 August, with the parameters of the log-normal fitting curves given Table 7. It should be mentioned that for models that had

relatively low fire emissions, as noted earlier (AIRPACT, ARQI, NAQFC, and NCAR WRF-Chem), the distributions can't unambiguously represent smoke plumes from the Williams Flats fire and are likely driven by other sources and background aerosols, so the results are not exactly comparable with the observations. These models tend to overpredict the ratios, which is consistent with the larger ratios obtained for non-smoke cases as shown in the previous subsection. Overall, the results suggest no single model performing the best simultaneously in terms of the mean and standard deviation of the fitted distribution.

Slight to large overpredictions of the ratios are shown for most models, except for UIOWA WRF-Chem which shows a negative bias. This tendency is consistent with other studies that show discrepancies in the AOD and $PM_{2.5}$ performance for biomass burning smoke (Mangold et al., 2011; Reddington et al., 2016, 2019). For the models with prominent smoke AOD impacts, a shift in the distribution of ratio compared to observations could be explained by issues in assumptions for aerosol optical properties (e.g., a too high mass extinction efficiency can bias the ratio distribution towards the lower end) or biases in

the PBL heights (e.g., shallower PBL can lead to positively biased ratios).

    Meanwhile, most models display a narrower distribution of the ratio than observed except for HRRR-Smoke. The wider distribution is expected since HRRR-Smoke is a smoke tracer model. The background $PM_{2.5}$ and AOD due to anthropogenic and other non-biomass pollution sources were not represented, which generates more extreme ratios. While, for the other models, the narrower distribution may suggest a smaller probability of extreme conditions (e.g., smoke aerosols lofted above

the PBL or confined near the surface). In other words, the narrow distribution tends to suggest smoke plumes getting mostly mixed in the PBL. Therefore, the misrepresentation of the distribution width can be improved by fixing issues with regards to

the vertical allocation of fire emissions that is estimated by parameterizations of plume injection. Further work is necessary to evaluate the contributions of relevant factors independently, e.g. fire size, fuel type, and thermo-dynamic stratifications.

As the fire activity changes drastically from day to day, the surface $PM_{2.5}$ to AOD ratios also show temporal variations, which can be found in the spread of the ratios over the hours of comparison (Fig. 23). The observations show a decreasing trend, especially for the 10th percentile, which is likely associated with deeper PBL and/or lofted smoke owing to stronger plume injections on 7 and 8 August compared to the previous days. However, the models rarely captured this feature or show flatter decreasing trends than observed, which can be associated with the less plume injections in models as days went by, consistent with the evaluation results against the DIAL-HSRL data in section 3.4.2.

The model representation of the ratios also suggests discrepancies between model performance for AOD and $PM_{2.5}$. Figure 24 shows how the model performance of $PM_{2.5}$ compares to that of AOD. Ideally, the target would be for the dots to fall to the 1:1 line, meaning that the $PM_{2.5}$ and AOD are biased by the same amount, and thus if emissions were corrected the forecast could achieve a close to zero bias in both quantities simultaneously. However, the dots often fall far from the 1:1 to line, and the further they are from the 1:1 line the stronger biases are generally seen in the $PM_{2.5}$ to AOD ratios (see Fig. S43 for the 1050    ideal relationship between NMBs of the ratio, AOD and $PM_{2.5}$), showing that the surface $PM_{2.5}$ to AOD ratio can be a good indicator of the discrepancies. While multiple forecasting systems show nearly unbiased ratios for some cases (i.e. the dots close to the 1:1 line), discrepancy in the AOD and surface $PM_{2.5}$ performance occurred for all the models. This means that the modeling systems cannot be fully improved by only revising the smoke emissions. Changes in structural configuration needs to be explored, including better representations of the aerosol optical properties, vertical allocation of the emissions, timing of 1055    plume injections, as well as the meteorological fields and thermodynamic processes in the lower troposphere (e.g. PBL heights and evolution).

## 4 Conclusions and recommendations to improve wildfire smoke forecasts

Predictions of wildfire smoke impacts on local to regional air quality by numerical forecasting systems have been a crucial 1060    tool in decision making and understanding of large wildfire events. However, the wildfire smoke forecasts relating to biomass burning emissions still bear a large uncertainty. In this paper, we present an intercomparison and evaluation of the wildfire smoke predictions produced by twelve state-of-the-art forecasting systems under the same framework. These forecast models are drastically different from each other with respect to the gas/aerosol emissions, complexity of chemical processes, and use of AOD data assimilation, etc. Focusing on the active burning period of the Williams Flats fire (3-9 August 2019), the 1065    evaluation is carried out to reveal model performance in multiple dimensions, including fire emissions, total column loading of smoke aerosols, surface $PM_{2.5}$ concentrations, and plume injection. The major findings and recommendations for improved wildfire smoke simulation and forecast are summarized as follows.

#### 4.1 Wildfire smoke emissions

The intercomparison of predicted smoke emissions suggests a substantial uncertainty in forecasted emission inventories. We find an overall large spread in daily total BB OC emissions, with the factor between the maximum and the minimum being about 20 to 50 on 5-9 August due to different methodologies used for the emission estimates. Overall, the FRP-based fire emissions are relatively higher than the satellite-fire-detection-based emissions. The large spread is likely driven by dry matter differences, and not emission factors, as reported by Carter et al. (2020). Additionally, the diurnal fire activity observed by the geostationary FRP data shows substantial day-to-day variations, while this can't be well represented by the fixed diurnal patterns used by the models. Discrepancies are shown in terms of the magnitude of diurnal variation, timing of the peak, as well as the nighttime fire activity. The limited representation of complex fire emission evolution in models greatly affected and challenged the performance on characterizing the impact of smoke on air quality.

#### 4.2 Total column smoke loading and surface impacts: sAOD and sPM$_{2.5}$

Statistics for sAOD and surface sPM$_{2.5}$ show well predicted magnitudes for a few models, while none of the models managed to realistically describe their spatial distributions. Nearly unbiased predictions are present for sAOD (UIOWA WRF-Chem and CAMS) and sPM$_{2.5}$ (FireWork, NAQFC, and RAQMS), and as expected, the high-resolution regional models show relatively better capability in depicturing the fine-scale plume structures. However, low correlation coefficients with $r < 0.55$ (0.36) and large errors with NME > 60 % (70 %) are found for all models for sAOD (sPM$_{2.5}$), which indicates inconsistencies in the spatiotemporal variations of smoke plumes between the forecasts and observations.

The FRP-based fire emissions and assimilation of satellite AOD tend to yield better model performance in terms of sAOD. In accordance with the emission magnitudes, the models driven by FRP-based fire emissions (CAMS, GEOS-FP, HRRR-Smoke, UCLA WRF-Chem, UIOWA WRF-Chem) produce larger sAOD and outperform than those driven by satellite-fire-detection-based emissions, showing less underestimations and better agreements for smoke plume areas. FRP-based emission inventories generally use AOD observations to tune their conversion of FRP to emissions, which could explain this advantage. Assimilating satellite AOD data in initial condition and forecasts (CAMS, RAQMS, GEOS-FP) or as constraints of fire emissions (UCLA WRF-Chem) also helped offer better performance. Other factors such as complexity of chemical mechanisms, chemical LBCs, horizontal resolution, initial time of forecast, and dynamic core used to drive the meteorological dispersion and transport do not seem to be determining for these metrics.

For sPM$_{2.5}$, however, the two operational air quality models for Canada and the U.S. (FireWork and NAQFC) perform the best, which often use surface monitoring stations data as the primary metric of their evaluation. In contrast, all models using FRP-based emissions tend to exhibit remarkable overestimations, especially for the stations that are closely impacted by fresh smoke plumes (MB > 6.0 μg m$^{-3}$, NMB > 40 %). The inconsistent biases in sAOD and sPM$_{2.5}$ demonstrate discrepancies in the performance for total column and surface air pollution levels, which is partly attributable to errors in vertical allocation of

emissions, as confirmed by the evaluation against DIAL-HSRL observations. The percentage of emissions injected into the free troposphere in models is generally assumed as a constant depending on fuel category. While, compared to the lidar data, most models show larger proportions of smoke within the PBL rather than injected into the free troposphere, which lead to a higher amount of smoke aerosols close to the land surface. Besides, model representation of PBL evolution, assumptions to diagnose optical properties of smoke aerosols, and missing chemical processes, e.g. enhanced evaporations of organic aerosols near the surface, are also potential factors associated with this discrepancy

The diurnal cycles of sPM$_{2.5}$ suggests additional inconsistencies in certain processes within models and observations. Overall, the overestimations of sPM$_{2.5}$ proved to be even higher during the late afternoon and nighttime hours, thus producing much stronger diurnal variations of sPM$_{2.5}$ than observed. The enlarged positive biases are presumably due to the overestimated proportion of emissions within the PBL. Earlier collapse of daytime PBL and lower PBL heights in the late afternoon and nighttime can also be a reason, associated with the disregarded sensible heat released from wildfire burning process and the PBL parameterizations used in models. An evaluation of surface PM$_{2.5}$ forecasts has suggested an inconsistency in the nocturnal PBL mixing within WRF-Chem (McKeen et al., 2007) that very little turbulent exchange is present during stable nighttime conditions, leading pollutants to build up unreasonably in the lowest model level. Further investigation is needed to understand more their relative contributions.

The day-to-day variation of model performance for sAOD illustrates the significant limitation of the assumption of persistence used for predicting fire emissions within the forecasting period. Remarkable underestimations of sAOD are shown on 8 August for all models, which appears to be mainly due to the drastic expansion of the burned area on the previous day not being captured by the models assuming persistence of fire activity. In contrast, the impact of persistence on the performance for sPM$_{2.5}$ on 8 August is not as significant as for sAOD, which is likely due to a cancellation of errors with overestimated proportions of smoke emissions within the PBL. Grouping sPM$_{2.5}$ sites by affected by fresh plume or not further confirms this behavior.

### 4.3 Plume injections

The vertical plume structure and plume injections are further quantitatively evaluated against the DIAL-HSRL observations acquired during FIREX-AQ. While the observations show a considerable day-to-day variation in the plume heights, all the models have smaller spreads. Overall, there is a large inter-model difference in the predicted plume heights. For the flight transects presenting injections within or around the PBL heights (3 and 6 August), most models show overestimated plume heights, and the models that usually put emissions below the PBL height perform better (e.g. CAMS, RAQMS, and GEOS-FP). Physical parameterizations of plume injections (e.g., Freitas et al., 2007, 2010, based on convective energy) yield slight overestimations on the 3$^{rd}$ and 6$^{th}$, and managed to depict enhanced injections on the 7$^{th}$ with stronger free-tropospheric injections, although with underpredictions. Additionally, insufficient representations are found for the strong injections owing to deep convection of the pyroCb on the 8$^{th}$. More work is needed to improve their inferring method from

input variables (e.g. fire intensity, fire size, meteorological fields) to plume injections to accurately depict different scenarios. It is also noteworthy that even for the models using the same plume injection scheme (e.g. NCAR WRF-Chem, UIOWA WRF-Chem, UCLA WRF-Chem, and WISC WRF-Chem), they often show substantially different results which might relate to uncertainties in meteorological fields, inputs to the plume rise parametrization, and grid resolution that need to be investigated further. The assessments for transported plumes (older than a day) show consistency with the injection performance as revealed for the fresh plume on the day before. It confirms the comparison results of models showing overestimated $sPM_{2.5}$, which tend to be associated with the earlier decay of plume injections and overestimated proportion of smoke loading with the PBL in the late afternoon. This result further emphasizes how the errors in timing and vertical allocation of emissions can propagate into model skill over transported plumes.

**4.4 Discrepancy in performance for surface $PM_{2.5}$ and AOD**

Surface $PM_{2.5}$/AOD is suggested as a measure to assess the vertical distribution of smoke as well as the discrepancy in model performance between the two individual terms. Note that the ratios can be dominated by the background or other emission sources when the fire emissions are low. The evaluation for probability distributions of the $PM_{2.5}$ to AOD ratios emphasize two aspects of model improvements. First, most models show positive biases in the means of the (log-normal) distributions, which suggests misrepresented aerosol optical properties (e.g., relatively lower mass extinction coefficients), and/or shallower mixing volume (e.g., relating to the lower PBL). Second, the narrower distributions indicate underpredicted possibility of cases with smoke plumes reaching the land surface or lofted above the PBL. The analysis of AOD, surface $PM_{2.5}$, and their ratios for coincident samples further confirms discrepancies in the model performance for AOD and $PM_{2.5}$. The biases in AOD and $PM_{2.5}$ can be effectively reduced simultaneously by adjusting the fire emissions for the models showing fewer discrepancies in their ratio. While, large discrepancies in the ratios point to the need of taking other factors into consideration, including the representation of aerosol optical properties, vertical allocation of smoke aerosols, and PBL evolution as well.

**4.5 Recommendations for future improvements**

Model evaluations of smoke emissions and sAOD suggests the advantage of using FRP-based fire emissions and data assimilation in providing less biased forecasts for total column smoke aerosol loading, compared to other differences in model features. The fact that all estimated fire emissions exhibit a large spread in magnitude demonstrate a need of future work to close the gap between these estimates and reduce their uncertainty. Leveraging FRP detections from geostationary satellites could provide beneficial information in improving the representation of temporal variation of fire emissions and to overcome the limitation of fixed diurnal patterns. This would be important especially for severe wildfires with unusual diurnal activity. As the forecasted air quality impacts still show limitations due to the persistence assumption, methodologies to describe and predict evolution of fire burning needs to be developed. A relevant system is available in Europe adopting a modeling strategy of hourly-sequential warm start runs (Solomos et al., 2015, 2019), with the emissions updated every hour using geostationary satellite detections. For the U.S., hourly emissions derived by blending GOES Advanced Baseline Imager (ABI) and polar-

orbiting satellite VIIRS fire products at 3-km spatial resolution will be incorporated into operational fire smoke models, namely NAQFC and HRRR-Smoke. These studies provide an efficient way of removing the minor or extinguished and at the same time to enhance emissions from actual burning fires, thus tracking the diurnal cycle of biomass burning. However, challenges still exist in making assumptions about the fire intensity and spread over the next hours/days. Also, compared to Europe, the fire intensities and their durations are on much larger scales in North America. The large spatial variability of fuels, complex topography, and different ecosystems in the U.S. adds the complexity.

Forecast performance for sAOD and surface sPM$_{2.5}$ and their discrepancies highlight key modeling processes to be improved. The proportion of emissions getting injected above the PBL appears to evolve with fire burning intensity, as indicated by the lidar observations, while all models tend to show underestimated free-troposphere smoke emissions and plume heights towards the days with enhanced burning. This illustrates a key need to improve plume rise parameterizations and vertical partition of fire emissions that is closely relevant to accurate representation of smoke plumes and the performance discrepancies. Modeled aerosol components and late afternoon/nighttime evolution of PBL structures are also closely relevant. Besides, although not evaluated specifically in this work, model assumptions of aerosol optical properties are important for both the AOD forecast performance and the discrepancies. Most models use optical properties that come embedded in the model version, and these are usually out of date. Thus, it is needed to more proactively update optical property modules that reflect all that we have learned from field campaigns and satellite observations, and make them easy to be incorporated into the community models. Given these various factors, future sensitivity and retrospective runs and analysis on these processes would help to identify the determinant factor(s) and their relative contributions for improving smoke forecasting.

**Data availability**

Flight observational data from FIREX-AQ, along with the Fuel2Fire emissions analysis, NIROPS burn area, and GOES-17 fire detections and FRP data during the field campaign are archived by NASA/LARC/SD/ASDC (https://doi.org/10.5067/SUBORBITAL/FIREXAQ2019/DATA001, https://www-air.larc.nasa.gov/missions/firex-aq/). AERONET and DRAGON network AOD observations can be accessed at the AERONET website (https://aeronet.gsfc.nasa.gov/new_web/DRAGON-FIREX-AQ_2019.html). MODIS MAIAC AOD retrievals (MCD19A2 Version 6) are available online (https://lpdaac.usgs.gov/products/mcd19a2v006/). Surface PM$_{2.5}$ observations are available at OpenAQ (https://openaq.org) and U.S. EPA's Air Data (https://www.epa.gov/outdoor-air-quality-data). Model forecasts and smoke emissions are provided by their modeling groups and collected by XY and PES, and the data are available upon request.

**Author contribution**

XY and PES designed the study and conducted UCLA WRF-Chem forecasts. XY analyzed the data with help from all the co-authors. RA, EJ, and GAG provided HRRR-Smoke forecast data and emissions. BP and AK provided RAQMS and WISC WRF-Chem data. PM and JC provided ARQI data. DD provided FireWork data. GRC and GF provided UIOWA WRF-Chem

data. JM and JH provided NAQFC data. RK and LE provided NCAR WRF-Chem data. FLH-T provided AIRPACT data. MP, RE, and V-HP provided CAMS data. AS, EG, and EW developed Fuel2Fire emission data. JWH, MF, and TS conducted DIAL-HSRL observations. AL and YW developed MODIS MAIAC AOD data. SK developed satellite products (fire emissions and AOD from VIIRS) used by some models either as input or for verification. BH and DMG conducted AERONET and DRAGON observations and helped with analysis. XY wrote the paper with contributions from co-authors.

**Competing interests**

The authors declare that they have no conflict of interest.

**Acknowledgments**

This work has been supported by the following grants:  NASA 80NSSC18K0629 (P. Saide), 80NSSC18K0681 (L. Emmons), 80NSSC18K0685 (A. Soja), NOAA NA18OAR4310107, NSF 2013461. Resources supporting this work were provided by the NASA High-End Computing (HEC) Program through the NASA Center for Climate Simulation (NCCS) at Goddard Space Flight Center (GSFC). R. Ahmadov and E. James thank NOAA's JPSS PGRR program for funding and the rest of the HRRR-Smoke team and collaborators for helping with the model development. The National Center for Atmospheric Research is sponsored by the National Science Foundation. We thank the FIREX-AQ project scientists Jim Crawford, Carsten Warneke, and Jack Dibb, as well as the pilots and crew of the NASA DC-8.

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

**Table 1. Summary of the forecast systems evaluated in this study. N/A is used when it is not applicable.**

| Model | Forecast domain | Institution | Horizontal grid spacing | Initial time (UTC) | Output interval | Dynamic core | Chemical mechanism complexity | Assimilation of satellite AOD | Chemical BC | Fire emission (Instrument, observable) | Parameterization of plume injection | Main refere |
|---|---|---|---|---|---|---|---|---|---|---|---|---|
| GEOS-FP | Global | NASA GMAO | 0.25° lat x 0.3125° lon | 00 | 3 hr | GEOS-5 | Simplified | yes | n/a | QFED (MODIS FRP) | Distribute within PBL | Randles et al |
| CAMS | Global | ECMWF | 0.4 deg | 00 | 3 hr | IFS | Detailed | yes | n/a | GFASv1.2 (MODIS FRP) | IS4FIRES (Sofiev et al., 2012) | Inness et al. |
| RAQMS | Global | University of Wisconsin | 1.0 deg | 12 | 6 hr | RAQMS | Detailed | yes | n/a | RAQMS (MODIS, hotspots) | Distribute within PBL for high severity, lowest layer for low and medium severity fires | Pierce et al. |
| WISC WRF-Chem | CONUS | University of Wisconsin | 8 km | 00 | 1 hr | WRF-ARW | Simplified | yes (IC) | RAQMS | PREP-CHEM (GOES-15, hotspots) | Freitas et al. (2007, 2010) | Skamarock e Grell et al. (2 |
| HRRR-Smoke | CONUS | NOAA/GSL | 3 km | 00 | 1 hr | WRF-ARW | Smoke tracer | no | RAP-Smoke | (VIIRS and MODIS FRP) | Freitas et al. (2007, 2010), Paugam et al. (2015) | Ahmadov et |
| NAQFC (CMAQ) | CONUS | NOAA NCEP | 12 km | 12 | 1 hr | GFS (FV3) | Detailed | no | Monthly climatology | HMS/Bluesky (hotspots) | Pouliot et al. (2005) | Lee et al. (20 |
| ARQI (FireWork experimental) | NW US & SW Canada | ECCC | 2.5 km | 12 | 1 hr | GEM 4.9.8 | Detailed | no | MOZART monthly climatology | CFFEPSv4.0 (hotspots) | Modified based on Chen et al. (2019) | Makar et al. |
| NCAR WRF-Chem | CONUS | NCAR/ACOM | 12 km | 00 | 1 hr | WRF-ARW | Simplified | no | WACCM | FINNv1.5 (MODIS, hotspots) | Freitas et al. (2007, 2010) | Kumar et al. |
| UCLA WRF-Chem | W US | UCLA | 4 km | 00 | 1 hr | WRF-ARW | Smoke tracer | yes (inversion of fire emis.) | Monthly climatology | QFEDv2.5 (EOS-MODIS FRP) | Freitas et al. (2007, 2010) | Saide et al. ( |
| AIRPACT (CMAQ) | NW US | Washington State University | 4 km | 08 | 1 hr | WRF-ARW | Detailed | no | WACCM | SMOKE, SMARTFIRE2 (hotspots) | A modified DEASCO3 plume rise approach (Mavko and Morris, 2013) | Herron-Thor |
| UIWOAWRF-Chem | W US | University of Iowa | 8 km | 12 | 3 hr | WRF-ARW | Detailed | no | WACCM | QFED (MODIS FRP) | Freitas et al. (2007, 2010) | Skamarock e Grell et al. (2 |
| FireWork | North America | ECCC | 10 km | 12 | 1 hr* | GEM 5.0 | Detailed | no | MOZART monthly climatology | CFFEPSv2.1 (hotspots) | Chen et al. (2019), Anderson et al. (2011) | Pavlovic et a et al. (2019) |

*Note that AOD forecasts by FireWork were only available at +9 hours and +21 hours relative to the initialization time.

**Table 2. Definition of categories for a binary event.**

| Category | | Observed | |
|---|---|---|---|
| | | Yes | No |
| Forecasted | Yes | a | b |
| | No | c | d |

5    **Table 3. Statistics of comparison of modeled smoke AOD (sAOD) against MODIS MAIAC AOD observations for twelve models on 4-8 August 2019. The statistics are shown for all data (All), fresh-plume areas (Fp), and other areas (Ot), respectively. The best four models per statistical metric are highlighted in bold.**

| Model | n | | | r | | | MB | | | ratio | | | NMB (%) | | | RMSE | | | NME (%) | | |
|---|---|---|---|---|---|---|---|---|---|---|---|---|---|---|---|---|---|---|---|---|---|
| | All | Fp | Ot | All | Fp | Ot | All | Fp | Ot | All | Fp | Ot | All | Fp | Ot | All | Fp | Ot | All | Fp | Ot |
| ARQI | 622623 | 17517 | 605106 | 0.23 | 0.29 | 0.21 | -0.067 | -0.167 | -0.064 | 0.23 | 0.18 | 0.24 | -76.8 | -81.6 | -76.5 | 0.132 | 0.258 | 0.127 | 83.7 | 84.2 | 83.7 |
| HRRR-Smoke | 460609 | 13122 | 447487 | **0.43** | 0.27 | **0.46** | -0.046 | **-0.055** | -0.046 | 0.47 | **0.73** | 0.45 | -52.9 | **-27.1** | -54.7 | 0.131 | 0.365 | **0.117** | 79.2 | 92.0 | 78.3 |
| AIRPACT | 238236 | 7241 | 230995 | 0.18 | 0.13 | 0.19 | -0.055 | -0.085 | -0.054 | 0.32 | 0.37 | 0.32 | -68.1 | -62.8 | -68.3 | 0.125 | **0.204** | 0.121 | 79.4 | 92.7 | 78.7 |
| UCLA WRF-Chem | 255865 | 7305 | 248560 | **0.47** | **0.41** | **0.47** | -0.033 | -0.103 | **-0.030** | 0.62 | 0.50 | 0.63 | -37.7 | -50.2 | **-36.7** | **0.110** | **0.228** | **0.104** | **63.7** | 67.9 | **63.4** |
| UIOWA WRF-Chem | 79106 | 2235 | 76871 | 0.31 | 0.32 | 0.30 | **-0.004** | 0.071 | **-0.006** | 0.96 | 1.34 | 0.93 | **-4.3** | 34.5 | **-7.1** | 0.165 | 0.518 | 0.143 | 87.1 | 113.5 | 85.2 |
| WISC WRF-Chem | 72251 | 2119 | 70132 | 0.07 | 0.06 | 0.08 | -0.041 | -0.136 | -0.038 | 0.49 | 0.27 | 0.51 | -51.0 | -73.3 | -49.3 | 0.130 | 0.237 | 0.126 | 82.3 | 86.9 | 82.0 |
| FireWork | 52917 | 1508 | 51409 | **0.35** | 0.26 | **0.36** | -0.052 | **-0.066** | -0.052 | 0.39 | **0.68** | 0.37 | -60.6 | **-32.3** | -62.6 | **0.124** | 0.262 | 0.118 | 75.5 | **79.2** | 75.2 |
| NAQFC | 37840 | 1056 | 36784 | 0.13 | **0.33** | 0.12 | -0.052 | -0.154 | -0.049 | 0.40 | 0.24 | 0.42 | -59.6 | -76.0 | -58.4 | 0.145 | 0.235 | 0.142 | 82.4 | 82.1 | 82.4 |
| NCAR WRF-Chem | 34304 | 978 | 33326 | 0.20 | 0.29 | 0.16 | -0.070 | **-0.166** | -0.067 | 0.13 | 0.11 | 0.13 | -87.4 | -89.4 | -87.2 | 0.131 | 0.234 | 0.126 | 90.2 | 90.6 | 90.1 |
| GEOS-FP | 8680 | 246 | 8434 | 0.32 | **0.39** | 0.31 | **-0.031** | **-0.016** | -0.032 | **0.64** | **0.92** | 0.62 | **-36.0** | **-8.0** | -38.0 | 0.134 | 0.337 | 0.123 | **73.0** | 89.6 | **71.8** |
| CAMS | 4199 | 116 | 4083 | **0.50** | **0.48** | **0.48** | **-0.005** | **0.007** | **-0.005** | **0.94** | **1.03** | 0.94 | **-5.6** | **3.3** | **-6.3** | **0.105** | **0.205** | **0.101** | **61.4** | **59.1** | **61.6** |
| RAQMS | 610 | 26 | 584 | 0.26 | 0.28 | 0.26 | **-0.018** | -0.068 | **-0.016** | 0.78 | 0.56 | 0.80 | **-21.8** | -44.4 | **-19.9** | **0.109** | **0.105** | **0.109** | 68.2 | **56.5** | 69.2 |

Table 4. Statistics of surface smoke PM$_{2.5}$ enhancements (sPM$_{2.5}$) compared against hourly observations from AirNow stations on 4-9 August 2019. The models are ranked by horizontal grid resolution. The columns of "All" are the results for all the stations, "Fp" refers to fresh-plume stations, and "Ot" refers to the other stations. The total number of pairs of model and observation data points included in the comparisons are 11083 over all 83 stations, 1930 for the 14 Fp stations, and 9153 for the 69 Ot stations. The best four models for each statistical metric and each station classification are highlighted in bold. For each statistical metric and each model, the better performance between the Fp and Ot stations for each of the metrics is underlined.

| Model | r | | | MB (ug m$^{-3}$) | | | ratio | | | NMB (%) | | | RMSE (μg m$^{-3}$) | | | NME (%) | | |
|---|---|---|---|---|---|---|---|---|---|---|---|---|---|---|---|---|---|---|
| | All | Fp | Ot | All | Fp | Ot | All | Fp | Ot | All | Fp | Ot | All | Fp | Ot | All | Fp | Ot |
| ARQI | 0.19 | _0.16_ | **0.10** | -1.80 | -4.57 | **_-1.21_** | 0.77 | **0.69** | **_0.81_** | -22.60 | -31.10 | _-18.50_ | 11.90 | 23.10 | **_7.70_** | **72.70** | 79.50 | **_69.50_** |
| HRRR-Smoke | **0.28** | **_0.29_** | **0.10** | -1.18 | 6.67 | _-2.83_ | **0.85** | 1.45 | _0.57_ | **-14.80** | 45.40 | _-43.30_ | 24.90 | 49.20 | _15.40_ | 100.30 | 126.00 | 88.10 |
| AIRPACT | 0.19 | _0.12_ | 0.08 | -2.34 | **-4.41** | _-1.90_ | 0.71 | 0.70 | _0.71_ | -29.30 | **-30.00** | **_-29.00_** | **10.70** | 19.10 | **_7.90_** | **74.60** | 81.50 | **_71.30_** |
| UCLA WRF-Chem | 0.10 | 0.03 | _0.12_ | 3.38 | 8.60 | _2.27_ | 1.42 | 1.59 | _1.35_ | 42.40 | 58.60 | _34.80_ | 42.90 | 90.30 | _22.60_ | 112.40 | 141.50 | _98.60_ |
| UIOWA WRF-Chem | 0.09 | 0.03 | _0.04_ | 4.66 | 10.49 | _3.43_ | 1.59 | 1.71 | _1.52_ | 58.60 | 71.40 | _52.50_ | 33.90 | 70.90 | _18.30_ | 124.20 | 139.90 | _116.80_ |
| WISC WRF-Chem | 0.04 | _0.03_ | 0.02 | 3.28 | **_-1.29_** | 4.24 | 1.41 | **_0.91_** | 1.65 | 41.20 | **_-8.80_** | 64.80 | 16.00 | **18.80** | _15.30_ | 101.90 | _79.90_ | 112.30 |
| FireWork | **0.35** | **_0.29_** | **0.22** | **-0.32** | 0.55 | **_-0.50_** | **0.96** | **_1.04_** | 0.92 | **-4.00** | 3.70 | **_-7.70_** | **10.80** | 19.50 | **_7.80_** | **72.20** | **77.80** | **_69.50_** |
| NAQFC | **0.26** | **_0.28_** | **0.16** | **0.02** | -2.40 | _0.53_ | **1.00** | 0.84 | _1.08_ | **0.30** | -16.30 | _8.20_ | **10.40** | **15.00** | _9.20_ | **71.40** | **62.90** | 75.40 |
| NCAR WRF-Chem | 0.21 | _0.23_ | 0.03 | -5.78 | -10.73 | _-4.74_ | 0.27 | 0.27 | _0.28_ | -72.60 | -73.00 | _-72.40_ | **10.80** | **18.10** | _8.40_ | 81.20 | **_79.20_** | 82.10 |
| GEOS-FP | 0.16 | _0.10_ | 0.09 | 5.07 | 11.88 | _3.63_ | 1.64 | 1.81 | _1.56_ | 63.70 | 80.90 | _55.50_ | 29.70 | 59.80 | _17.80_ | 109.40 | 138.50 | _95.60_ |
| CAMS | **0.29** | **_0.22_** | **0.18** | 5.51 | 6.00 | _5.41_ | 1.69 | _1.41_ | 1.83 | 69.20 | _40.80_ | 82.70 | 12.50 | 20.20 | _10.20_ | 102.80 | _85.90_ | 110.90 |
| RAQMS | 0.11 | _0.13_ | 0.00 | **-0.44** | -5.29 | _0.59_ | **0.95** | 0.64 | _1.09_ | **-5.50** | -36.00 | _9.00_ | **9.80** | **15.70** | _8.10_ | 75.50 | **_67.10_** | 79.50 |

**Table 5. Summary of DC-8 flight transects selected in this study.**

| Flight transect | Date in August 2019 (PDT) | Start time (hh:mm, PDT) | End time (hh:mm, PDT) | Smoke plume sampled (F: fresh; A: aged) |
|---|---|---|---|---|
| D3T1 | 03 | 14:44 | 15:00 | F |
| D3T2 |  | 17:06 | 17:26 | F |
| D3T3 |  | 19:33 | 19:49 | F |
| D6T1 | 06 | 11:45 | 12:07 | F |
| D6T2 |  | 13:28 | 13:45 | F |
| D6T3 |  | 14:46 | 14:59 | F |
| D7T1 | 07 | 14:34 | 14:53 | A |
| D7T2 |  | 15:30 | 15:55 | A |
| D7T3 |  | 16:02 | 16:22 | F |
| D7T4 |  | 17:48 | 18:05 | F |
| D7T5 |  | 19:21 | 19:41 | F |
| D8T1 | 08 | 14:35 | 15:07 | A |
| D8T2 |  | 17:09 | 17:36 | A |
| D8T3 |  | 18:11 | 18:18 | F |
| D8T4 |  | 18:18 | 18:27 | F |

**Table 6. Statistics of the forecasted median plume heights for the 11 transects that sampled through fresh plumes compared against DIAL-HSRL observations. The best four members for each of the columns has been highlighted in bold.**

| Model | n | r | ratio | MB (km) | NMB (%) | RMSE (km) | NME (%) |
|---|---|---|---|---|---|---|---|
| ARQI | 8 | -0.27 | 1.20 | **-0.76** | **-13.1** | 3.11 | 43.3 |
| HRRR-Smoke | 11 | 0.73 | 1.30 | **0.27** | **6.4** | **1.88** | 36.1 |
| AIRPACT | 8 | 0.20 | **1.04** | -1.45 | -24.9 | 2.97 | 44.5 |
| UCLA WRF-Chem | 10 | **0.85** | **1.14** | **-0.33** | **-7.0** | **1.83** | **29.9** |
| UIOWA WRF-Chem | 8 | 0.74 | **0.88** | -1.57 | -26.9 | 2.74 | **34.4** |
| WISC WRF-Chem | 11 | 0.49 | 0.75 | -1.66 | -39.1 | 2.71 | 50.6 |
| FireWork | 11 | **0.83** | 1.49 | **0.72** | **17.0** | **2.02** | 42.9 |
| NCAR WRF-Chem | 9 | **0.91** | 1.52 | 1.25 | 24.1 | **1.83** | **30.8** |
| GEOS-FP | 10 | **0.88** | **0.87** | -1.40 | -29.9 | 2.39 | 38.1 |
| CAMS | 10 | 0.79 | 0.83 | -1.36 | -29.2 | 2.35 | 34.6 |
| RAQMS | 8 | 0.75 | 0.70 | -1.82 | -31.1 | 2.64 | **32.8** |

25 **Table 7. Parameters of mean ($\mu$) and standard deviation ($\sigma$) (natural logarithmic values) of the log-normal distributions fitted for surface PM$_{2.5}$/AOD ratios using observations and model forecasts on 4-8 August 2019. The subscript "o" denotes observation, and "m" denotes model. The relative difference is calculated as ($\mu_m - \mu_o$)/$\mu_o$ and ($\sigma_m - \sigma_o$)/$\sigma_o$. The best four values for the relative differences are shown in bold.**

| Model | n | $\mu_o$ | $\exp(\mu_o)$ | $\mu_m$ | $\exp(\mu_m)$ | relative difference of $\mu$ | $\sigma_o$ | $\sigma_m$ | relative difference of $\sigma$ |
|---|---|---|---|---|---|---|---|---|---|
| ARQI | 428 | 4.06 | 57.89 | 4.37 | 78.87 | 7.6% | 0.54 | 0.39 | -27.6% |
| HRRR-Smoke | 324 | 4.08 | 59.14 | 4.24 | 69.28 | **3.9%** | 0.50 | 1.10 | 120.2% |
| AIRPACT | 429 | 4.06 | 57.71 | 4.18 | 65.56 | **3.1%** | 0.53 | 0.36 | -32.1% |
| UCLA WRF-Chem | 353 | 4.11 | 60.92 | 4.29 | 73.02 | 4.4% | 0.51 | 0.50 | **-1.7%** |
| UIOWA WRF-Chem | 414 | 4.05 | 57.54 | 3.75 | 42.50 | -7.5% | 0.50 | 0.44 | **-11.7%** |
| WISC WRF-Chem | 344 | 4.07 | 58.56 | 4.69 | 108.85 | 15.2% | 0.53 | 0.44 | **-17.0%** |
| FireWork | 416 | 4.07 | 58.77 | 4.39 | 81.01 | 7.9% | 0.53 | 0.53 | **0.6%** |
| NAQFC | 432 | 4.06 | 57.89 | 4.89 | 132.99 | 20.5% | 0.51 | 0.38 | -26.5% |
| NCAR WRF-Chem | 279 | 4.04 | 57.03 | 4.42 | 83.35 | 9.4% | 0.52 | 0.33 | -36.3% |
| GEOS-FP | 274 | 4.06 | 57.90 | 4.27 | 71.35 | 5.1% | 0.49 | 0.35 | -28.2% |
| CAMS | 412 | 4.06 | 57.75 | 4.17 | 64.85 | **2.9%** | 0.51 | 0.27 | -48.0% |
| RAQMS | 311 | 4.07 | 58.29 | 4.17 | 64.80 | **2.6%** | 0.52 | 0.39 | -23.9% |

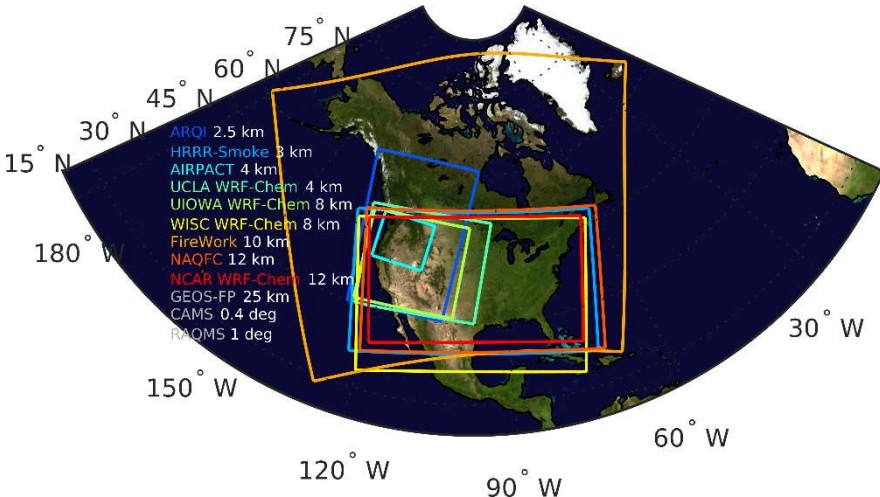

**Figure 1. Map of forecast domains for all regional models included in this study. The horizontal grid spacings of the models are labeled in white (see also Table 1). Note that the domains of the three global forecasting systems, GEOS-FP, CAMS, and RAQMS are not shown.**

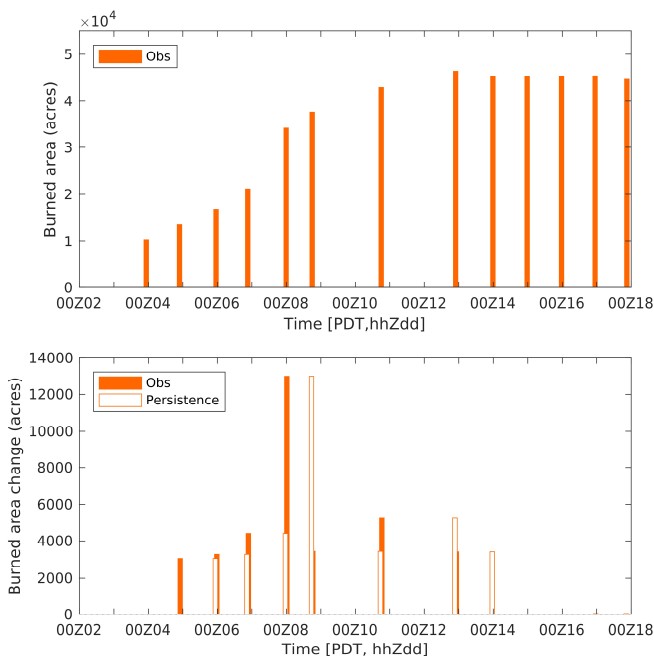

**Figure 2. Time series of burned area from the NIROPS IR observations for the Williams Flats Fire on 2-17 August 2019 (upper panel) and day-to-day increment of burned area on 4-9 August (lower panel). The observation hour and day is shown in PDT (UTC-**

**7) in "hhZdd" (hour and day). Daily increment of burned area based on the assumption of persistent fire activity are shown by the open orange bar in the lower panel, which equals to the observed value on the previous day.**

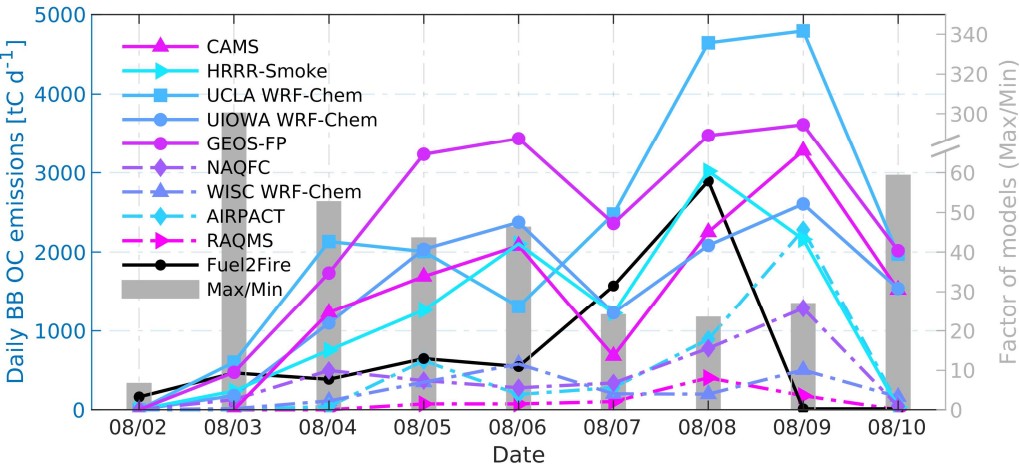

**Figure 3. Time series of daily total biomass burning OC emissions from the Williams Flats Fire predicted in different models. Models using FRP-based emissions are shown with solid lines, and those using hotspot-based emissions are shown with dash-dot lines. The solid black line with dots stands for Fuel2Fire emissions analysis. Grey bars represent factors between the maximum and minimum for all models.**

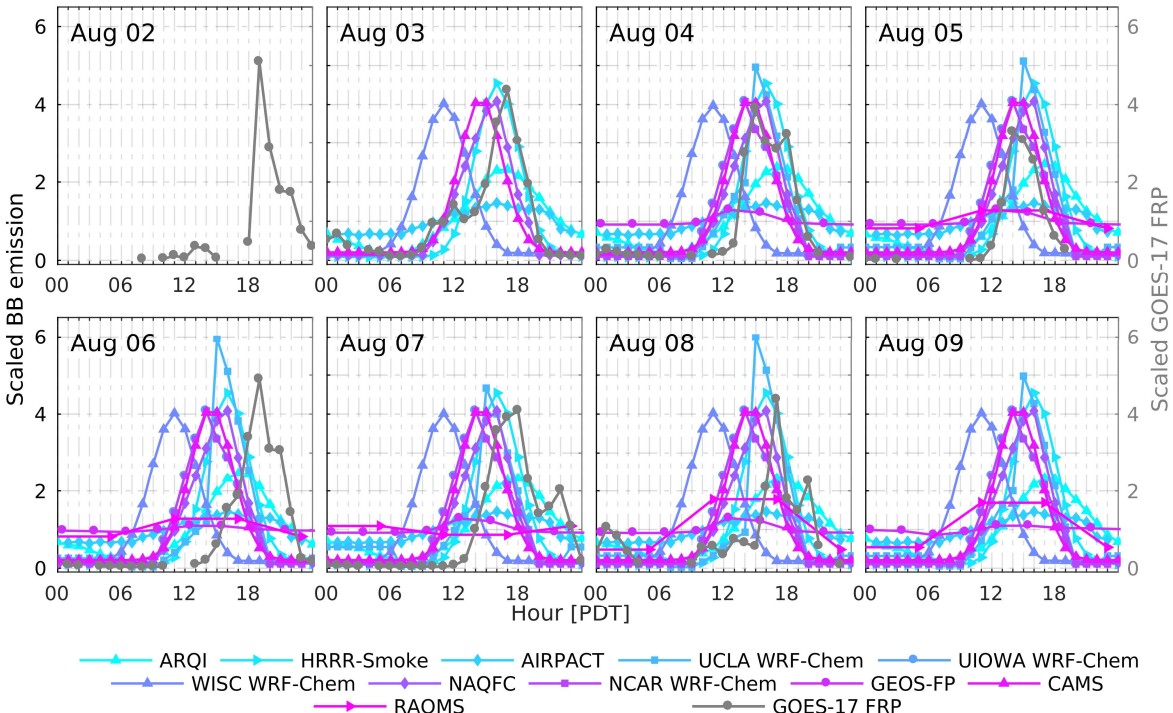

**Figure 4. Diurnal variation factors of biomass burning emissions from the Williams Flats Fire on 2-10 August 2019 scaled by daily average value. The colored lines with markers represent different models. The grey lines with dots represent the scaled GOES-17 Fire Radiation Power (FRP). The lines for NCAR WRF-Chem overlap with UIOWA WRF-Chem.**

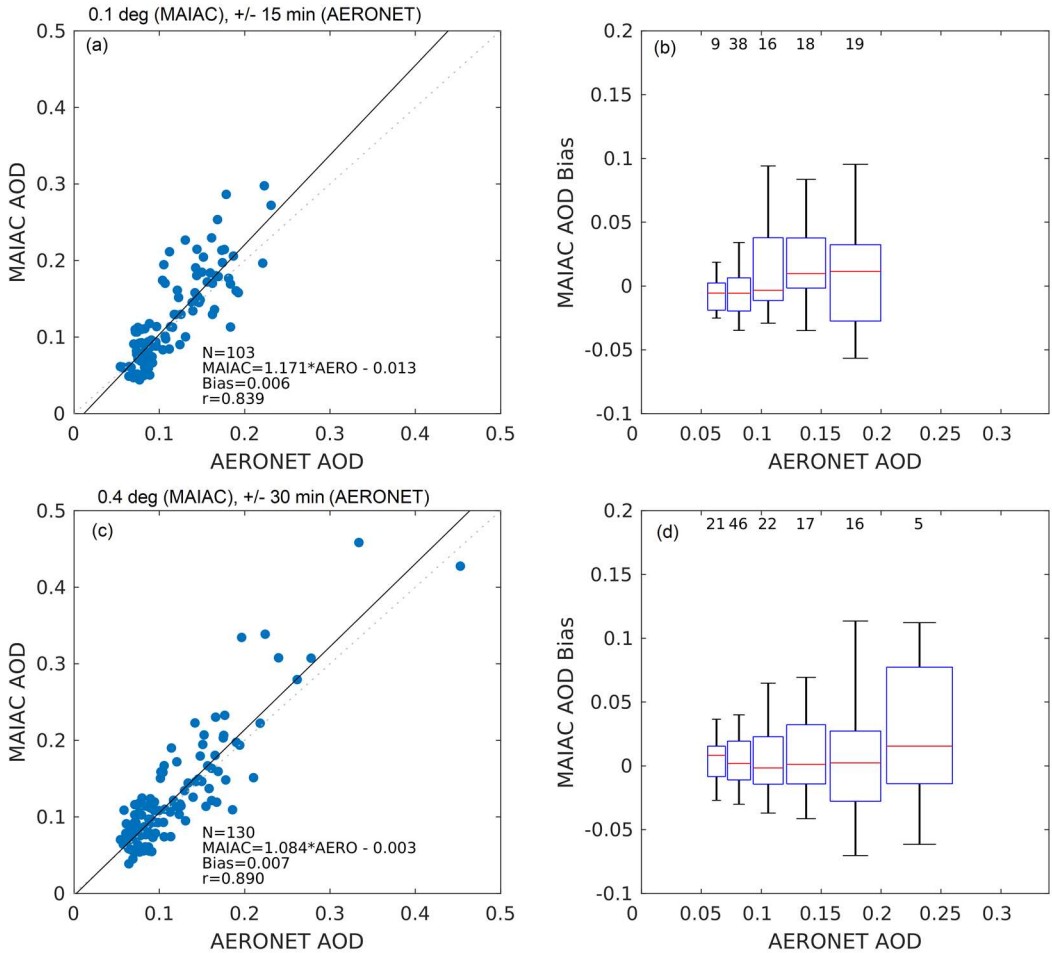

**Figure 5. Scatterplots (a, c) of the relationship between AERONET AOD and MAIAC during 3-8 August 2019. Results are shown for two sets of data collocation methods. The dotted line represents the 1:1 line, and the solid line represents the linear regression model provided in the figure. The box and whisker plots (b, d) show the dependence of MAIAC biases compared against AERONET. Missing boxes are due to the lack of matchups (<5) in that AOD bin. The edges of boxes and the red line represent 25[th], 75[th] percentiles**

55   **and median. The whiskers represent 5[th] and 95[th] percentiles. The numbers at the top of (b) and (d) are the amounts of matchups in each bin of AERONET AOD.**

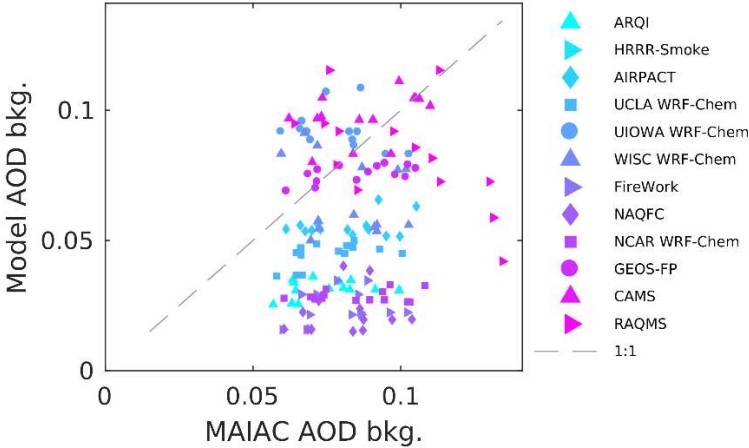

**Figure 6. Scatter plot of background AOD values derived from MODIS MAIAC AOD data and modeled results per hourly scene during 4–8 August 2019. Note that the background of HRRR-Smoke is clean since it doesn't include non-smoke emissions.**

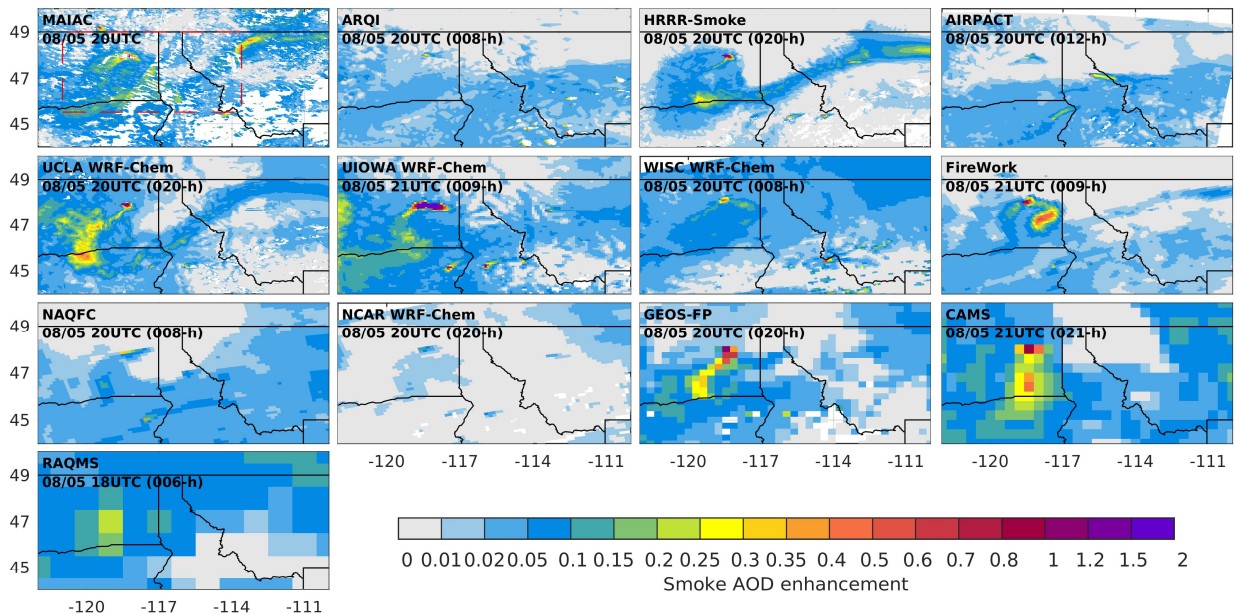

**Figure 7. Map of observed smoke AOD enhancements (sAOD) by MODIS MAIAC data and the model forecasts for 20:00 UTC 5 August 2019. The valid time of forecast is shown on each panel (with the lead time in parenthesis, e.g. 008-h). The red dashed box on the observation map represents the area of interest for the evaluation of sAOD magnitude and spatial extent of the smoke plume.**

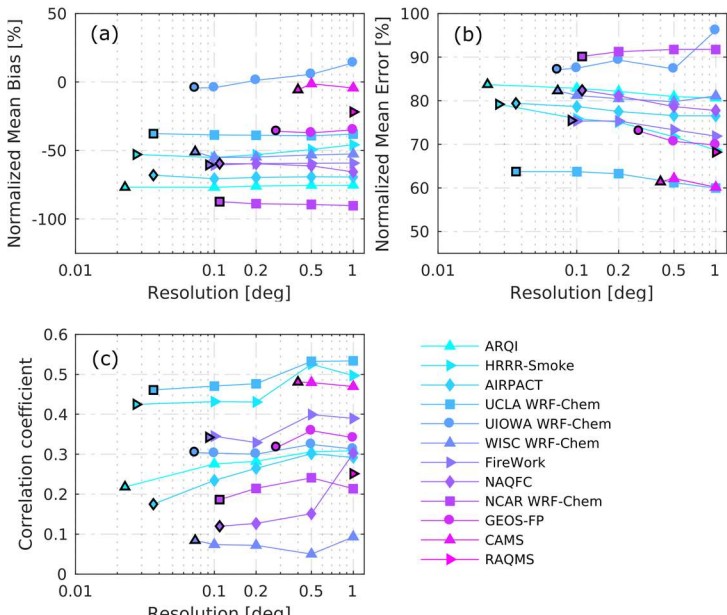

Figure 8. Model performance statistics for smoke AOD enhancements (sAOD) compared to MODIS MAIAC retrievals on 4-8 August 2019 at different horizontal re-gridding resolutions: (a). normalized mean bias (NMB); (b). normalized mean error (NME); (c). correlation coefficient (*r*). The x axes are shown in log scale. Each line represents one model (see the legends for model names, and the models are ranked by horizontal grid resolution). The markers with black edges indicate the results for the original model grid resolutions.

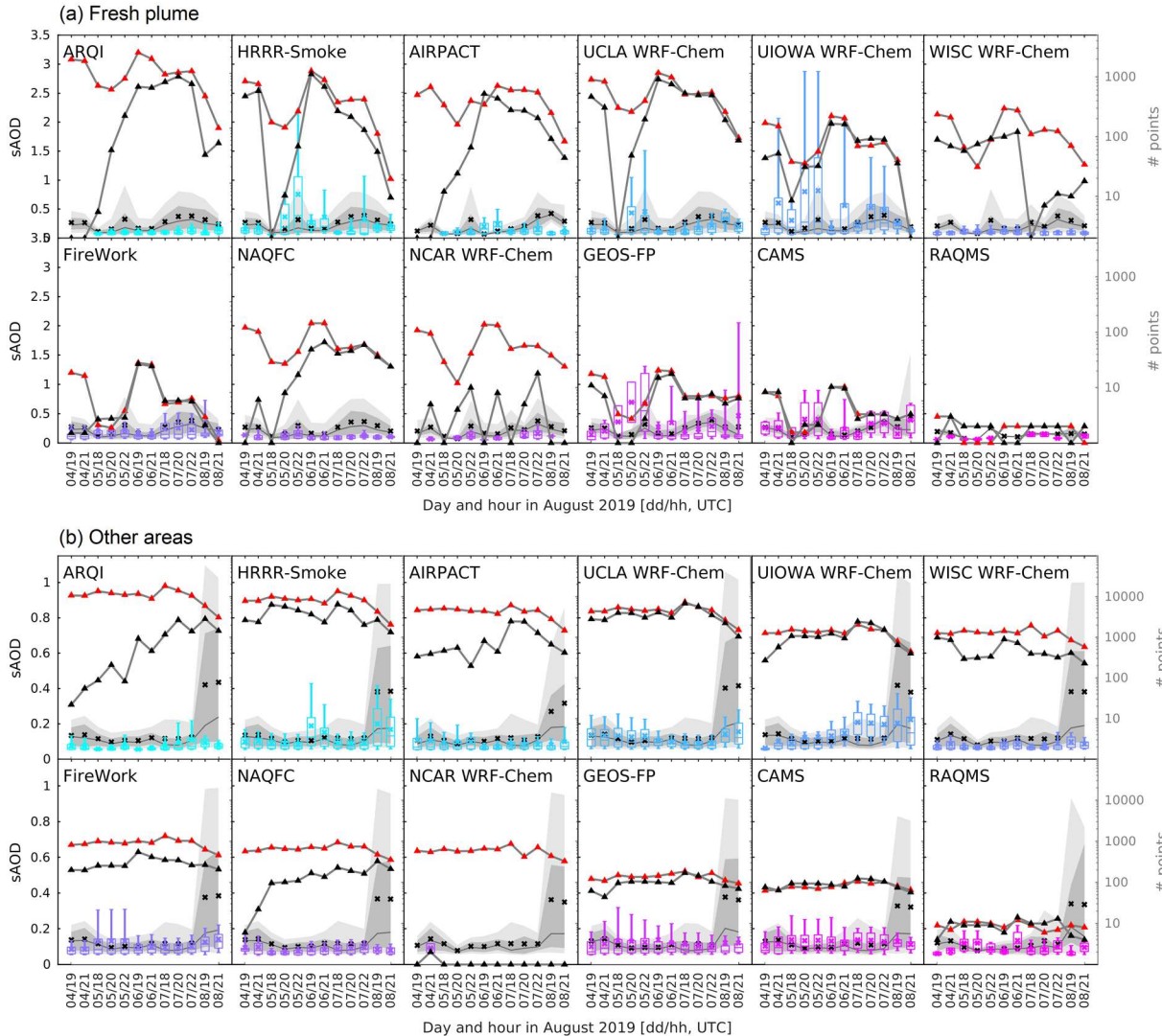

**Figure 9. Boxplots of predicted and observed smoke AOD enhancements (sAOD) for sAOD>0.05 per hourly snapshot of MODIS measurements over the areas of (a) fresh plume and (b) other areas. The central mark of a box indicates the median, and the bottom and top edges of the box indicate the 25th and 75th percentiles of model results, respectively. The whiskers extend to the 10th and 90th percentiles. The colored and black "x" signs are the average value for model and observations, respectively. The grey solid lines with red and black triangles represent the observed and modeled total number of grid cells incorporated into the comparison (sAOD>0.05).**

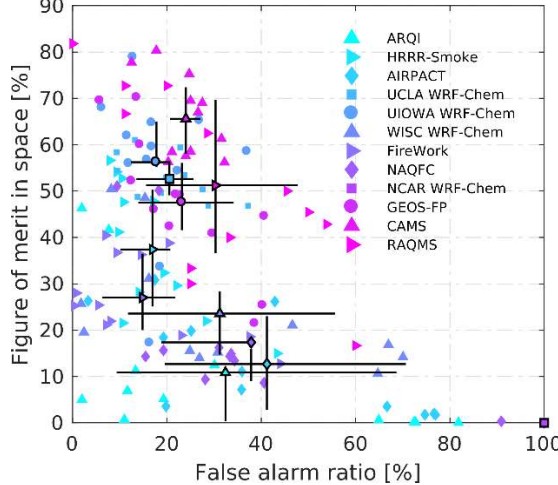

**Figure 10. FMS and FAR scores for fire smoke AOD exceedance events (sAOD>0.05) forecasted by models compared against MODIS MAIAC AOD retrievals per hourly snapshot during 4-8 August 2019. The scores are derived using re-gridded satellite data at the original grid resolutions of models. For each model, the markers with black edges represent median values and the horizontal and vertical black bars are the 25th to 75th percentiles.**

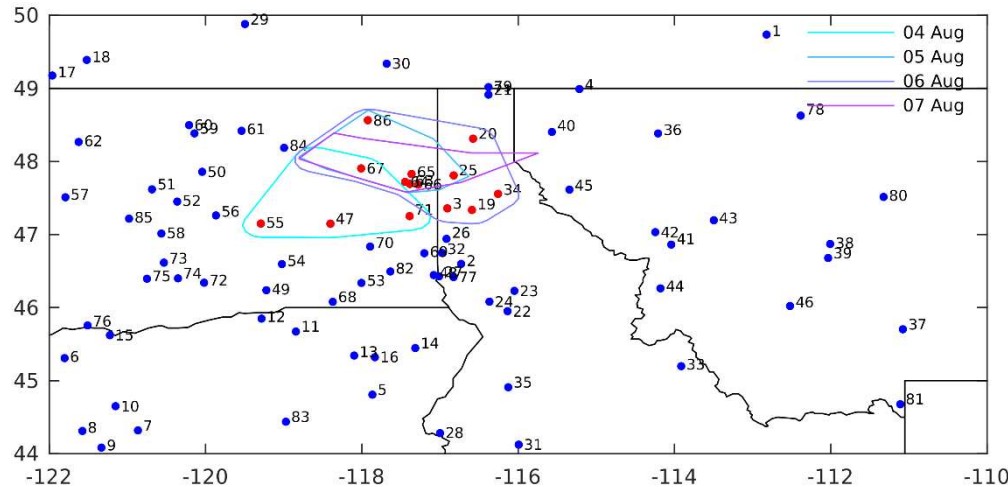

**Figure 11. Locations of the AirNow monitoring stations providing surface PM2.5 mass concentrations. The colored lines represent boundaries of fresh fire smoke plumes from the Williams Flats Fire on 4-7 August 2019, which are visually defined using the GOES-17 visible images and MODIS MAIAC AOD. The red dots stand for "fresh-plume stations" located within the fresh plume areas, and the blue dots stand for all the other stations.**

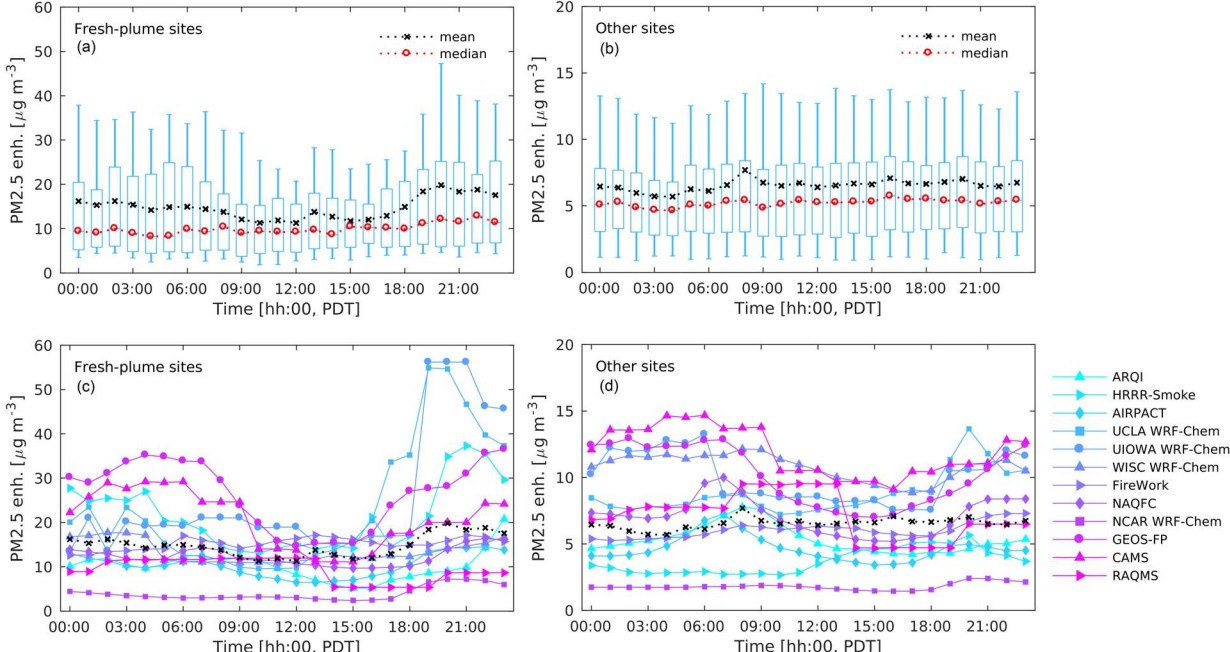

**Figure 12. Diurnal variations of observed sPM₂.₅ (a and b) and comparison of the modeled and observed mean sPM₂.₅ (c and d) over the two categories of monitoring stations: fresh-plume (Fp) sites and other (Ot) sites. In (a) and (b) the observed mean sPM₂.₅ are shown by the black dotted lines with crosses, and observed medians are shown by the red dotted lines with open circles.**

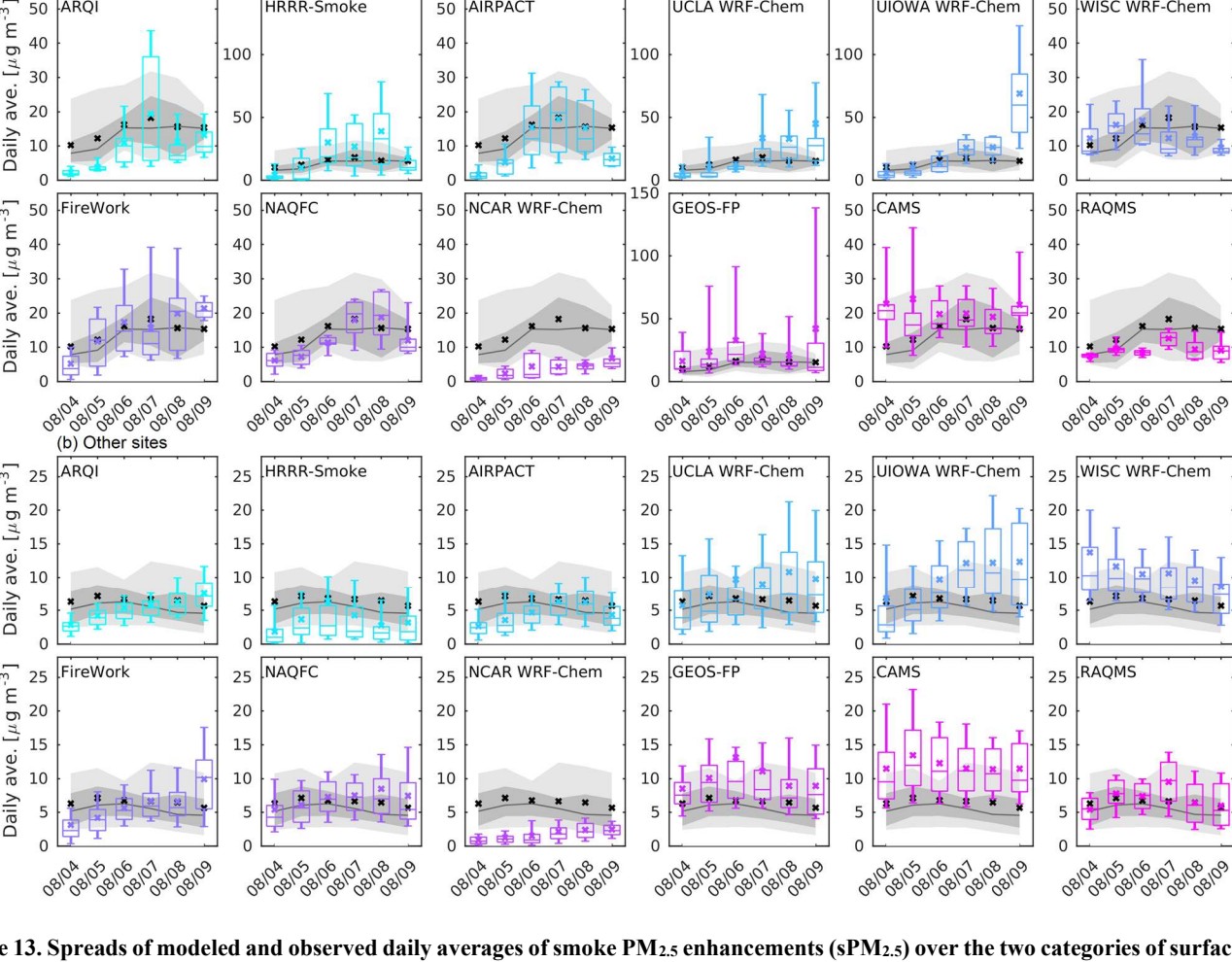

**Figure 13. Spreads of modeled and observed daily averages of smoke PM$_{2.5}$ enhancements (sPM$_{2.5}$) over the two categories of surface sites, i.e. (a) fresh-plume sites and (b) other sites during 4-9 August 2019. The dates are labeled at the x-axes as "month/day". The box and whiskers (as in Fig. 9) stand for model results. The corresponding ranges of the 10$^{th}$ to 90$^{th}$ percentiles for observations are represented by the light grey shading, and the range of 25$^{th}$ to 75$^{th}$ percentiles are represented by the medium grey shading. The observed median and mean are denoted by the dark grey lines and black crosses, respectively.**

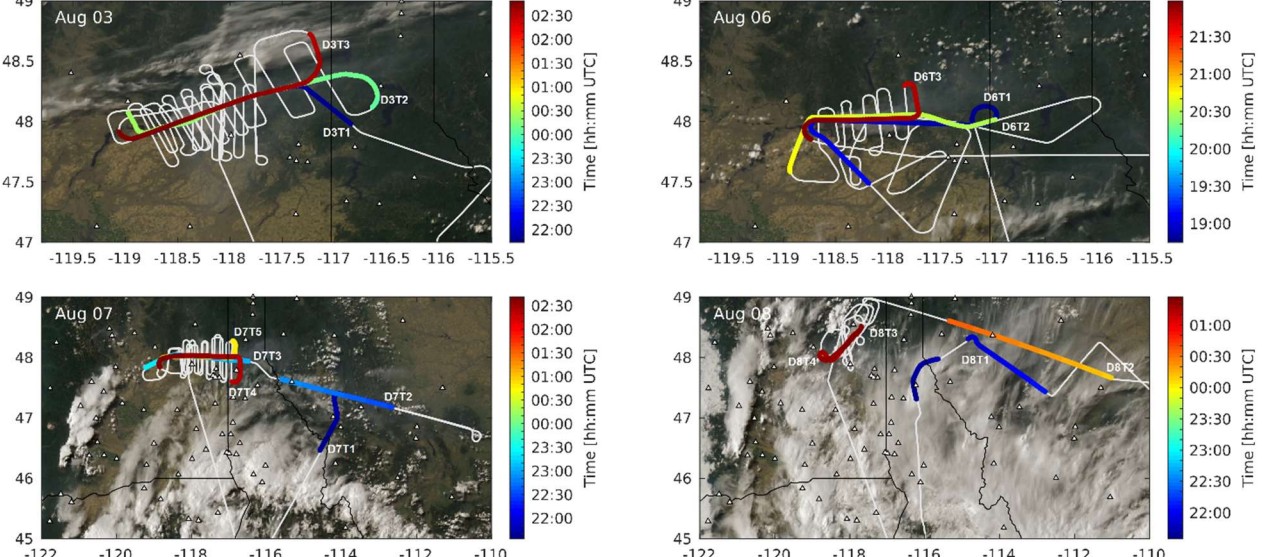

**Figure 14. Maps of DC-8 flight tracks on 3, 6, 7, and 8 August 2019, operated during the FIREX-AQ field campaign. The selected flight transects (see Table 5 for details) are colored by the observation time (UTC) on each day, overlaid on visible images of GOES-17 at 17:01 PDT. Note that the map coverage for 7 and 8 August are larger, in order to show the flight transects that sampled the aged plume. White triangles represent locations of surface monitoring sites of the AirNow network.**

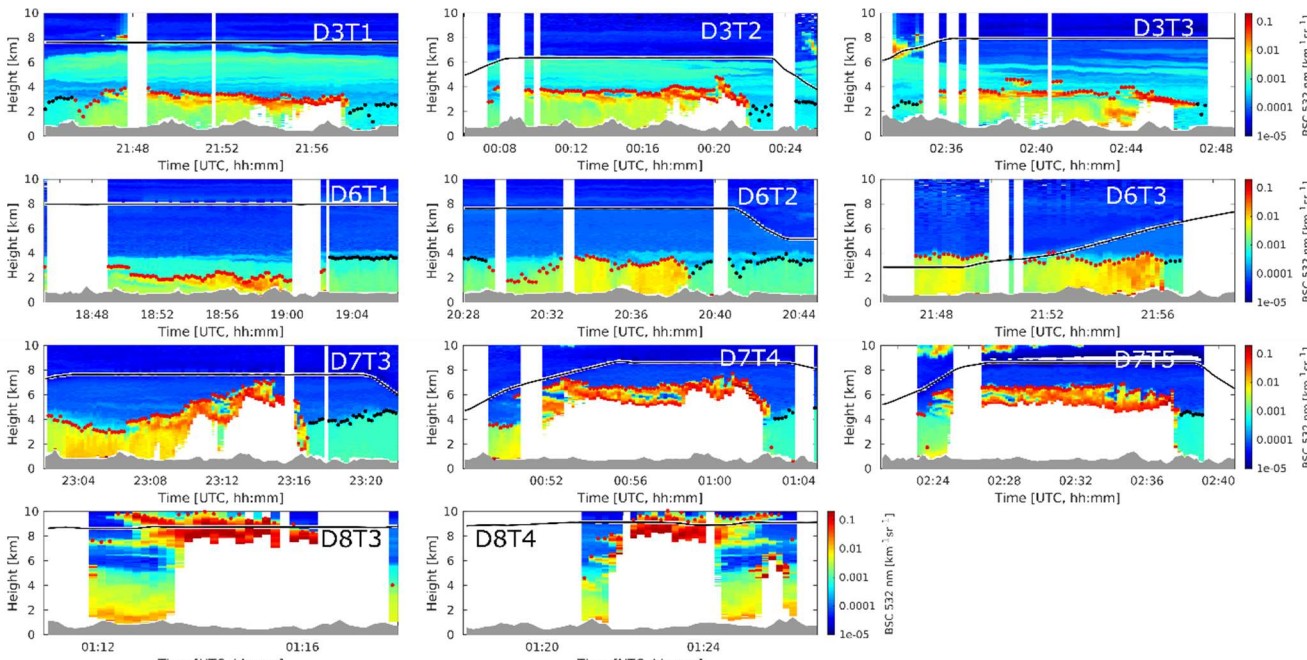

**Figure 15. Curtain plots of backscatter coefficient (532 nm) observed along flight transects. See Table 5 for the details of the selected transects. The red dots represent the $h_{plume}$ determined from the in-plume profiles; the black dots are the $h_{PBL}$ determined from profiles out of plume. The black solid line shows the aircraft altitude.**

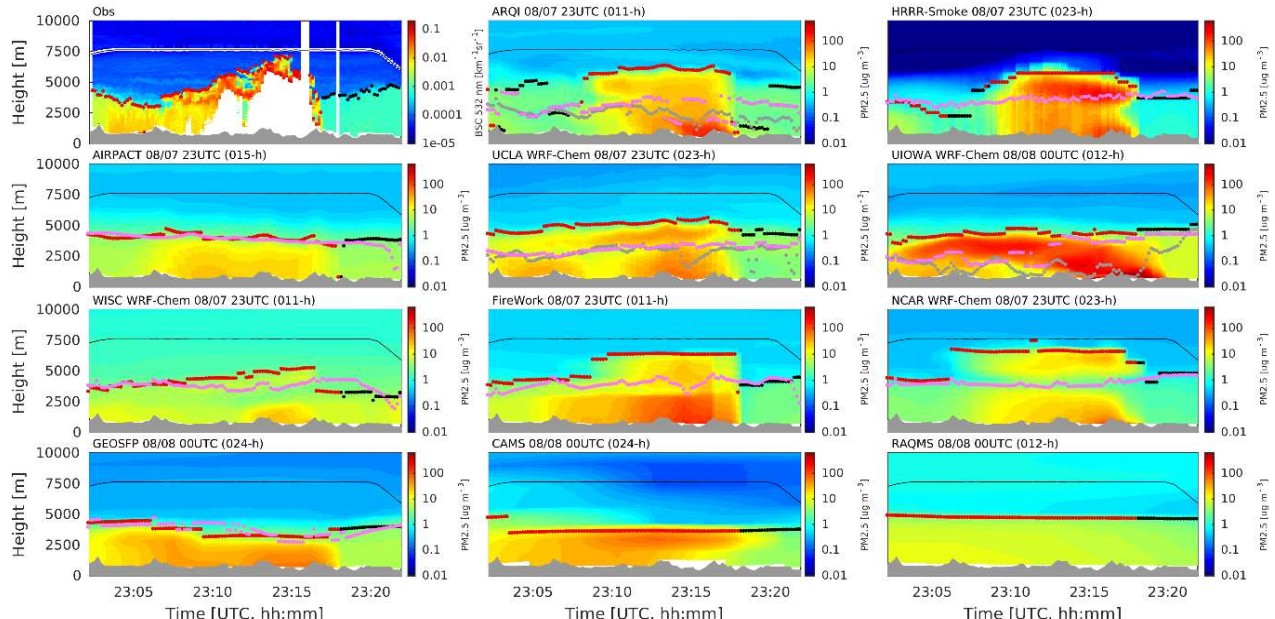

**Figure 16. Comparison of vertical smoke structure based on the DIAL-HSRL observations along the transect D7T3 on 7 August (see details in Table 5) through the smoke plume from the Williams Flats fire. The observed backscatter coefficient at 532 nm (Obs panel) and PM$_{2.5}$ mass concentrations forecasted by different models (model panels) are shown. The red and black dots are plume heights and mixed layer heights determined by using the observed backscatter or modeled PM$_{2.5}$ profiles. The pink dots are PBL heights derived from model diagnosis or forecasted virtual potential temperature (for ARQI, UCLA WRF-Chem, and UIOWA WRF-Chem, and their diagnosed PBL heights by PBL parameterization schemes are denoted by the grey dots). The black line shows the aircraft altitude.**

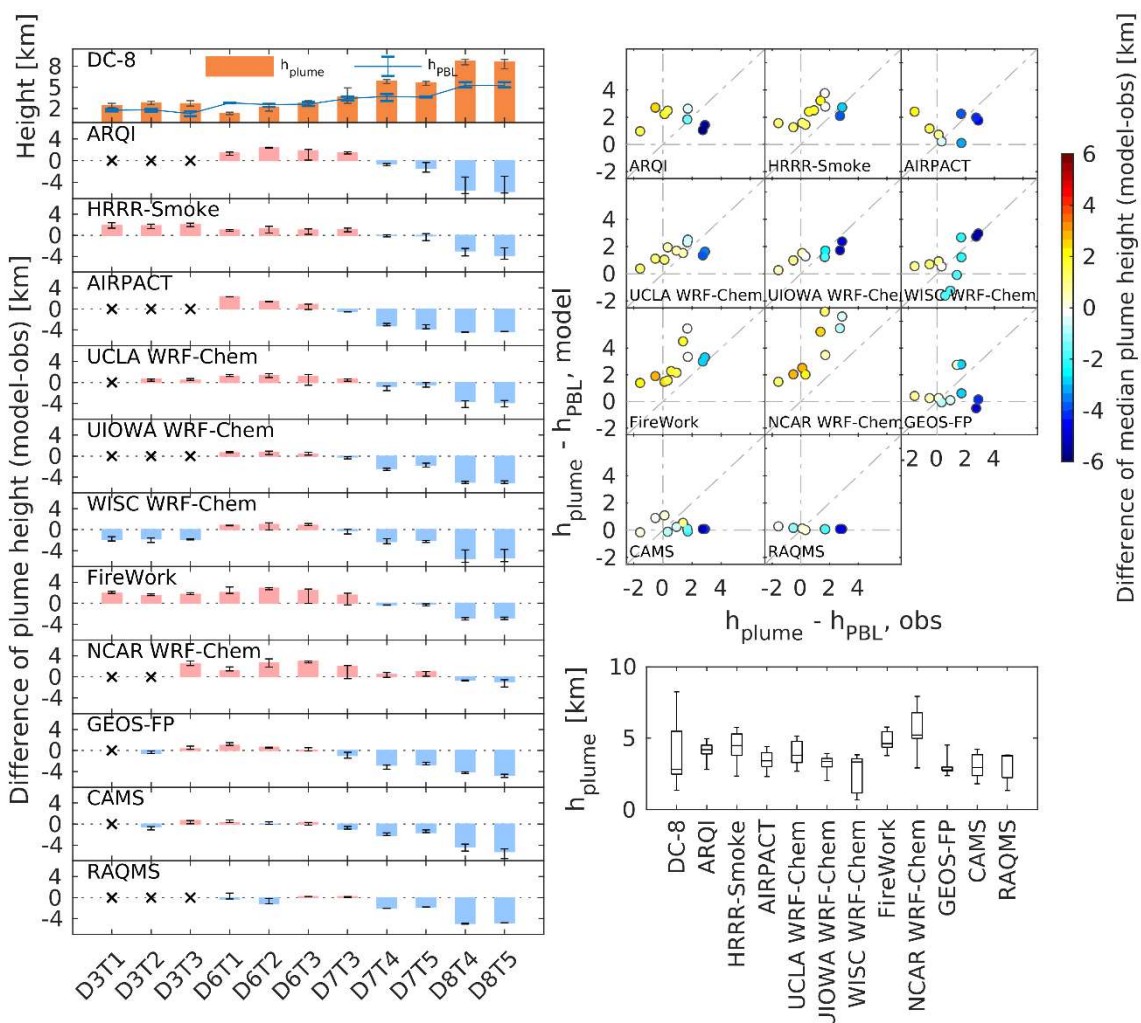

**Figure 17. (a)** Median plume top-heights and PBL heights estimated using DIAL-HSRL observations (top panel) and deviation of modeled median plume heights compared against the observed values along each transect (lower panels). All the heights are shown in units of above ground level. The error bar represents the interquartile range. The "x" signs in model panels denote excluded transects for which the fire had not been active yet. **(b)** Scatterplots of observed and modeled differences between the median plume top-height and median PBL height along each transect. The dots are colored by the difference of modeled and observed median plume top-heights. **(c)** Box plot of the median plume top-heights for the 11 transects for observations and model forecasts. Box borders show the interquartile range, and whiskers are the minimum and maximum.

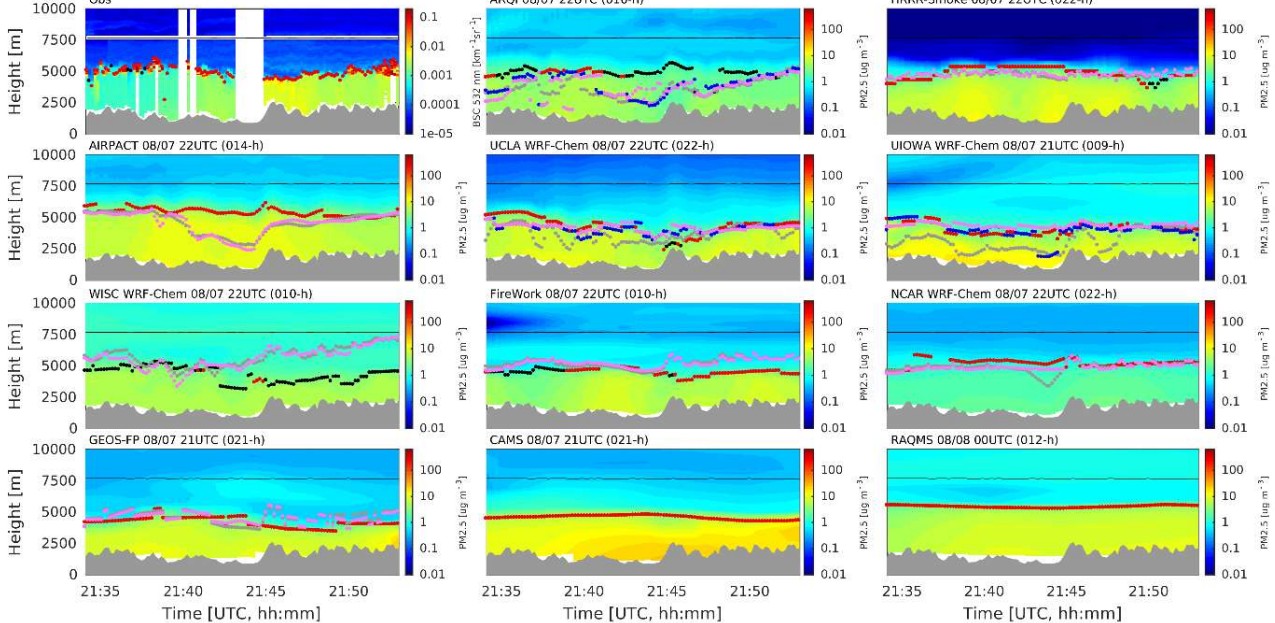

Figure 18. Similar to Fig. 16, but for the transect D7T1 that sampled through the aged plume. The annotations for the colored dots are the same as in Fig. 16, except that the blue dots represent PBL heights estimated by virtual potential temperature for the labeled hour; the pink dots are PBL heights for models at 16:00 PDT.

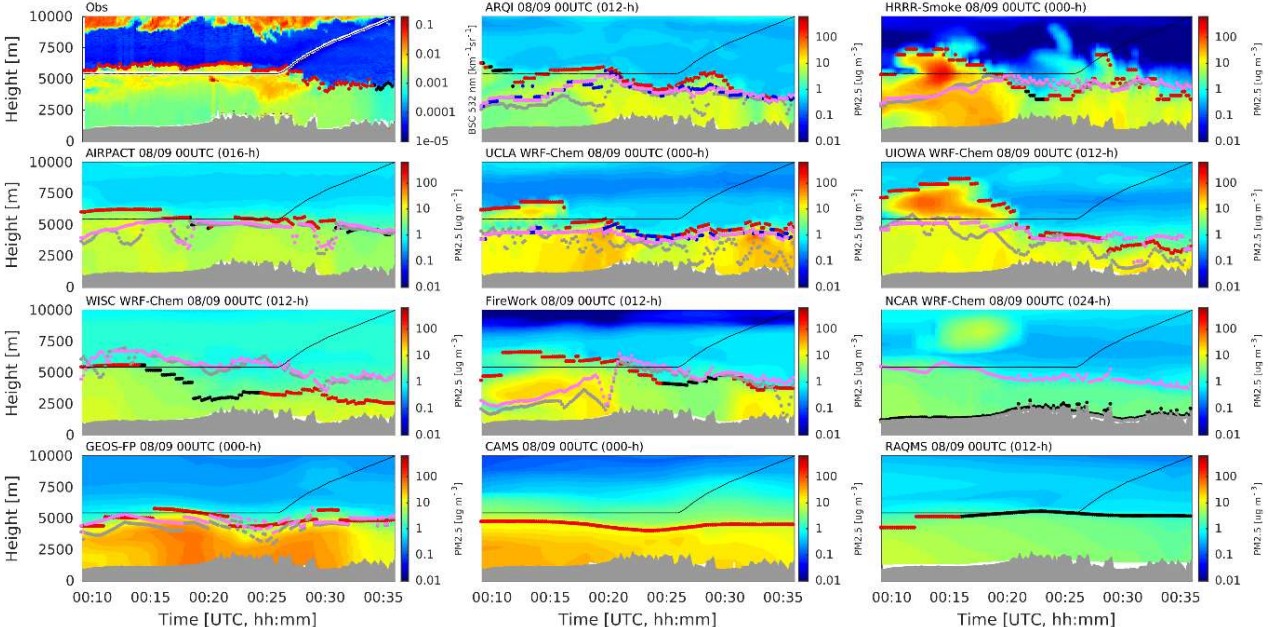

**Figure 19. Similar to Fig. 18, but for the transect D8T2 for the aged plume.**

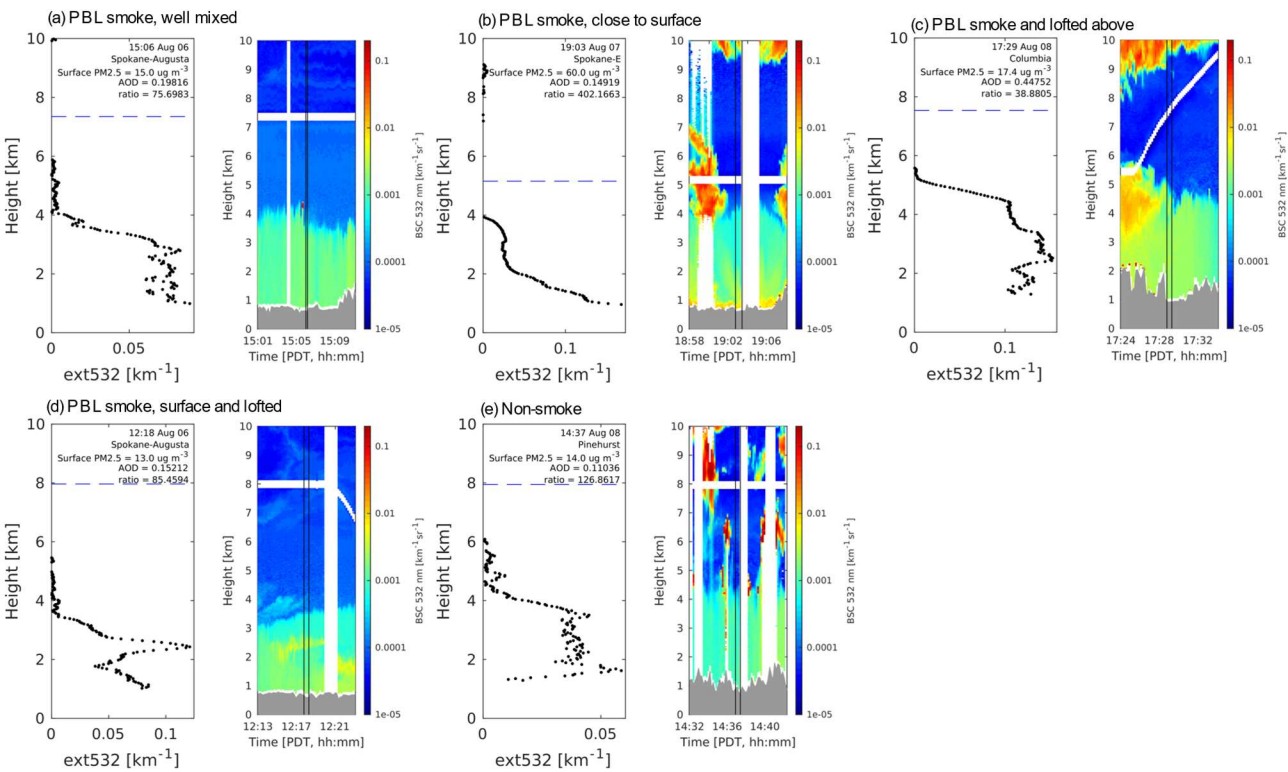

**Figure 20. Example of surface PM$_{2.5}$ to AOD ratios observed under different smoke vertical distributions. The extinction coefficient at 532 nm profile is shown along with the backscatter coefficient distribution over a 10-min flight track. The extinction profiles are averaged from lidar observations during the time slot when the distance from the surface PM$_{2.5}$ monitoring site is less than 5 km, as denoted by the black lines in the color shading plots. The station name, observation time (PDT), surface PM$_{2.5}$, AOD, and their ratio are annotated on the extinction profile plots. The blue dashed line represents aircraft altitude.**

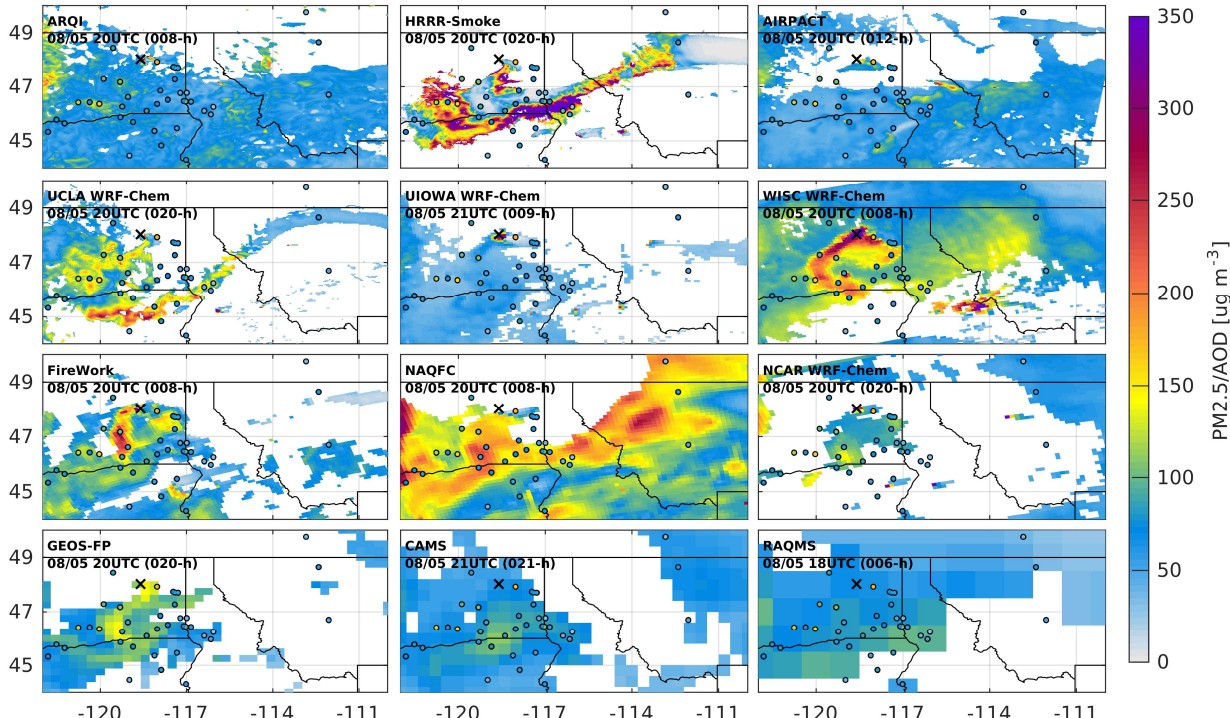

**Figure 21. Maps of surface PM$_{2.5}$ to AOD ratio forecasted by models (color shading) and observed by MODIS MAIAC AOD and PM$_{2.5}$ at surface monitor sites (colored dots) at 20:00 UTC 5 August. The data have been screened based on thresholds of sAOD to extract areas impacted by fire smoke aerosols (see section 3.5.2 for details). The black cross on each panel denotes the location of the Williams Flats fire.**

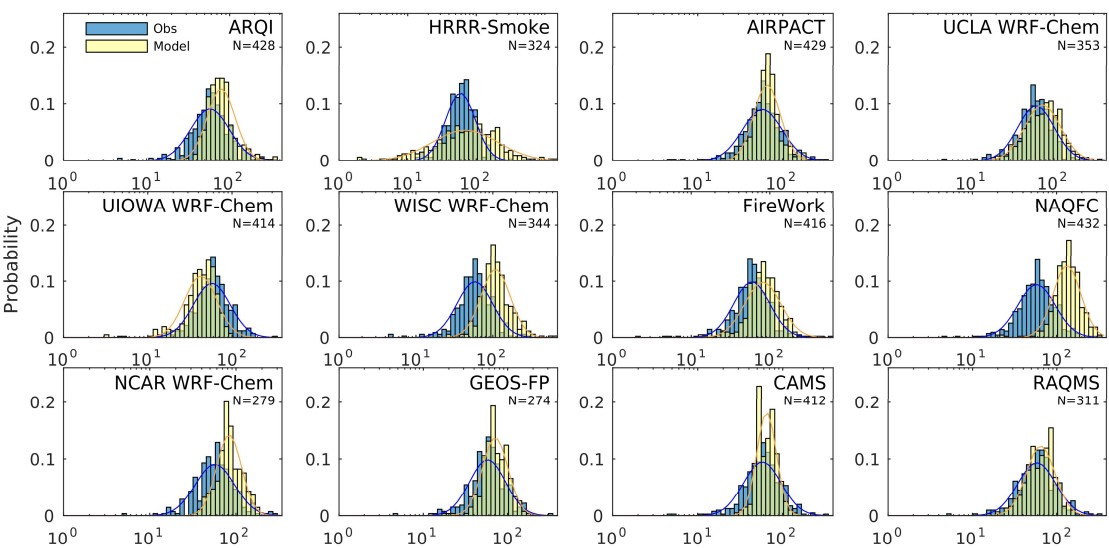

Figure 22. Probability distributions of PM$_{2.5}$/AOD ratios for model and observations. The lines show the curves fitted by log-normal distributions. The data have been screened based on thresholds of sAOD to reflect areas impacted by fire smoke aerosols (see section 3.5.2 for details).

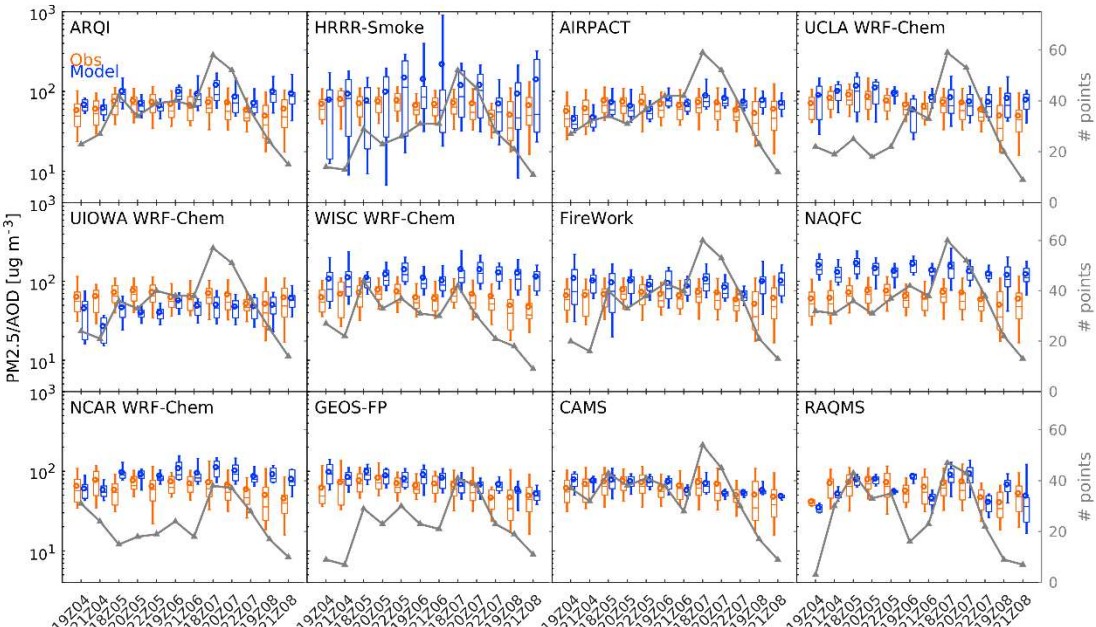

**Figure 23. Boxplots of observed and modeled PM₂.₅/AOD ratios by hours of coincident PM₂.₅ and MODIS AOD data. Data have**
**been filtered based on thresholds of sAOD to focus on impact of fire smoke aerosols. Note that the y-axis is shown in log scale. The**
**central mark of each box indicates the median, and the bottom and top edges of the box are the 25$^{th}$ and 75$^{th}$ percentiles respectively.**
**The whiskers extend to the 10$^{th}$ and 90$^{th}$ percentiles. The grey line shows total number of filtered data pairs for each group of boxes**
**for model and observation.**

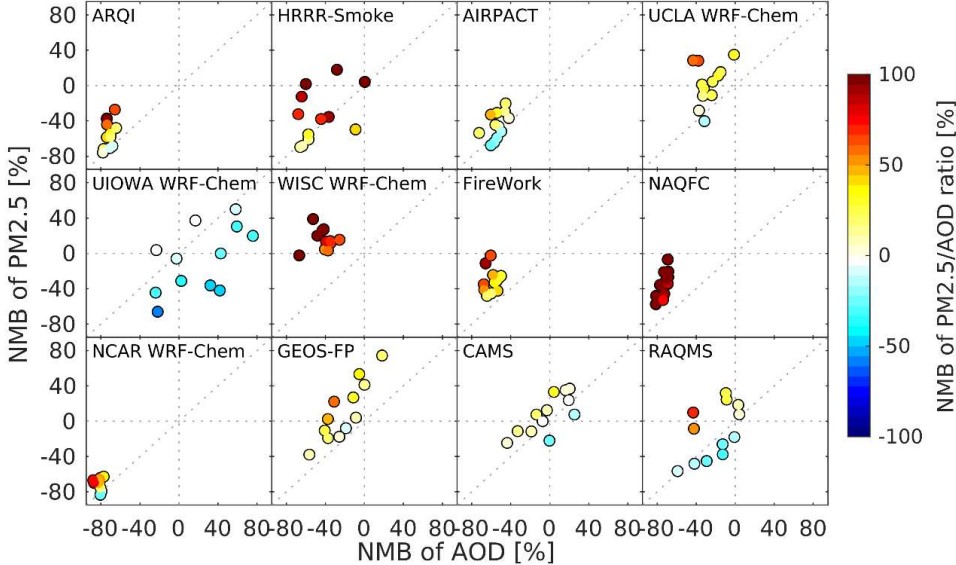

**Figure 24. Scatterplots of normalized mean bias (NMB) of AOD and surface PM₂.₅ for different models evaluated by each hour of**
**coincident PM₂.₅ and MODIS AOD observations. The scattered circles are color-coded by the NMB of PM₂.₅/AOD ratio in that hour.**