# Peer review of "Evaluation and intercomparison of wildfire smoke forecasts from multiple modeling systems for the 2019 Williams Flats fire"

_Atmospheric Chemistry and Physics, 2021_

## Author Comment (AC1)

Responses to RC1

We greatly appreciate the thorough review and helpful comments and suggestions provided by referee #1. The point-by-point responses to the comments are as follows.

The present study highlights the importance of various physical processes in smoke dispersion from wildfire biomass burning. The manuscript is clearly structured, it is well written and presents important analysis regarding the comparison between twelve state-of-the-art atmospheric models during a specific case study in US. Overall, it will be a very useful addition to current literature, and I suggest publication with a few minor comments as shown below:

1. It is clear from this work that there is still significant lack of knowledge regarding several processes like the quantification of fire emissions and their diurnal cycles, plume injection heights, calculation of AOD and PM2.5, the spatiotemporal representation of smoke plumes in forecasting models etc... However, this is something more-or-less known due to the complexity of the mechanisms involved in these events. Since none of the models managed to provide a realistic description of the case study, the study remains somehow non-conclusive. It might be useful to elaborate more in the conclusions section and provide more physical interpretation on the reported differences between the models as well as some quantification on which of the analyzed parameterizations are more important for similar studies (e.g. plume rise, FRP emissions, etc.). This could help future studies to focus on improving certain model features and discard those that look problematic.

**Reply**: Thanks for this suggestion! The conclusion section in the old version was structured by summarizing the main findings in each previous section, and several processes and issues affecting the model performance were mentioned. We recognize that a conclusive statement and more physical explanations are lacking. We have extensively revised the conclusions section (section 4) to elaborate the results, try to give more physical interpretations, and highlight the key factors that are significant for future model improvements. Subtitles are added for a better structure. In brief:

1) Model evaluations of smoke emissions and sAOD suggests the advantage of using FRP-based fire emissions and data assimilation in providing less biased forecasts for total column smoke aerosol loading, compared to other differences in model features.

2) The fact that all estimated fire emissions exhibit a large spread in magnitude demonstrate a need of future work to close the gap between these estimates and reduce their uncertainty.

3) As the forecasted air quality impacts still show limitations due to the persistence assumption, methodologies to predict evolution of fire burning needs to be developed.

4) Forecast performance for sAOD and surface sPM2.5 and their discrepancies highlight a key need to improve plume rise parameterizations, in terms of plume rise height and vertical partition of fire emissions as well, which is closely relevant to accuracy of representation of smoke plumes and the performance discrepancies for sPM2.5 and sAOD.

These have been summarized in section 4.5. Please see the revised version for more details.

We agree that it's necessary to quantify which of the analyzed parameterizations are more important, and it would be instructive and important for similar studies. As these forecast models are drastically different from each other with respect to multiple aspects, we think it would be difficult and arbitrary to

comment on the relative contributions of the abovementioned factors. We are planning on some follow-up work to further identify their relative significance based on sensitivity analysis.

2. The computation of modeled smoke AOD in Section 3.2.2 should be further explained with regards to the optical properties of smoke used in each model.

**Reply**: The treatment of aerosol processes and assumptions of aerosol physical and optical properties have been briefly explained in the descriptions for each model in section 2.1 in the revised paper, and summarized in section 2.2, bullet point 7. The relevant references have also been added. Please refer to the revised paper for more details.

In summary, regarding the diagnosis of AOD, UCLA WRF-Chem, HRRR-Smoke, WISC WRF-Chem, RAQMS, and GEOS-FP used mass-concentration-based aerosol extinction, while CAMS, UIOWA WRF-Chem, NCAR WRF-Chem, NAQFC, AIRPACT, and ARQI used the Mie theory formalism for scattering spheres, and FireWork used a post-processing diagnostic calculation to generate AODs from PM2.5 speciation. We are aware that the uncertainty of AOD calculations owing to the different methods and assumptions can lead to different performance. The assumptions about the chemical species mixing state, density, refractive index, and hygroscopic growth has been estimated to lead to uncertainty in AOD of about 30 – 35 % (Curci et al., 2015), with a tendency of calculated AOD to be less than satellite observations of AOD. Due to the differences in the aerosol models employed and many more differences in meteorology, radiation transfer, chemistry mechanisms, emissions, etc. among the models, this uncertainty introduced by these factors can not be treated explicitly in this work, and it would be necessary to further investigate in our future work by sensitivity analysis.

Curci, G., Ferrero, L., Tuccella, P., Barnaba, F., Angelini, F., Bolzacchini, E., Carbone, C., Denier van der Gon, H. a. C., Facchini, M. C., Gobbi, G. P., Kuenen, J. P. P., Landi, T. C., Perrino, C., Perrone, M. G., Sangiorgi, G. and Stocchi, P.: How much is particulate matter near the ground influenced by upper-level processes within and above the PBL? A summertime case study in Milan (Italy) evidences the distinctive role of nitrate, Atmos. Chem. Phys., 15, 2629–2649, doi:10.5194/acp-15-2629-2015, 2015.

3. At several places (e.g. Lines 89-94, 414-419, 607-610) the authors discuss the importance of including the diurnal variation of smoke emissions inside the forecasting window and the possibility to incorporate data from geostationary satellites. A similar system is available in Europe adopting a modeling strategy of hourly-sequential warm start runs with FLEXPART-WRF, driven by METEOSAT geostationary observations (Solomos et al., 2015, 2019). In this approach, the emissions are updated every hour from the MSG/SEVIRI detections, and each simulation is initialized with the smoke from the previous run (warm start). This provides an efficient way of removing the minor or extinguished fires from the simulation and at the same time to enhance the emissions from the actual burning fires, thus representing the diurnal cycle of biomass burning.

Solomos, S., V. Amiridis, P. Zanis, E. Gerasopoulos, F.I. Sofiou, T. Herekakis, J. Brioude, A. Stohl, R.A. Kahn, C. Kontoes, Smoke dispersion modeling over complex terrain using high resolution meteorological data and satellite observations – The FireHub platform, Atmospheric Environment, Volume 119, October 2015, Pages 348–361, doi:10.1016/j.atmosenv.2015.08.066, 2015

Solomos S., A. Gialitaki, E. Marinou, E. Proestakis, V. Amiridis, Holger Baars, Mika Compula, Albert Ansmann, Modeling and remote sensing of an indirect Pyro-Cb formation and biomass transport from Portugal wildfires towards Europe, Atmospheric Environment, https://doi.org/10.1016/j.atmosenv.2019.03.009 , 2019

**Reply:** Thanks for this comment. This system is very relevant to our study. We have included some brief descriptions of it and added the above references in section 4.5 in the revised manuscript. We also wanted to mention the special challenges in forecasting fire smoke in the U.S. compared to the Europe fires. Specifically, the following sentences have been added:

"As the forecasted air quality impacts still show limitations due to the persistence assumption, methodologies to describe and predict evolution of fire burning needs to be developed. A relevant system is available in Europe adopting a modeling strategy of hourly-sequential warm start runs (Solomos et al., 2015, 2019), with the emissions updated every hour using geostationary satellite detections. This provides an efficient way of removing the minor or extinguished and at the same time to enhance emissions from actual burning fires, thus tracking the diurnal cycle of biomass burning. However, challenges still exist in making assumptions about the fire intensity and spread over the next hours/days. Also, compared to Europe, the fire intensities and their durations are on much larger scales in North America. The large spatial variability of fuels, complex topography, and different ecosystems in the U.S. adds the complexity."

4. Line 320 typo "biome maps"

**Reply**: Thanks for noticing this. It has been corrected to "fuel categories".

---

## Author Comment (AC2)

We greatly appreciate the review and comments to help improve this manuscript. Please see the point-by-point responses as follows.

Overview

This paper compares fire emission parameters derived from twelve different fire emission models forecasting a 2019 wildfire event in the United States. Parameters are compared among models and observed data. By doing so, the authors aim to derive meaningful insights into the current progress of fire emission models and hence suggest ways to further improve the efficacy of such models. The parameters compared in this paper include biomass burning organic carbon emissions, smoke AOD (magnitude and spatial coverage), surface PM2.5, plume rise height and ratio of smoke AOD and PM2.5; hence covering both physics and chemistry aspect of fire emission modeling. The paper suggests areas which current fire emission models can improve on, which includes methodologies to represent diurnal evolution of fire emissions, improved vertical distribution of emitted pollutants and better representation of plume injection heights.

The models and methodology used are clearly described and the paper is well written. I have only two suggestion and a few minor suggestions/clarifications to make.

Major Suggestions

Line 877: It would be highly insightful to understand how the type of emission injection method (within PBL, intermediate, deep) affect the model skill in predicting plume rise heights. Certain emission injection method may be more useful for a certain kind of fire plume (fresh, aged, fire characteristics: smoldering, raging fire, etc.) and not others. If we can associate a better emission injection method with a corresponding type of fire plume, we can improve fire modelling skill. Indeed, further investigations in this regard is necessary and will definitely be a good follow up work.

**Reply:** Thanks for this comment! The plume injection method is a very interesting field and has not been well understood. It is a critical process to be considered and improved in simulation and forecasting of wildfire smoke. During wildfires, convective energy is generated in the lower troposphere due to a large amount of sensible heat and evaporations that happens along with burning process. The smoke plume travels vertically before releasing the majority of smoke gases and particles into the atmosphere. The plume injections are parameterized in models, usually using empirical distributions based on fire severity, or using more physical estimations based on fire intensity (e.g., fire convection energy related to FRP, fire size), fuel type, and meteorological conditions. The different methods have been briefly reviewed in the introduction and Table 1.

As seen in this work, all models failed to represent the large day-to-day variation of plume heights and show narrower spreads compared to the lidar observations. We find that the global models (CAMS, RAQMS, and GEOS-FP) tend to put the emissions within the PBL, which performed well on 3 and 6 August, when most emissions were in the PBL. However, these models could not capture the intensive injections on the 7th and 8th. For the physical parameterizations (e.g. Freitas et al., 2007, 2010), the models tend to better capture the injections above the PBL, although with slight overestimations on the

3$^{rd}$ and 6$^{th}$, and underestimations on the 7$^{th}$ and 8$^{th}$. Therefore, the relation between fire burning activity and plume injections can be very complex. Physical parameterizations show better potential to capture different plume injection scenarios, but more work is needed to improve their inferring method from the input variables (e.g. fire intensity, convection energy, meteorological fields) to plume injections.

It's also noteworthy that even for the models using the same plume injection schemes (e.g. NCAR WRF-Chem, UIOWA WRF-Chem, UCLA WRF-Chem, and WISC WRF-Chem), they show substantially different results which might relate to uncertainties in their input variables (meteorological fields, fire detections) that need to be investigated further.

The analysis has been added to the conclusions in section 4.3. Please refer to the revised manuscript.

Line 981: I may not be proficient enough in this aspect, so this is just some thoughts. There might be inherent problems using surface smoke PM2.5 to smoke AOD ratio when you have different sAOD filters for different models. For models with small sAOD (denominator), the ratio will tend to be bigger and hence result in larger spread. This is consequentially seen in the large nominal mean bias. For example, ARQI and NAQFC have 0.01 sAOD threshold and consequentially have a very large NMB. HRRR smoke and WISC WRF-Chem have 0.02 sAOD threshold and also have very large NMBs. CAMS have a larger threshold, 0.05, and consequentially have smaller magnitude NMBs. AIRPACT is the exception here. This may affect both the magnitude and spread of the ratio calculated and may lead to unfair comparisons between the models. This may affect the model evaluation.

Reply: Thanks for this comment. It's correct that very small sAOD could lead to very large values of the ratio. To avoid the probability distribution being biased due to these extreme values, we chose to use full values (AOD and PM2.5) but not the smoke enhancements (sAOD and aPM2.5) in this section of evaluation.

The threshold of sAOD is only applied to filter the data samples that got impacted by the fire smoke. As mentioned in section 3.5.1 and Fig. 20, the background aerosol conditions can make the relationship ambiguous between the ratio and vertical aerosol allocation. Therefore, we need to filter the samples representing conditions with substantial smoke aerosols in the columns, in order to unambiguously interpret the ratio and column aerosol distribution feature.

However, for model forecasts, due to the remarkably large range of the forecasted fire emissions, there is not a single sAOD threshold that can be appropriate for all models. That's the reason for selecting different thresholds of sAOD. Thus, it's inevitable that the models having very low sAOD enhancements (due to low fire emissions) and low thresholds would show biased ratio distributions. We agree with the limitation, and we have included some statements about this in section 3.5.2:

"It should be noted that for models with relatively low emissions, the areas impacted by smoke plumes cannot be well distinguished from background and other sources (e.g. Fig. 7), thus the distribution of the ratio is higher biased due to inclusion of low background AOD conditions."

"It should be mentioned that for models that had relatively low fire emissions, as noted earlier (AIRPACT, ARQI, NAQFC, and NCAR WRF-Chem), the distributions can't unambiguously represent smoke plumes from the Williams Flats fire and are likely driven by other sources and background aerosols, so

the results are not exactly comparable with the observations. These models tend to overpredict the ratios, which is consistent with the larger ratios obtained for non-smoke cases mentioned as shown in the previous subsection."

And the result about shifted distribution of modeled ratios compared to observations has been stated to be valid only for models with significant impacts of fire smoke:

"For the models with prominent smoke AOD impacts, a shift in the distribution of ratio compared to observations could be explained by issues in assumptions for aerosol optical properties (e.g., a too high mass extinction efficiency can bias the ratio distribution towards the lower end) or biases in the PBL heights (e.g., shallower PBL can lead to positively biased ratios)."

**Minor Suggestions**

Line 36: For 2.5 in PM2.5, suggest to be written in subscript.

**Reply**: We have revised through the manuscript to make "2.5" written in subscript.

Line 85: There is one multi-model comparison done by Li, et. al., 2019. Atmos. Chem. Phys., 19, 12545–12567 for many different fire models, which may be worthy to look at.

**Reply**: Thanks for recommending this paper. The paper by Li et al. reported constructions of global historical fire emissions, based on nine dynamic global vegetation models (DVGMs) with interactive fire modeling on long-term and large-scale fire emissions. It has been added in the introduction section.

Line 157: If the forecast system produces more than 1 cycle per day, how is the data treated? Is the data averaged?

**Reply**: The coincident predictions at the closest hour relative to the observation times were derived from the most recent forecast cycle (initialized within 24 hours). The initial time of the forecast cycles used for the models are included in Table 1.

Line 220: The style of writing for Section 2.1.6 seems to be slightly different from the rest of the paper. Consider revising.

**Reply**: Section 2.1.6 has been revised. The reason it looks slightly different from other models might be that it's the model we ran by our own group using a simplified microphysics scheme, which is different than the other models based on WRF-Chem. This model also assimilated satellite AOD data to improve fire emissions.

Line 378: I would like to clarify if the models were in a spun-up condition when model forecasts were extracted to compare with observed data.

**Reply**: The spin-up condition results have been excluded. This has been noted in section 3.2.1 in the revision.

Line 446: AERONET is already defined in line 435.

**Reply**: Thanks for noticing this, the duplication of definition has been revised.

Line 534: It may be insightful to suggest a reason why FRP-driven models results in higher sAOD compared to hotspot driven models.

Reply: The higher sAOD as predicted using the FRP-based fire emissions is in consistency with the relative emissions magnitudes. As shown in section 3.1, the FRP-based emissions are overall higher than the satellite-fire-detection-based emissions. This has been added to the revised version.

Line 655: It may be problematic to compare point derived ground-based measurement station data against model grid predictions of smoke PM2.5. Perhaps a small discussion about this issue will be helpful.

Reply: We have added some sentences to note this issue:

"The $PM_{2.5}$ observations are hourly averages reported at the end of each hour, namely centered on the half hour. For the modeled counterparts, surface $PM_{2.5}$ values are derived by bilinearly interpolating the modeled 2-D forecasts at the lowest model level onto the latitude and longitude coordinates of each monitoring station. The inconsistency in spatiotemporal representations of the model forecasts and observations could be a source of model-observation differences."

Line 660 and a few other places: May want to revise the use of 'it's'.

**Reply:** We have checked through the manuscript and revised "it's" into "it is".

Line 730: Consider revising this sentence.

Reply: Thanks for this comment. This sentence and relevant analysis have been revised:

"The diurnal variation of the average $sPM_{2.5}$ for the Ot sites also show patterns deviating from the observations. Most models (except for UCLA WRF-Chem, NCAR WRF-Chem, and HRRR-Smoke) show a common feature of a valley in the afternoon, however it is not seen for the observations. It could be a joint consequence of the larger dispersion volume owing to higher PBL depth than reality, and/or issues with other sources, e.g. anthropogenic activities."

Figure 5: Spelling of AERONET in the figure and caption.

**Reply**: The spelling has been corrected in the figure and caption.